# Etiology of oncogenic fusions in 5,190 childhood cancers and its clinical and therapeutic implication

Yanling Liu[1], Jonathon Klein[2], Richa Bajpai[2], Li Dong[1], Quang Tran[1], Pandurang Kolekar[1], Jenny L. Smith[3], Rhonda E. Ries[3], Benjamin J. Huang[4], Yi-Cheng Wang[5], Todd A. Alonzo[6], Liqing Tian[1], Heather L. Mulder[1], Timothy I. Shaw[7], Jing Ma[8], Michael P. Walsh[8], Guangchun Song[8], Tamara Westover[8], Robert J. Autry[9,13,14], Alexander M. Gout[1], David A. Wheeler[1], Shibiao Wan[10], Gang Wu[10], Jun J. Yang[9], William E. Evans[9], Mignon Loh[11], John Easton[1], Jinghui Zhang[1], Jeffery M. Klco[8] ✉, Soheil Meshinchi[3] ✉, Patrick A. Brown[12] ✉, Shondra M. Pruett-Miller[2] ✉ & Xiaotu Ma[1] ✉

Oncogenic fusions formed through chromosomal rearrangements are hallmarks of childhood cancer that define cancer subtype, predict outcome, persist through treatment, and can be ideal therapeutic targets. However, mechanistic understanding of the etiology of oncogenic fusions remains elusive. Here we report a comprehensive detection of 272 oncogenic fusion gene pairs by using tumor transcriptome sequencing data from 5190 childhood cancer patients. We identify diverse factors, including translation frame, protein domain, splicing, and gene length, that shape the formation of oncogenic fusions. Our mathematical modeling reveals a strong link between differential selection pressure and clinical outcome in *CBFB-MYH11*. We discover 4 oncogenic fusions, including *RUNX1-RUNX1T1*, *TCF3-PBX1*, *CBFA2T3-GLIS2*, and *KMT2A-AFDN*, with promoter-hijacking-like features that may offer alternative strategies for therapeutic targeting. We uncover extensive alternative splicing in oncogenic fusions including *KMT2A-MLLT3*, *KMT2A-MLLT10*, *C11orf95-RELA*, *NUP98-NSD1*, *KMT2A-AFDN* and *ETV6-RUNX1*. We discover neo splice sites in 18 oncogenic fusion gene pairs and demonstrate that such splice sites confer therapeutic vulnerability for etiology-based genome editing. Our study reveals general principles on the etiology of oncogenic fusions in childhood cancer and suggests profound clinical implications including etiology-based risk stratification and genome-editing-based therapeutics.

Since the discovery of Philadelphia chromosome in chronic myeloid leukemia[1,2], intensive efforts to decipher the genetic underpinnings of both adult[3–5] and childhood[6,7] cancers have uncovered numerous cancer driver alterations including oncogenic fusions. Longitudinal genomics studies[8,9] on patient tumors under therapeutic intervention have further revealed comprehensive insights into the clonal evolution of tumors[10] where cancer-driving alterations can be eradicated by therapy or de novo acquired[8,9]. In these patients, subtype-defining

A full list of affiliations appears at the end of the paper. ✉e-mail: Jeffery.Klco@stjude.org; smeshinc@fredhutch.org; Patrick.Brown@bms.com; Shondra.Miller@stjude.org; Xiaotu.Ma@stjude.org

oncogenic fusions (e.g., *BCR-ABL1* in Philadelphia chromosome-positive patients[1,2]) typically persist through the lifetime of a tumor[8,9] and can serve as stable biomarkers for curative outcomes. Moreover, successes in targeted inhibition of oncogenic fusions (e.g., imatinib for *BCR-ABL1*[11]) have inspired the notion of "oncogene addiction"[12] that posits the therapeutic potential of targeting oncogenic fusions.

The promise of therapeutically targeting oncogenic fusions is further reinforced by the recent development of genome-editing tools enabled by CRISPR-Cas9 system[13], where highly specific targeted editing can be achieved using a guide RNA complementary to the target DNA. Therefore, a natural hypothesis is that oncogenic fusions can harbor therapeutic vulnerability for genome editing-based therapeutics. However, a comprehensive mechanistic understanding of the etiology of oncogenic fusions remains elusive. What molecular factors contribute to the formation and prevalence variability of oncogenic fusions? What are the expression characteristics of involved genes? Are oncogenic fusions subject to alternative splicing? Do oncogenic fusions encode information on natural selection? Most importantly, are there any immediate therapeutic vulnerabilities in oncogenic fusions? To address these questions, and with the consideration of childhood cancers are underrepresented in targeted therapies[14], we comprehensively studied oncogenic fusions by using tumor transcriptome sequencing datasets of 5190 patients from publicly available childhood cancer cohorts.

Here, we show general forces shaping the formation of oncogenic fusions, including gene length, splicing, translation, and protein domains. We introduce a mathematical model that can detect differential selection pressure, which is validated in *CBFB-MYH11*-positive acute myeloid leukemia (AML) to confer superior prognostic value on

event-free survival (EFS) than other well-known clinical features. We discover a subset of oncogenic fusions to harbor promoter-hijacking-like features that may offer a unique opportunity for developing drug targets with reduced toxicity to normal cells of the corresponding lineage. We uncover a subset of oncogenic fusions that are subject to alternative splicing regulation. Importantly, we discover oncogenic fusions utilizing neo-splice sites that are not observed in normal tissues. By in vitro CRISPR-Cas9 editing in relevant cell lines, we demonstrate that these splice sites confer therapeutic vulnerabilities for genome editing. In summary, our study reveals general principles governing the etiology of oncogenic fusions and suggests profound clinical implications including etiology-based risk stratification and genome-editing-based therapeutics.

## Results

### Model of fusion etiology and study design

Oncogenic fusions typically involve two genomic loci (genes) denoted as N' gene (N-terminus) and C' gene (C-terminus). We enumerated theoretically possible scenarios of gene fusion (Fig. 1a), where intron/exon structure and translation frame are the main constraints. This theoretical analysis revealed five fusion categories: (1) neo-translational, where part of the untranslated region (5' UTR) in C' gene is converted into a coding region; (2) intronic versioning, where multiple introns are available to form slightly different fusion proteins; (3) neo-splicing, where the DNA breakpoint disrupts the natural splicing structure and neo-splice sites and cryptic exons are created to compensate such disruption; (4) chimeric exon, when DNA breakpoints fall into the coding regions of both N' gene and C' gene; and (5) promoter/enhancer hijacking (e.g., *IGH-CRLF2* in B-lineage acute lymphoblastic leukemia (B-ALL)[15]). These five categories encompassed all possible combinations of promoter/enhancer, intron-intron, intron-exon, and exon-exon rearrangements. In this work, we focused on categories (1)–(4) by using tumor RNAseq data from 5190 childhood cancer patients (Fig. 1b) because promoter/enhancer hijacking does not create a chimeric protein per se and DNA sequencing data are required for an unbiased study (Supplementary Note 8b). Candidate oncogenic fusions were detected using methods (Arriba[16], STAR-Fusion[17], CICERO[18], and FusionCatcher[19]; Methods) reported to have superior performance[18,20]. A workflow was developed to identify oncogenic fusions from a large number of predictions of these methods (Methods, Supplementary Notes 8–12).

To classify the detected oncogenic fusions into one of the above four categories, we developed tools Neo-Versioner (to classify intronic versioning) and Neo-Splicer (to classify neo-splicing; see Methods and Supplementary Fig. 1). If the fusion does not belong to either intronic versioning (i.e., no reads supporting natural exonic junctions between N' and C' genes) or neo-splicing categories by our automated analysis, we manually review and classify the fusion into the categories of either chimeric exon or neo-translational.

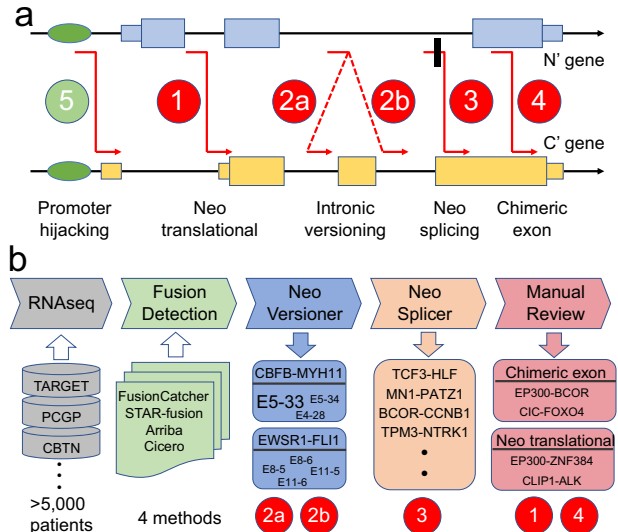

**Fig. 1 | Model of fusion etiology and study design. a** Theoretical mechanisms of oncogenic fusion formation. Scenario 1: the DNA breakpoints (red lines) can lead to the fusion of coding exons (thick boxes) from N' gene to 5' untranslated region (UTR; thin boxes) of C' gene and result in the conversion of the corresponding UTR into coding region, hence "neo-translational". Scenario 2: the DNA breakpoints can lead to the fusion of a coding exon from N' gene to multiple possible coding exons of C' gene, hence "intronic versioning". Scenario 3: the DNA breakpoints falling into a coding exon may disrupt the normal splice sites, and the cancer cell may utilize a neo-splice site to ensure the inclusion of the corresponding exon, hence "neo-splicing". In this scenario, a cryptic exon (black box) might be created. Scenario 4: the DNA breakpoints may directly fuse two coding exons, hence "chimeric exon". Scenario 5: a well-known phenomenon is promoter/enhancer hijacking, which is not studied in this work because it does not lead to chimeric protein. **b** Study design. We analyzed tumor RNA sequencing data using four fusion detection methods, and classified the detected fusions into intronic versioning, neo-splicing, neo-translational, and chimeric exon (see Methods).

### Landscape of childhood oncogenic fusions

We detected 2012 oncogenic fusion events (from 2005 patients and involving 272 gene pairs) in our cohort of 5190 childhood cancer patients (Fig. 2a and Supplementary Data 1–3), including 55.7% of leukemia (1470/2638), 22.5% of brain tumor (329/1459) and 18.8% of solid tumor (206/1093) patients, respectively. Among the 2005 fusion-positive patients, only 7 of them (0.35%) have two different fusions, such as *BCR-ABL1* and *CBFB-MYH11* in patient PARBLV, *FGFR1-TACC1* and *FGFR3-TACC3* in patient PT_7DTGJYA7, *KMT2A-MATR3* and *MEF2D-DAZAP1* in patient SJBALL020141 and SJBALL020142, *CBFB-MYH11* and *RUNX1-RUNX1T1* in patient SJCBF124 and SJCBF149, *KLHL7-MET* and *VCL-NTRK2* in patient SJHGG009. Among the oncogenic fusions, we detected 34 neo-splicing (Fig. 2b and Supplementary Data 4), 33 neo-translational (Fig. 2c and Supplementary Data 5) and 11 chimeric exon (Fig. 2d and Supplementary Data 6) events. The remaining fusions

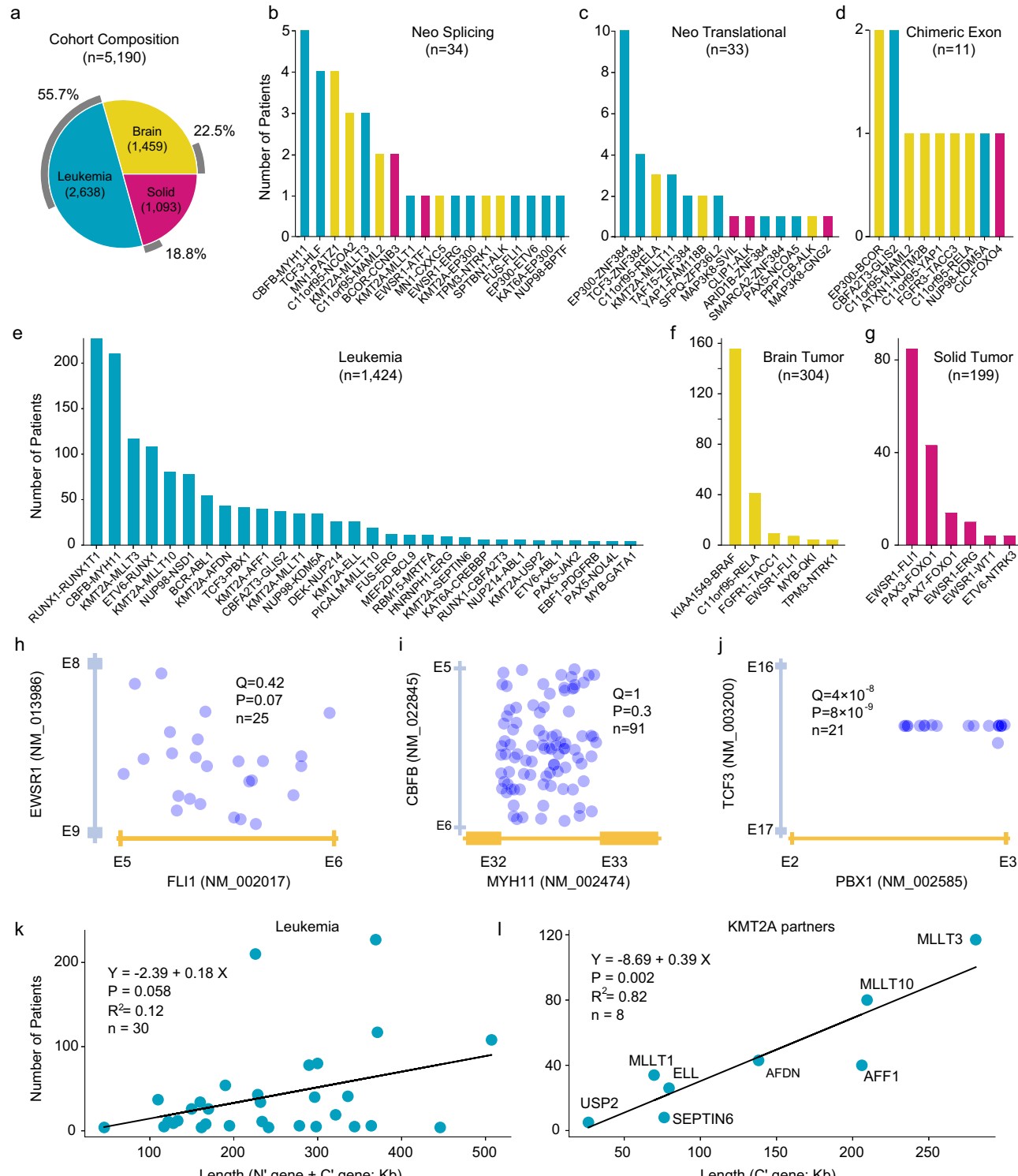

**Fig. 2 | Fusion landscape in childhood cancers. a** Cohort composition. We analyzed 2638 leukemia (blue), 1459 brain tumors (yellow), and 1093 solid tumors (magenta), totaling 5190 childhood cancer patients. Percent patient tumors with oncogenic fusion detected are indicated with gray rings. **b–d** Spectrum of neosplicing (**b**), neo-translational (**c**) and chimeric exon (**d**) fusions. **e–g** Spectrum of canonical fusions in leukemia (**e**), brain tumor (**f**) and solid tumor (**g**). In panels **b–g**, bars are color-coded according to tumor types in panel **a**. Distribution of DNA breakpoints (light blue dots) for oncogenic fusions is uniformly distributed in corresponding intronic regions for *EWSR1-FLI1* (**h**) and *CBFB-MYH11* (**i**), but not for

*TCF3-PBX1* (**j**). *P* values (and *Q* values after Bonferroni correction for multiple testing) of the uniformity test (two-sided Kolmogorov–Smirnov test; see Methods) are indicated along with sample size in panels **h–j**. **k** Prevalence of oncogenic fusions (*y*-axis) demonstrates a weak but marginally significant ($P = 0.058$) association with gene length (*x*-axis) in leukemia. **l** Statistically significant ($P = 0.002$) association between prevalence and gene length when the analysis is conditional on *KMT2A*-rearranged leukemia. Linear model, *P* value and $R^2$ value are indicated for panels **k–l**. Source data are provided accordingly as files **a–g**, **h–j** and **k–l** in Source Data file.

from 1927 patients (Supplementary Data 7) belong to the category of intronic versioning for leukemia ($n = 1424$), brain tumor ($n = 304$), and solid tumor ($n = 199$), respectively (Fig. 2e–g; see Supplementary Fig. 2 for a complete map of recurrent ($n \geq 3$) childhood oncogenic fusions). Leukemias have the highest number of recurrent (observed in >3 patients) oncogenic fusions ($n = 30$), followed by brain tumor ($n = 6$) and solid tumor ($n = 6$).

Among the 272 detected oncogenic fusion gene pairs, the prevalence of oncogenic fusions varies considerably. For example, in leukemia, *RUNX1-RUNX1T1* was observed in 227 patients, while *KMT2A-ELL* was observed in 26 patients (Fig. 2e–g). Although it has been noted that the prevalence of oncogenic fusions varies widely[21,22], no systematic studies on potential mechanisms are proposed in literature. Since oncogenic fusions are formed through DNA rearrangements (breakpoints), we studied the distribution of DNA breakpoints predicted from RNAseq data with a detection rate of 40.5% (813 out of 2005 patients) and accuracy rate >90% as validated by DNA sequencing data in applicable cases (see Methods; Supplementary Data 8, Supplementary Fig. 3a, and Supplementary Note 9). We focused on six oncogenic fusions (*RUNX1-RUNX1T1*, *CBFB-MYH11*, *EWSR1-FLI1*, *ETV6-RUNX1*, *NUP98-NSD1*, *TCF3-PBX1*) with DNA breakpoint detected in more than 20 patient tumors (to allow power for two-dimensional Kolmogorov–Smirnov test; Methods; Fig. 2h–j and Supplementary Fig. 3b–d). We discovered in *TCF3-PBX1* fusion that the DNA breakpoints tend to cluster in intron 16 of *TCF3* and the second half of intron 2 of *PBX1* (Fig. 2j), which is consistent with previous observation[23]. Next, we discovered that DNA breakpoints are clustered in the promoter region (upstream to first exon) but not in the first intron of *RUNX1T1* in oncogenic fusion *RUNX1-RUNX1T1* (Supplementary Fig. 3d), despite these DNA breakpoints, all generate the same fusion protein by using *RUNX1T1* starting from exon 2 (Supplementary Data 7). Therefore, some local DNA properties must have facilitated the formation of such rearrangements in intron 16 of *TCF3* and in *RUNX1T1* promoter region. By contrast, we discovered four oncogenic fusions, including *EWSR1-FLI1* (Fig. 2h), *CBFB-MYH11* (Fig. 2i), *ETV6-RUNX1* (Supplementary Fig. 3b) and *NUP98-NSD1* (Supplementary Fig. 3c), to demonstrate a near-uniform distribution ($Q > 0.1$ after Bonferroni correction) of DNA breakpoints in relevant introns. Among all 42 fusions (Fig. 2e–g), only 2 fusions (<5%) were detected with a clustered distribution of DNA breakpoints (Supplementary Data 8), which indicates that, for most fusions, every base pair of the intronic region can contribute to the formation of oncogenic fusions via rearrangement. In another word, longer introns may increase the prevalence of corresponding oncogenic fusions. This leads to our hypothesis that gene (intron) length may play a role in patient prevalence. We analyzed oncogenic fusions for leukemia, solid tumor, and brain tumor, separately. A marginally significant linear association (Fig. 2k and Supplementary Fig. 3e; $R^2 = 0.12$; $P = 0.058$, $n = 30$) was observed between patient prevalence and total length of the involved gene pairs, and no significance was observed in brain and solid tumors (Supplementary Fig. 3f, g). By limiting the analysis to leukemia with oncogenic fusions with ≥5 fusion partners (*KMT2A*[24], *ETV6*, and *PAX5*), we obtained a statistically significant linear association for *KMT2A* either when fusions with recurrence >3 were considered ($R^2 = 0.82$; $P = 0.002$; $n = 8$; Fig. 2l) or when fusions with recurrence >1 were considered ($R^2 = 0.86$; $P = 1.5 \times 10^{-5}$; $n = 12$; Supplementary Fig. 3i) but not for *ETV6* and *PAX5* ($P > 0.1$; Supplementary Fig. 3j, k). The above observations are also observed if only involved introns are considered (Supplementary Fig. 3l–r). Excluding *KMT2A* fusions resulted in an insignificant association in leukemia ($P = 0.22$; $n = 22$; Supplementary Fig. 4). The overall insignificant association between gene length and patient prevalence in oncogenic fusions (except *KMT2A*) indicated that additional molecular factors (such as protein domain, frame, and splicing) might play a major role in the formation or selection of oncogenic fusions, as will be demonstrated next.

## Expression patterns of oncogenic fusions

Inspired by the fusions formed by promoter/enhancer hijacking (e.g., *IGH-CRLF2* or *IGH-DUX4* fusion in B-ALL[25]) that leads to aberrant activation of a target gene that otherwise is silenced in the corresponding normal lineage of the corresponding tumor, we studied the expression characteristics of the recurrent ($n \geq 10$; to allow power to detect differential expression) fusions by measuring the relative expression ratio (in $\log_2$ scale) between C′ gene and N′ gene using the fused portion (Methods; Fig. 3a) with an expression dominance score (EDS), where a low EDS score indicates that the C′ gene is expressed at a lower level than that of the N′ gene. To account for the diverse tissue origins, we categorized samples of matched cancer types (Supplementary Data 1, 7) into three groups: (1) samples with the fusion-of-interest; (2) samples with fusions other than the fusion-of-interest; (3) samples without known fusions. In the first group, the fused portion of the C′ gene must be expressed because of the fusion, while in the second and third groups, the C′ gene may or may not be expressed, and these two groups can cross-validate each other. As a result, the EDS score fluctuates between −1.9 and 0.8 (95% confidence interval of medians) among samples with the fusion-of-interest (Fig. 3b and Supplementary Data 9). On the other hand, the median EDS score in samples of groups 2 and 3 can be as low as −10 in fusions including *RUNX1-RUNX1T1*, *TCF3-PBX1*, *CBFA2T3-GLIS2* and *KMT2A-AFDN*, indicating that corresponding C′ genes are typically not expressed in normal cells of host lineages. For example, gene *PBX1* (in fact, only the fusion portion) is expressed in B-ALL sample SJE2A059 that harbors *TCF3-PBX1* fusion but is not expressed in B-ALL sample SJBALL021772, which is *TCF3-PBX1* negative (Fig. 3c). By contrast, *NSD1* is constitutively expressed in AML samples both positive (SJAML064746) and negative (SJAML064774) for *NUP98-NSD1* fusion (Fig. 3d). With this observation, we performed Wilcox rank sum test (one-sided) of EDS scores between group 1 and group 2 and 3 samples followed by Bonferroni correction for multiple testing. As it turned out, nine intronic versioning fusions demonstrated a statistically significant difference in EDS scores between group 1 and group 2 and 3 samples (Fig. 3b, asterisks). These fusions are therefore classified as "promoter-hijacking-like fusions", and the remaining fusions are classified as conventional chimerism. Because these fusions generate chimeric proteins, this group of promoter-hijacking-like fusions is fundamentally different from conventional promoter-hijacking fusions (e.g., *IGH-CRLF2/EPOR/DUX4*) where no chimeric proteins are involved. By further calculating expression level measured as fragments per kilobase of exon per million mapped fragments (FPKM; Methods) of the corresponding C′ genes (Supplementary Fig. 5c), we confirmed that *RUNX1T1* in *RUNX1-RUNX1T1*, *PBX1* in *TCF3-PBX1*, *GLIS2* in *CBFA2T3-GLIS2*, and *AFDN* in *KMT2A-AFDN* have FPKM < 1 and are considered non-expressed in the normal lineage of corresponding cancer subtypes (blue asterisks in Fig. 3b). Interestingly, we also observed several fusions, including *KIAA1549-BRAF*, *C11orf95-RELA*, and *PAX3/PAX7-FOXO1*, to have higher EDS score in group 2 and 3 samples, indicating a highly active role of the C′ genes (*BRAF*, *RELA*, *FOXO1*) in corresponding normal lineages. Collectively, because the C′ gene is silenced or lowly expressed in the corresponding normal lineage for promoter-hijacking-like fusions, we propose that corresponding C′ genes (*RUNX1T1*, *PBX1*, *GLIS2*, *AFDN*, *FLI1*, and *MYH11*) can serve as good drug targets because the expected "on-target, off-tumor" toxicity can be minimal to the normal cells of the corresponding lineage of cancer cells (but not for other normal cells where the C′ gene has housekeeping expression). By contrast, the "on-target, off-tumor" toxicity can be much higher in the conventional chimerism group. To corroborate our findings, we further studied expression patterns of these fusions in 9525 RNAseq samples from healthy donors in GTEx database[26]. As it turned out, blood samples from GTEx data provided clear support for oncogenic fusion *TCF3-PBX1* where *PBX1* is not expressed (Supplementary Fig. 5b and Supplementary Data 9 and 10). Of note, analysis of GTEx samples without discriminating tissue source did not provide

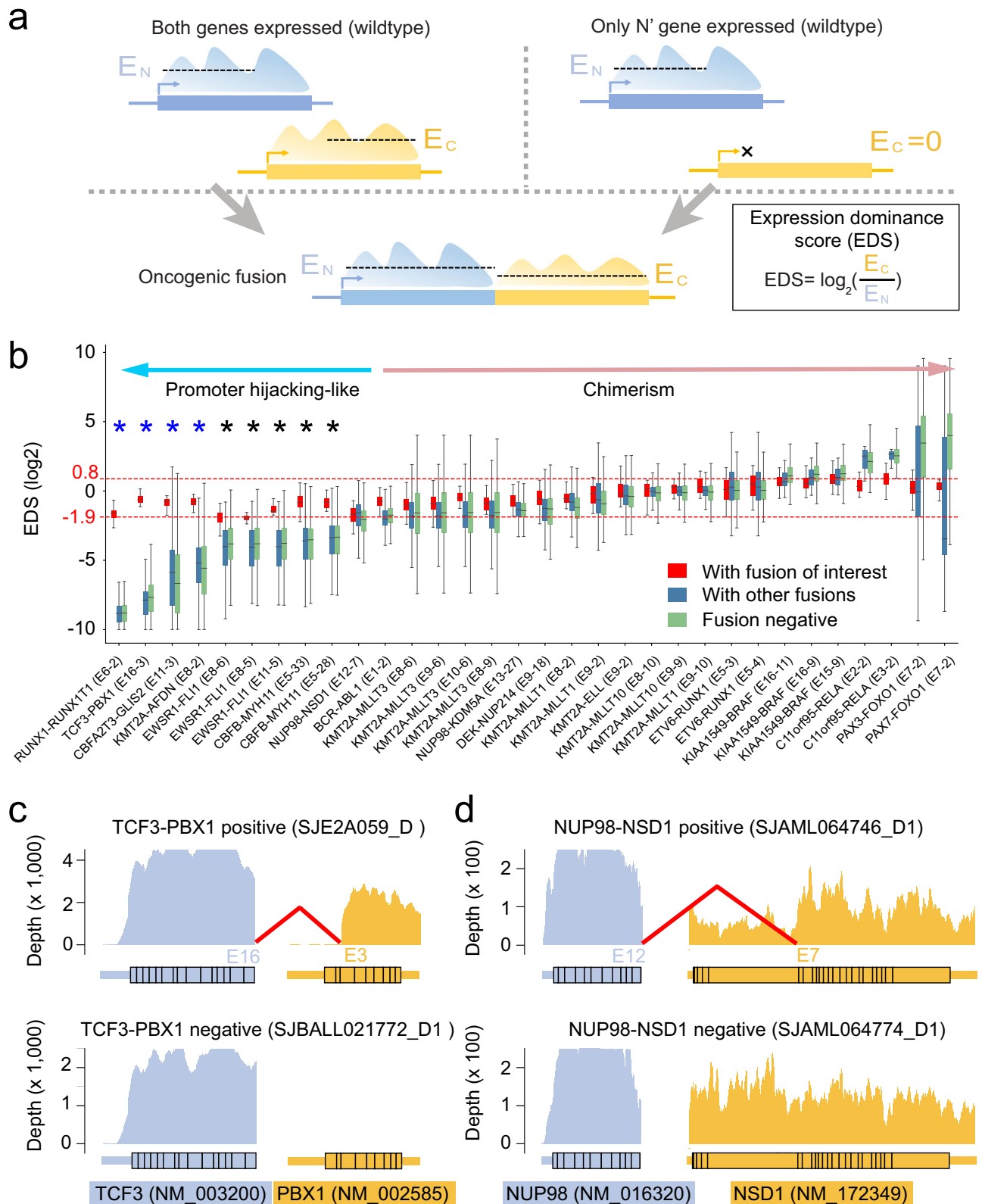

**a**

Both genes expressed (wildtype)

$E_N$

Only N' gene expressed (wildtype)

$E_N$

$E_C$

$E_C = 0$

Oncogenic fusion

$E_N$ $E_C$

Expression dominance score (EDS)

$$EDS = \log_2\left(\frac{E_C}{E_N}\right)$$

**b**

EDS (log2)

Promoter hijacking-like        Chimerism

- With fusion of interest
- With other fusions
- Fusion negative

**c**

TCF3-PBX1 positive (SJE2A059_D )

Depth (x 1,000)

E16        E3

TCF3-PBX1 negative (SJBALL021772_D1 )

Depth (x 1,000)

TCF3 (NM_003200)        PBX1 (NM_002585)

**d**

NUP98-NSD1 positive (SJAML064746_D1)

Depth (x 100)

E12        E7

NUP98-NSD1 negative (SJAML064774_D1)

Depth (x 100)

NUP98 (NM_016320)        NSD1 (NM_172349)

support for our conclusion (Supplementary Fig. 5a), indicating the importance of proper classification of RNAseq samples according to tissue origin. Indeed, top oncogenic fusions demonstrating promoter-hijacking-like features (Fig. 3b) are mostly from childhood AML which did not have a normal counterpart in GTEx cohort and therefore no conclusion can be made from these oncogenic fusions (Supplementary Fig. 5b and Supplementary Data 1 and 10).

**Alternative splicing in oncogenic fusions**

Since alternative splicing is a general phenomenon in normal physiological conditions[27], we next asked if alternative splicing can play a role in oncogenic fusions. As shown in Fig. 4a, an oncogenic fusion may or may not be subject to regulation by alternative splicing. Toward this possibility, we designed a splicing dominance score (SDS) to measure the percentage of junction reads that supports the canonical splicing

**Fig. 3 | Expression of oncogenic fusions. a** Expression model. For oncogenic fusions, promoters of N' genes are constitutionally active, while promoters of the C' gene may or may not be constitutionally active. We propose an expression dominance score (EDS) to measure the ratio of expression level (median sequencing depth) of chimeric portions between C' gene and N' gene for each tumor. **b** Distribution of EDS scores for oncogenic fusions in samples (of matched lineages) with the fusion of interest (red), with other fusions (blue), and negative for fusions (green). In boxplot, the lower, center and upper limits indicate 25th, 50th, and 75th percentile, respectively. Whisker is defined using 1.5 IQR. Dotted horizontal red lines indicate 95% confidence interval of EDS scores determined in fusion-positive samples. Based on EDS scores, oncogenic fusions were classified into promoter-

hijacking-like and chimerism groups. Asterisks indicate $Q$ value <0.01 (one-sided Wilcox rank sum test after Bonferroni correction for multiple testing, $n = 32$), where blue asterisks indicate C' genes considered non-expressed in fusion-negative samples (FPKM <1; Supplementary Fig. 5c). Also illustrated are example samples from promoter-hijacking-like category where the chimeric portion of C' gene is only expressed in the fusion-positive (top, E16-E3) samples (**c**), and from chimerism category where the chimeric portion of C' gene is expressed in both fusion-positive (top, E12-E7) and fusion-negative (bottom) samples (**d**). Y-axis indicates RNA sequencing depth in panels **c** and **d**. Source data are provided accordingly as sheets **b** and **c** and **d** in Source Data file.

($X_1$) over all junction reads spanning exons in N' gene and exons in C' gene (Fig. 4a, black and red arrows; Supplementary Data 11). To determine whether alternative splicing is dependent on the rearrangement, we also studied tumor samples without the fusion of interest, in which we calculate the SDS score as the percentage of canonical splicing over all junctions that encompass the involved intron of N' gene and C' genes, respectively (Supplementary Fig. 6a). By applying our method to all recurrent ($n > 3$) fusions, we discovered that majority (78%) of oncogenic fusions are not subject to regulation by alternative splicing (Fig. 4b, c). Interestingly, fusions involving *KMT2A* appear to be strongly affected by alternative splicing (Fig. 4b). The detailed splicing patterns of three representative oncogenic fusions are illustrated in Fig. 4d, where alternative usage of exon 10 in *KMT2A* is clearly observed in both *KMT2A*-rearranged AML tumors and AML tumors without *KMT2A* fusions. By contrast, fusion *NUP98-KDM5A* is not subject to alternative splicing. On the other hand, *CBFB-MYH11* appears to have negligible (<1%) alternative splicing caused by weak exon 5 (of *CBFB*) skipping that is observed in both fusion-positive tumors and tumors without *CBFB-MYH11*. For exons involved in alternative splicing, we also investigated whether they could match any known isoforms of the host gene. We detected only one recurrent alternative exon (exon 12 in NM_016320) of *NUP98* that matches a second isoform NM_001365129 (Supplementary Data 11). This data indicated that alternative splicing is likely a property of the host gene that is not affected by somatic alterations for oncogenic fusions. To further study whether this is true in non-cancer tissues, we performed our analysis in 9525 RNAseq samples from healthy donors in GTEx[26]. As it turned out, alternative splicing in *ETV6-RUNX1* (identified in B-cell leukemia) is recapitulated in *RUNX1* gene in normal GTEx blood samples, and alternative splicing in *C11orf95-RELA* (identified in Ependymoma, a brain tumor) is recapitulated in *C11orf95* in normal GTEx brain samples (Supplementary Fig. 6c). Of note, alternative splicing in these genes is "averaged out" when all 9525 GTEx samples are used indiscriminately (Supplementary Fig. 6b). Indeed, alternative splicing involving *KMT2A* is not recapitulated in GTEx dataset by our analysis, which is reflected by the lack of myeloid specimens in GTEx samples (Supplementary Data 10). Together, our data indicated a clear involvement of alternative splicing in 22% of oncogenic fusion events, although such regulation is not specific to tumors and therefore is likely an intrinsic property of the host gene. It would be interesting to functionally study whether the exons subject to alternative splicing (such as *KMT2A* exon 10) are dispensable to the host cancer cells, which may have profound implications for drug designing.

## Selection bias in intronic versioning

Because intronic versioning can cause amino acid differences in the fusion protein, which may, in turn, lead to potential functional differences, we hypothesized that intronic versioning could confer differential fitness to the host cells in some oncogenic fusions (Supplementary Note 13). To measure the effect size of potential selection bias between two intronic versions, we proposed a relative selection bias (RSB) score (Methods) based on the observation that DNA breakpoints are distributed in relevant introns in a near-uniform fashion in 95% fusions (Fig. 2j–l, Supplementary Fig. 3b–d, and

Supplementary Data 8). In this model, the patient prevalence of DNA breakpoints falling in an intron should be proportional to its length if the resultant protein versions are functionally equivalent (i.e., confers the same positive selection pressure). The statistical significance of selection bias is measured by comparing (using a $\chi^2$ test) the observed patient prevalence in all intronic versions and the corresponding expected patient prevalence under a null hypothesis that patient prevalence is proportional to intronic length. When the involved exon (Fig. 5a, star) encodes functionally important protein domain and thus leads to higher positive selection pressure, its corresponding intron will have disproportionately high patient prevalence.

A critical constraint to gene fusion products is splicing and translation, which is clearly illustrated by *CBFB-MYH11* fusion in childhood AML. Here we defined the translational frame for each coding exon by using the codon frame of its first base. Because all six coding exons of *CBFB* have a length of 3n ≡ 0 (modulo 3, the remainder of a division with denominator 3 equals 0), *CBFB* have all exons in frame 0. On the other hand, *MYH11* has exon frames encompassing all three possibilities of 0, 1, and 2. Although numerous exonic combinations (Fig. 5c, gray lines) can theoretically generate in-frame proteins, in patients, we only observed a limited variety of fusion versions (Fig. 5c, red lines), including E5-33 ($n = 183/214$), E5-28 ($n = 17/214$), etc. These data also indicate a potential selection bias due to critical protein domains encoded by involved exons. To illustrate this hypothesis, we generated a circuit plot to highlight intron length, where the N' gene is placed on y-axis and C' gene is placed on x-axis (Fig. 5d), and the axes are proportional to gene length (see scale bars). Conditional on exon 5 of *CBFB*, we can see a clear discrepancy between patient prevalence and intronic length for different intronic versions: intron 32 of *MYH11* (corresponding to fusion version E5-33, $n = 183$ patients having this version) has a length of only 370 bps, while intron 27 (corresponding to fusion version E5-28, $n = 17$ patients) has a longer length of 5509 bps. With this data, we observed a RSB score of 160.3, indicating a strong positive selection pressure for version E5-33 relative to version E5-28 ($\chi^2$ Q < $7.7 \times 10^{-15}$; Supplementary Data 12).

To validate our hypothesis that fusion versioning may influence clinical outcome, we compared hazard ratios for EFS across the *CBFB-MYH11* AML cohort as a function of fusion versions and several well-established prognostic variables, including exon 17 *KIT* mutation status, WBC count at diagnosis, patient age at diagnosis, and initial response to therapy as measured by end of induction 1 (EOI1) minimal residual disease. We discovered E5-33 version of fusion *CBFB-MYH11* is the best prognostic variable in this analysis (Fig. 5e), followed by exon 17 *KIT* mutation status, confirming our hypothesis that selection bias in version E5-33 can predict clinical outcome[28].

Among 19 oncogenic fusions with intronic versioning (Supplementary Data 12), *KMT2A* and *C11orf95* fusions were excluded due to strong alternative splicing (Fig. 5b, asterisks); *RUNX1-RUNX1T1* was excluded due to potentially biased DNA rearrangements before selection), selection bias is detected in five oncogenic fusions at false discovery rate Q < 0.01. In addition to *CBFB-MYH11*, *ETV6-RUNX1* and *KIAA1549-BRAF* have high patient prevalence in both major and minor intronic versions and are illustrated in Fig. 5f. By contract, fusion

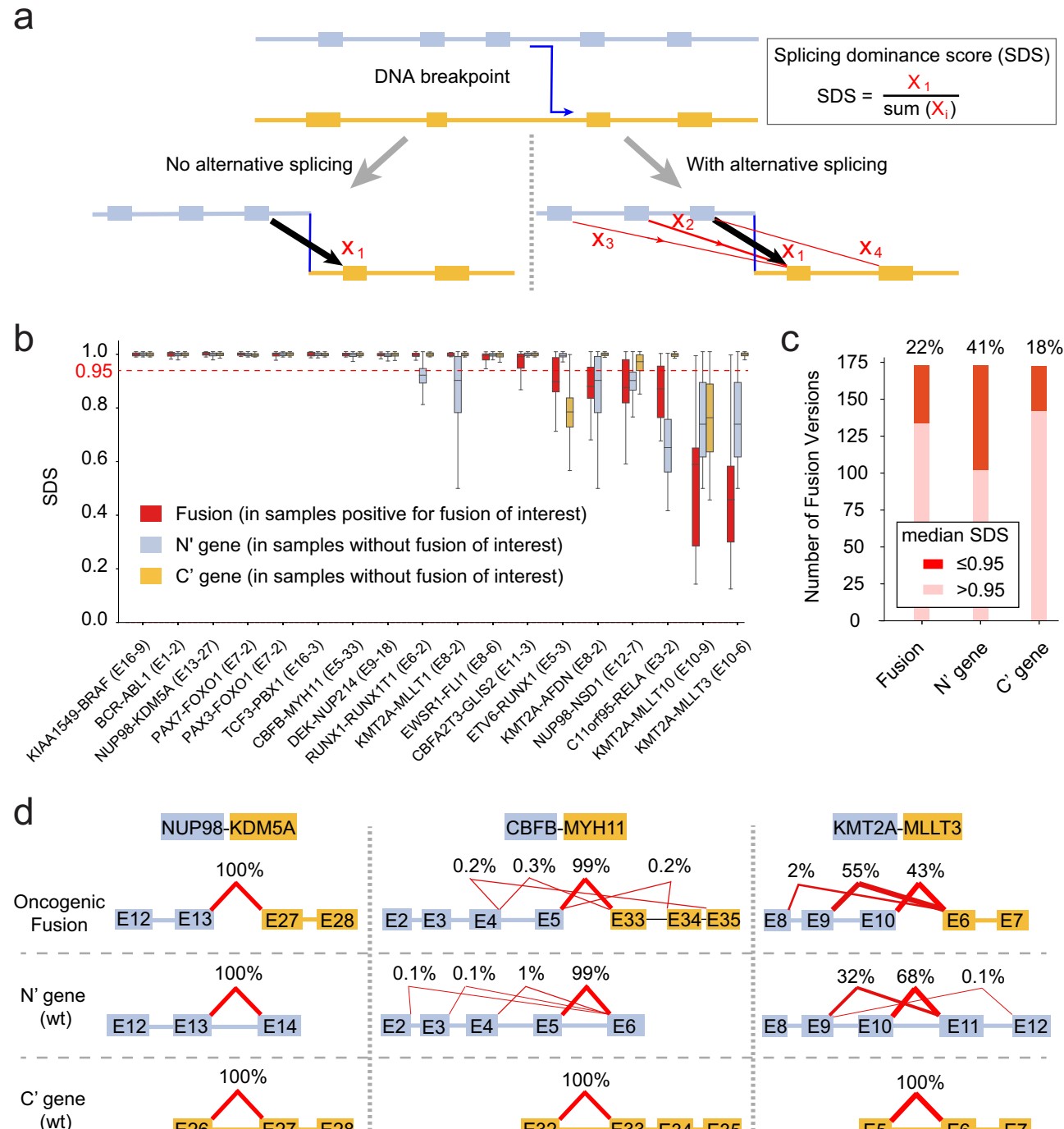

**Fig. 4 | Alternative splicing in oncogenic fusions. a** Splicing model. The oncogenic fusion defined by a DNA breakpoint (blue line) may or may not be subject to alternative splicing (red lines). We propose a splicing dominance score (SDS) to measure the alternative splicing as a ratio of the count of splicing reads supporting the canonical splicing pattern ($X_1$) to the count of splicing reads spanning both the N' gene and the C' gene ($X_1$–$X_4$). A similar score is defined for the wildtype genes (Methods; Supplementary Fig. 6). **b** SDS score distribution for fusion genes (red) and wildtype N' (blue) and C' (orange) genes for 18 intronic versioning with recurrence ≥10, where alternative splicing (SDS <0.95, red dashed line) is observed in 6 (33% of 18) fusions. In boxplot, the lower, center and upper limits indicate 25th, 50th, and 75th percentile, respectively. Whisker is defined using 1.5 IQR. **c** A similar extent of alternative splicing is observed in 183 intronic versions with recurrence >2. **d** Example oncogenic fusions and splicing patterns. Splicing patterns for wild-type N' (blue) and C' genes (orange) are also presented. Black connections indicate canonical splicing, while red connections indicate alternative splicing. Source data are provided in sheets **b–d** in Source Data file.

*EWSR1-FLI1* (Fig. 5f) demonstrated an insignificant *Q* value of 0.1 (after Bonferroni correction for multiple testing). Two additional fusions (*NUP214-ABL1* and *BCR-ABL1*) also demonstrate a significant *Q* value <0.01. Of note, intronic versioning in *BCR-ABL1* has been associated with differential lymphoid leukemogenic activity in mice[29]. The low patient prevalence (*n* = 5) for *NUP214-ABL1* can render the statistical

analysis less robust. Collectively, these data highlight the important role of protein domain selection in shaping the patient prevalence of oncogenic fusions. Furthermore, intronic versioning analysis provided us with tools to study the potential functional importance of protein domains that might serve as therapeutic targets and prognostic biomarkers.

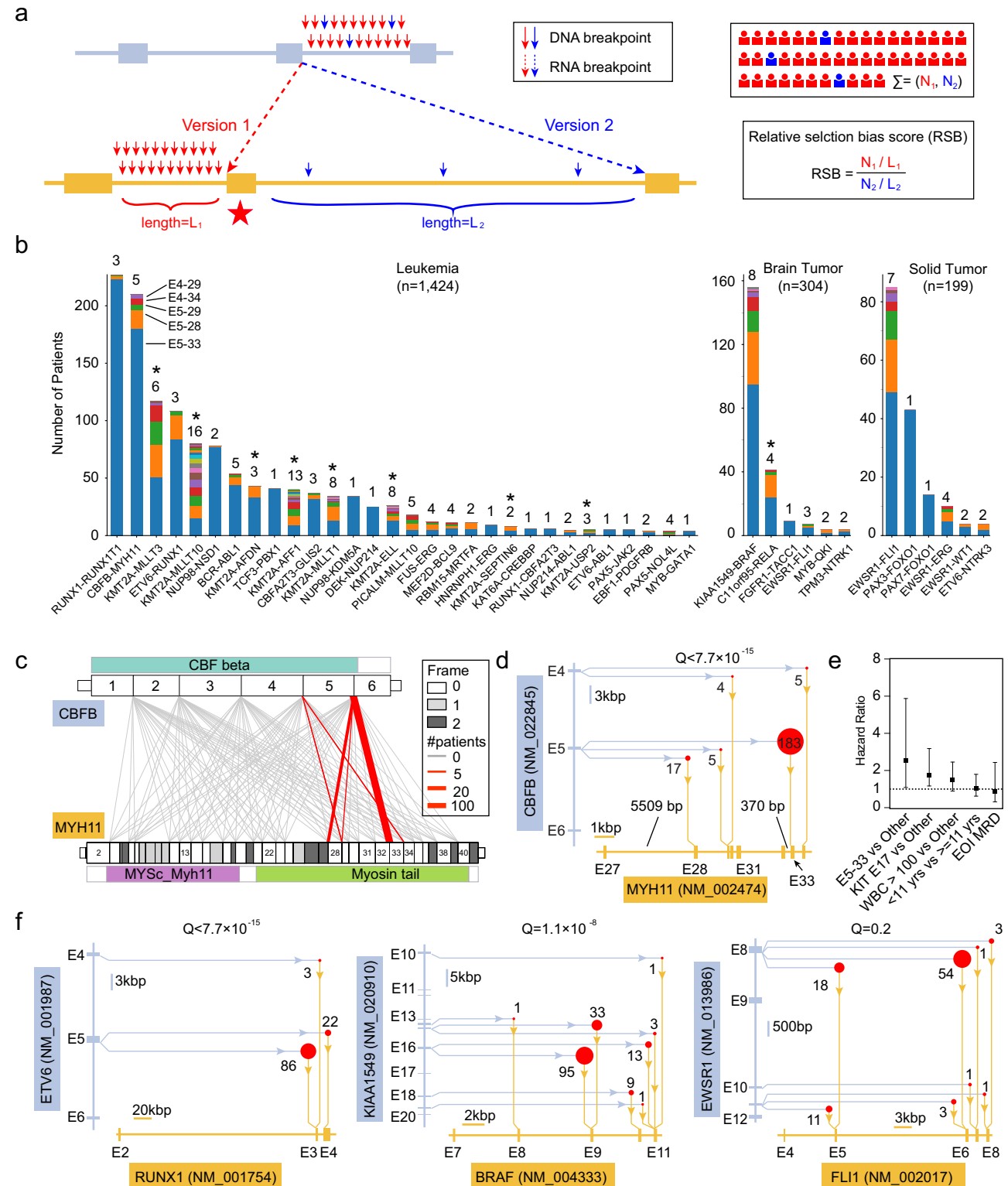

## Neo-splicing in oncogenic fusions

We detected oncogenic fusions harboring neo-splicing in 34 patients (Fig. 2b; see all neo-splicing samples in Supplementary Fig. 7; Supplementary Data 4). For example, brain tumor PT_E3ADF4ZB harbored oncogenic fusion *MN1-PATZ1*, where the DNA breakpoint resides in exon 1 of *PATZ1* and disrupts the normal splicing acceptor (Supplementary Fig. 7a, where all four patients with *MN1-PATZ1* in our cohort belong to neo-splicing category). To compensate this disruption, the cancer cell created a neo-splice acceptor (AG) at 26 base pairs

upstream of the DNA breakpoint in intron 1 of *MN1* gene. This example clearly indicated the flexibility of splicing machinery in recognizing neo-splice sites. Among the oncogenic fusions with neo-splicing, we discovered three patient tumors with *TCF3-HLF* fusion involved neo-splicing between exon 16 of *TCF3* and exon 4 of *HLF* (Fig. 6a), indicating a common mechanism governing the expression of this fusion. Indeed, close examination indicated that exon 16 of *TCF3* and exon 4 of *HLF* have incompatible translation frames (Supplementary Fig. 7e). Therefore, the neo-splice sites and corresponding cryptic exons are created

**Fig. 5 | Selection bias in oncogenic fusions. a** Model of selection. DNA breakpoints from the same intron have equivalent selection pressure because they generate the same fusion proteins. DNA breakpoints from different introns may have different selection pressure when the variable exon (red star) encodes critical protein domains and the corresponding intron may have disproportionally more patients than other introns. We propose a relative selection bias (RSB) score to measure such imbalance by accounting for patient counts (N) and intronic lengths (L) for intronic versions (red vs blue). **b** Spectrum of intronic versioning (colored bands within bar) across leukemia, brain, and solid tumors. Oncogenic fusions may have a single version (*TCF3-PBX1*) or multiple versions (number of versions labeled on top of each bar). Colors indicate different versions (exact fusion versions are indicated for *CBFB-MYH11*). Oncogenic fusions with alternative splicing are indicated by asterisks (*) and excluded from selection bias analysis. **c** Theoretically possible (gray lines) and observed (red lines) intronic versions in *CBFB-MYH11*. We define the translation frames (0, 1, 2) for each exon by using the frame position of its first nucleotide. A functional oncogenic fusion can only be generated by connecting translationally compatible exons (gray lines). Due to additional requirement of protein domains (e.g., Myosin Tail domain in *MYH11*), only a subset of

translationally compatible fusions can result in tumorigenesis (red lines), although patient prevalence can be dramatically different (thickness of red lines). **d** Analysis of selection pressure in *CBFB-MYH11*. Version E5-33 has a disproportionally high number of patients ($n = 183$) than version E5-28 ($n = 17$) although the corresponding intron 32 of *MYH11* has a length of 370 bps and intron 27 has a length of 5509 bps, indicating a strong selection bias (RBS = 160.3) between E5-33 and E5-28 (with a $\chi^2$ test $Q$ value $<7.7 \times 10^{-15}$). **e** Intronic versioning (E5-33) better predicts event-free survival (measured as hazard ratio) than known clinical features (*KIT* mutation status, while blood counts, age, and end of induction (EOI) MRD) for *CBFB-MYH11*-positive AML patients. Error bars represent hazard ratio ± 95% confidence interval. **f** Analysis of selection bias in *ETV6-RUNX1*, *KIAA1549-BRAF*, and *EWSR1-FLI1* fusions. In panels **d** and **f**, x-axes are the C′ genes, and y-axes are the N′ genes. Exon/intron lengths are indicated with scale bars in corresponding figures. Sizes of red dots are proportional to the number of patients for corresponding versions, and $\chi^2$ test $Q$ values (with Bonferroni correction for multiple testing) are indicated for each panel. Source data are provided accordingly as sheets **b** and **c**, and **d**, **f** and **I** in Source Data file.

by the host cancer cell to compensate the translation problem. Consistent with the previous report[30], in our dataset, we also discovered one patient SJALL018389 (Supplementary Data 7) to harbor natural in-frame fusion between exon 15 of *TCF3* and exon 4 of *HLF*. In this sample, a weak alternative splicing between exon 14 of *TCF3* and exon 4 of *HLF* is observed. Analysis of published RNAseq[31,32] data on *TCF3-HLF* cells with E15-4 version under various drug treatments (JQ1, A-485)[32] further confirmed that this fusion version is subject to a weak alternative splicing regulation (Supplementary Data 11). Although it has been suggested that the cryptic exons function to make up the translation frame problem in *TCF3-HLF* by the cancer cells[33,34], there is no functional evidence available to date. In the next, we sought to investigate the function of this cryptic exon and corresponding hypothetical neo-splice sites through CRISPR-based genome editing.

## CRISPR targeting of neo-splicing

B-ALL patients with *TCF3-HLF* fusion are currently considered incurable[31]. A *TCF3-HLF*-positive cell line HAL-01 (see Methods) harbors the neo-splicing pattern (Fig. 6b and Supplementary Data 4) and provides us an in vitro model to validate the function of neo-splice sites. Interestingly, this cell line harbored 27 base pairs (Fig. 6b, black shading) of non-template insertion as part of the cryptic exon. We therefore tested the essentiality of the cryptic exon using CRISPR-Cas9 ($g_1$) to create a double-stranded DNA break in the non-template insertion sequence (Methods; see Supplementary Data 14 for guide RNA sequences). We then measured the editing efficiency, including the size and frequency of each indel using targeted amplicon next-generation sequencing (NGS) (see Supplementary Data 15 for primers) from day 3 through day 19 post editing. Because indels with lengths of $3n + 1$ and $3n + 2$ will cause a frameshift in this cryptic exon, we expect these indels to reduce in frequency over time if the cryptic exon is essential to the cancer cells. Indeed, we observed a significant (Student's t-test P value of 0.0002) decrease in NGS read abundance (~66% at day 3 to <1% by day 19) of putative lethal indels (defined as indels with length $3n + 1$ and $3n + 2$ using CRIS.py[35]), corresponding to 220-fold decrease (Fig. 6c and Supplementary Data 15 and 16). By contrast, putative non-lethal indels (defined as indels with a length of 3n) demonstrated a stable increase in NGS read abundance from ~33% NGS reads at day 3 to 99% NGS reads at day 19 (Fig. 6c). These data indicated that the cryptic exon is essential for the HAL-01 cells. Indeed, RNA sequencing of CRISPR-edited HAL-01 cells confirmed a lack of alternative splicing (Supplementary Data 11) in *TCF3* (when exon 16 is used) so that the host cancer cells are completely dependent on cryptic exons via neo-splicing, which is in clear contrast with the weak E14-4 alternative splicing in B-ALL with E15-4 version of *TCF3-HLF* (SJALL018389 in Supplementary Data 7 and 13).

The essentiality of the cryptic exon enables us to investigate the functional importance of neo-splice sites in this locus. We designed a guide RNA $g_2$ to target the splice donor (Fig. 6b). Because the induced (random) indel may or may not completely disrupt the splice donor, we predicted the binding affinity (using a position-specific weight matrix (PWM) approach, see Methods) of residual splice donor site if the "GT" of the splice donor still exists after the indel has disrupted its context (see "NGS analysis of edited cell pools" in Methods). We simultaneously measured the translation frame status by assuming such residual splice donor site can be used by the host cell and only candidate donor sites with in-frame translations are evaluated for residual fitness as reflected by the abundance of NGS reads from amplicon sequencing. To account for the fact that binding affinity is a continuous variable, we divided the predicted binding affinity scores into bins, and studied the change of NGS read abundance for these score bins over time (from day 3 to day 19 post editing). Interestingly, a strong association between NGS read abundance and predicted binding affinity is observed (Fig. 6e). For example, NGS reads from editing that resulted in residual donor site with binding affinity score between 2 and 3 demonstrated a quick decrease from ~15% at day 3 to nearly 0% at day 19. NGS reads with binding affinity scores between 3 and 4 decreased from 30% at day 3 to ~1% at day 19. By contrast, NGS read abundance remained at a stable 10–20% abundance when the predicted binding affinity score was 4–5. NGS reads with binding affinity scores between 5 and 6 increased from <20% at day 3 to >30% at day 19. Notably, when the predicted binding affinity score was >6, the NGS read abundance increased from ~5% at day 3 to ~50% at day 19, indicating a strong gain of fitness of host cells. Collectively, by using a binding affinity score threshold of 4, our donor editing resulted in ~60% putative lethal on-target editing rate that is comparable to that (65%) of coding exon targeting.

We next targeted the neo-splice acceptor "AG" using guide RNA $g_3$. The analytical procedure was similar to that of splice donor targeting. Interestingly, although a significant proportion (~60%) of the induced indels fall into the coding region (and demonstrated expected lethal effect; Supplementary Fig. 8 and Supplementary Data 15 and 16), 10–15% of induced indels resulted in a complete loss of splice acceptor AG and demonstrated a significant reduction in NGS read abundance to nearly 0% at day 19. Like the donor-targeting experiment, this data clearly indicated the essentiality of the neo-splice acceptor, which is also a therapeutic vulnerability for HAL-01 cell. Of note, targeting regions outside the cryptic exon and splice site regions had no impact on fitness (Fig. 6d and Supplementary Fig. 8b, c), further corroborating our observation that the cryptic exon and its neo-splice sites are a specific vulnerability to HAL-01.

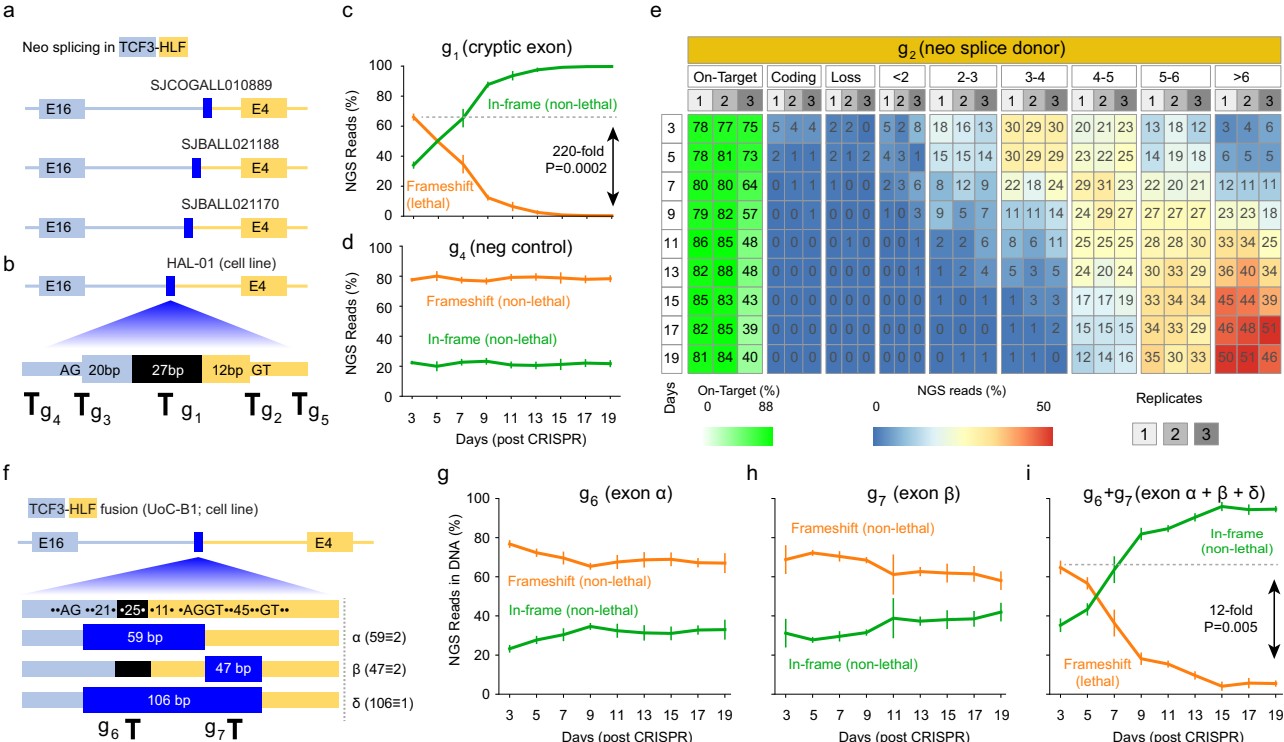

**Fig. 6 | Neo-splicing in oncogenic fusions and genome-editing-based therapeutic targeting. a** In our cohort, all samples with *TCF3-HLF* fusions harbor neo-splicing events due to incompatible exon frames between *TCF3* exon 16 and *HLF* exon 4 (Supplementary Fig. 7e). **b** We confirmed B-ALL cell line HAL-01 (DSMZ#: ACC 610) also harbors this pattern and designed guide RNAs to target the cryptic exon ($g_1$) and the neo-splice sites ($g_2$ and $g_3$) as well as negative control guides ($g_4$: 199 bps upstream of $g_3$; $g_5$: 52 bps downstream of $g_2$). Black shading indicates non-template insertion sequence (27 bp). **c** Cryptic exon is essential to HAL-01 by CRIPSR targeting using guide $g_1$, which leads to a 220-fold decrease of cells with lethal editing (two-sided *t*-test; *P* value = 0.0002; *n* = 3). Shown are percentages (*y*-axis) of putative lethal (orange) and non-lethal (green) editing measured using NGS reads as a function of time from day 3 to day 19 (*x*-axis) post editing for three replicates (error bars indicate standard deviation). Indels leading to frameshift of fusion transcripts are called lethal and in-frame indels are called non-lethal. **d** Negative control guide ($g_4$) that targets the upstream intronic region of the cryptic exon. Similar as panel **c**, percentage of putative lethal (frameshift; orange) and non-lethal (in-frame; green) editing measured by using NGS reads are shown from day 3 to day 19. **e** Neo-splice donor is essential to HAL-01 by CRISPR targeting using guide $g_2$. The induced indels that happened to fall into the coding region and lead to frameshift of *TCF3-HLF* are categorized into "Coding" group. Indels that directly disrupt the splice donor site are categorized into "Loss" group.

Most of the induced indels leave a residual GT that may still serve as a splice donor. The binding affinity of these residual donors is calculated using the position weight matrix (PWM) approach (see Methods) and the binding affinity scores are categorized into different bins (<2, 2–3, etc.). The percentage of NGS reads carrying induced indels are calculated for each bin from day 3 to day 19 post editing for three replicates. Also illustrated are on-target editing rate (green heatmap). **f** CRISPR targeting in the presence of alternative splicing. B-ALL cell line UoC-B1 also harbors *TCF3-HLF* fusion. However, in this fusion, we detected three neo-splicing patterns, α, β, and δ, where the first two splicing patterns can generate in-frame fusion proteins in the parental cells and δ cannot. We designed two guides ($g_6$ and $g_7$) to test the potential compensatory function of these isoforms. **g, h** Single guide editing led to marginal depletion of edited cells from day 3 to day 19 for putative lethal indels (orange) that can disrupt corresponding transcripts. **i** Double guide editing. Indels with putative lethal effect (orange) demonstrated a quick decrease (12-fold; two-sided *t*-test; *P* value = 0.005; *n* = 3) in abundance from day 3 to day 19, while indels with putative non-lethal effect demonstrated an increasing abundance. Data value and error bar at each time point represent mean of putative indels (orange for lethal; green for non-lethal as control) and standard deviation from three replicates in panels **c** and **d** and panels **g–i**. Source data are provided accordingly as sheets **a**, **b** and **c–e** in Source Data file.

## CRISPR targeting in the presence of alternative splicing

Although the above HAL-01 data indicate the feasibility of targeting the neo-splice sites as well as the cryptic exon of oncogenic fusions, the potential effect of alternative splicing cannot be studied due to the lack of natural alternative splicing in *TCF3-HLF* in HAL-01 (Supplementary Data 13). For this purpose, we used another *TCF3-HLF*-positive B-ALL cell line UoC-B1[36], which harbors a DNA breakpoint more upstream to intron 3 of *HLF* than that in HAL-01, so that there are more splice site options for UoC-B1. In this line, parental UoC-B1 cells can theoretically generate three splicing isoforms by utilizing the two candidate splicing acceptors AG and two candidate splicing donors GT (Fig. 6f). By using published RNA sequencing data available from NCBI SRA (accession number: SRR8816031) for UoC-B1 line, we confirmed all three possible splicing isoforms in parental cells: α (67% reads), β (12.5% reads), and δ (20.5% reads). Although isoforms α and β can help the UoC-B1 cells to resolve the translation frame problem between

*TCF3* exon 16 and *HLF* exon 4, isoform δ cannot. Therefore, we predicted that targeting isoforms α or β alone may not be effective due to compensatory splicing among them. To test this hypothesis, we designed one guide ($g_6$) to target isoform α and another guide ($g_7$) to target isoform β. As it turned out, $g_6$ and $g_7$ alone lead to a negligible reduction of putative "lethal" on-target editing that disrupts α and β, respectively (Fig. 6g, h and Supplementary Data 15 and 17). These data confirm the compensatory role of α and β exons when perturbed alone.

We next hypothesized that multiplexed editing that simultaneously disrupts all possible isoforms might lead to synthetic lethality. For this, we analyzed the theoretical possibilities (Supplementary Fig. 9c) of CRISPR targeting by using two guides $g_6 + g_7$. By categorizing the effect of induced indels into two states (being in-frame (I) or being out-of-frame (O)) for each of the three isoforms α, β, and δ, we predicted that only reads that lead to "O" state for all three isoforms

can result in a lethal effect—which comprises 37.5% (=3/8) of all on-target editing. Indeed, the putative lethal editing demonstrated a sharp decrease of NGS read abundance from ~37% at day 3 to nearly 0% at day 19 (Fig. 6i and Supplementary Data 17). By contrast, the putative non-lethal editing (with at least one of α, β, and δ in-frame) remained stable in NGS read abundance from day 3 to day 19. Because double guides theoretically can lead to indels at either a cut site or a single large deletion, we also studied the NGS reads of these two categories. Indeed, nearly 50% of lethal editing are large deletions, and both large deletions and double focal indels have a comparable decrease in NGS read abundance (Supplementary Fig. 9d). These data clearly demon-strated the functionally compensatory nature of alternative splicing in *TCF3-HLF* in UoC-B1 that posed a significant challenge in gene targeting using only single guide approach.

Together, our experiments indicated that neo-splicing in onco-genic fusions is essential for host cancer cells and offers therapeutic vulnerability. Our data also indicated the challenge in CRISPR targeting of oncogenic dependencies, where infidelity in splicing can generate isoforms that may compensate the editing assault. These data call for powerful computational approaches to accurately predict outcomes of CRISPR editing to enable rationale design of CRISPR guides to maximize the desired effect.

## Discussion

The data from 5190 childhood cancers have revealed several insights into the etiology of oncogenic fusions. First, gene length, splicing, translation frame, and protein domains together influence the for-mation of oncogenic fusions. Although some of these diverse factors have been sporadically reported in literature, our study provided a comprehensive picture of their contribution. For example, our math-ematical modeling provided insights into the association between protein domain, positive selection, and patient outcome in *CBFB-MYH11* AML patients (Fig. 5b). The fact that DNA breakpoints are dis-tributed uniformly in involved introns indicated that there are unlikely specific molecular mechanisms driving the formation of the majority of intronic rearrangements, although the local clustering of DNA breakpoints in fusions such as *TCF3-PBX1* indicate a specific molecular mechanism. Such mechanism, once elucidated, may lead to insights into cancer prevention. Notably, we only discovered *KMT2A* fusions with prevalence well predicted by gene length. For fusions that lack such association, our data strongly indicate alternative mechanisms, such as differential selection pressure due to protein domain usage in *CBFB-MYH11* (Fig. 5c–e) or the clustered DNA breakpoints in *TCF3* fusions (Fig. 2l), are at play. It is warranted to validate our findings for specific fusions with larger sample sizes. Second, certain oncogenic fusions appear to have promoter-hijacking-like feature, where the C′ gene is not or lowly expressed in host tissues. Such genes can be ideal drug targets with minimal "on-target, off-tumor" toxicity to normal cells of the corresponding lineage of cancer cells. Third, we discovered extensive alternative splicing in a subset of oncogenic fusions and demonstrated that such alternative splicing is likely an inherent property of the host gene rather than a tumor-specific phenomenon. Clearly, alternatively spliced exons and corresponding protein domains should be avoided in drug targeting due to their potentially dispensable nature. Our data also highlight the need to study the clinical implication of alternative splicing in oncogenic fusions. For example, in sample SJBALL030563_D1 (with fusion *ETV6-RUNX1*), we discovered 11 reads supporting junction E5-3 and 7 reads for junction E5-E4 (Supplementary Data 7). Although we determined the pre-dominant isoform (E5-3) by read supports, it would be interesting to study if one or both of the chimeric proteins are translated and the tumorigenesis function of these chimeric proteins. Fourth, we detec-ted clear selection bias in intronic versioning and established a strong link of such selection bias with clinical outcomes for *CBFB-MYH11*. These data indicated a differential oncogenicity of corresponding

fusions due to the inclusion/exclusion of protein domains. However, no other fusion pairs can be studied in this work due to the lack of publicly available clinical outcome data for corresponding cases for additional candidates with selection bias (*ETV6-RUNX1*, *KIAA1549-BRAF*, *NUT214-ABL1* and *BCR-ABL1*). It would be interesting to test these candidate pairs in future dedicated studies. Finally, we discovered a subset of tumors (*n* = 78) to harbor neo-splicing, neo-translation, and chimeric exon features. By using in vitro cell line model, we demon-strate the therapeutic potential of CRISPR-Cas9-based genome editing on neo-splicing events. We also demonstrate the complex nature of therapeutically targeting such events when alternative splicing is involved. These data call for innovative computational approaches that can accurately predict outcomes of genome editing to enable rationale design of guide RNAs. Although we did not show genome editing data on fusions in neo-translational and chimeric exon cate-gories, we envision such events are also ideal targets for genome editing because the guide RNAs can be highly specific to cancer cells and therefore minimize the "on-target, off-tumor" toxicity. For exam-ple, the neo-donor and acceptor sites (in *TCF3* locus) in HAL-01 were never found to be utilized from >500,000 RNAseq reads from 9,925 GTEx non-cancer samples. Therefore, we expect the corresponding guide RNAs to have no impact on normal cells. Furthermore, we elu-cidated cell line models that can serve as powerful in vitro systems to develop and optimize genome editing tools as well as to further understanding cis-regulatory elements for gene splicing.

This work also highlights an analytical framework to investigate oncogenic fusions. By focusing on the large variability in the patient prevalence of oncogenic fusions, we established the near-uniform nature of DNA breakpoints generating oncogenic fusions, which pre-dicts that gene length (or length of involved introns) would be asso-ciated with the patient prevalence of corresponding fusion gene pairs. However, only *KMT2A* fusions followed this prediction. To resolve this discrepancy, our systematic study discovered additional molecular factors (such as local DNA properties in *TCF3* and protein domain in *CBFB-MYH11*) that together shape patient prevalence. Interestingly, although a majority of fusions utilize existing exon/intron structure by connecting them to form chimeric proteins, cryptic exons are created in ~3% fusion-positive patients. The requirement of proper splicing sites and translation frames in these fusions may explain their rela-tively low prevalence. We expect this generic workflow to shed more biological and clinical insights when more samples are available for rare fusions.

Our study has a few limitations. First, although our cohort size of 5,190 patients has exceeded the two previous pan-cancer studies on childhood malignancies (1699 patients in Ma et al.[7] and 961 patients in Gröbner et al.[6]), it still does not solve the small sample size problem for fusions with low prevalence. We envision such limitations can be alleviated by dedicated study designs to enroll patients with specific disease subtypes including the NCI-COG Pediatric MATCH program. Such studies may reveal additional insights into fusions that are not well studied in this work due to limited sample sizes. Second, although our current targeting strategy is highly effective, it only covers 78/2005 = 3.9% of fusion-positive patients. However, we provide proof-of-concept data for *TCF3-HLF* fusion, a rare ALL subtype associated with a high rate of treatment failure[37] that clearly would benefit from such targeted therapy. It would be interesting to test if this strategy is broadly applicable to all oncogenic fusions with neo-splicing or chi-meric exon feature. Furthermore, additional studies are needed to develop innovative targeting strategies for patients without such "easy-to-design" targeting strategies. Third, although it is commonly assumed that oncogenic fusions can be a therapeutic target due to oncogenic addiction, it remains an untested hypothesis for many recently discovered oncogenic fusions. We believe it is of critical importance to verify the addictive role of every putative oncogenic fusion to accurately define a "targetable list" for the research

community. Fourth, we did not study the category of promoter- and enhancer-hijacking fusions. With the above argument of "absence of expression in corresponding normal cell", such fusions are optimal therapeutic targets. However, defining this category of fusions is currently a subject of intensive research[38]. For example, in 2021 alone, our studies have revealed two more genes in the promoter/enhancer hijacking category: *BCL11B* in MPAL/AML[39] and *MECOM* in therapy-related myeloid neoplasms[40]. It would be interesting to study the etiology of these fusions when the gene list is more comprehensive by using DNA and RNA sequencing data.

## Methods

### Patient cohort and RNAseq data

This study has been approved by St. Jude Children's Research Hospital Institutional Review Board (IRB). All patient data are from public resources detailed in Supplementary Data 2. Briefly, transcriptome sequencing (RNAseq) data from 5190 patients were collected from following public resources: (1) St. Jude cloud[41] (https://www.stjude.cloud/) that included the St. Jude/Washington University Pediatric Cancer Genome Project[42] cohort (PCGP; *n* = 777), the St. Jude Genomes for Kids study[43] (G4K; *n* = 253) and the St. Jude Real-time Clinical Genomics initiative (RTCG; *n* = 1027; as of January 1, 2021); (2) a collection of transcriptome study (*n* = 313) of childhood AML[40,44–50]; (3) a genomics study of relapsed childhood ALL[8] (*n* = 80; https://ngdc.cncb.ac.cn/gsa-human/browse/HRA000119); (4) NCI's Therapeutically Applicable Research to Generate Effective Treatments cohort[7] (TARGET; *n* = 759; https://ocg.cancer.gov/programs/target/data-matrix); (5) AML transcriptome data from Children's Oncology Group (*n* = 1088; https://portal.gdc.cancer.gov; study "TARGET-AML"); (6) Children's Brain Tumor Network (*n* = 725; CBTN; https://cbtn.org/; as of June 1, 2021) downloaded from Kids First data portal (https://portal.kidsfirstdrc.org); and (7) childhood rhabdomyosarcoma (RHB; *n* = 84; study identifier phs000720) and Ewing sarcoma (EWS; *n* = 84; study identifier phs000768 and phs000804) downloaded from dbGaP (https://www.ncbi.nlm.nih.gov/gap/). In addition, 9525 transcriptome datasets from GTEx project (https://gtexportal.org/home/; version 7) were used as normal controls in relevant analyses (Supplementary Data 10). All these data are under restricted access and an IRB approval was obtained to request access to these data. Sex and gender were not considered in the study design and samples were analyzed based on the availability in publicly available resources.

### Fusion detection

Oncogenic fusions were detected by using state-of-the-art methods reported to have superior performance[18,20], including Cicero (https://doi.org/10.1186/s13059-020-02043-x, v0.3.0)[18], Arriba (https://doi.org/10.1101/gr.257246.119, v1.2.0)[16], STAR-fusion (https://www.biorxiv.org/content/10.1101/120295v1, v1.6.0)[17], and FusionCatcher (https://www.biorxiv.org/content/early/2014/11/19/011650, v1.33)[19]. Cicero was run on bam files aligned with STAR v2.5.3a while Arriba, STAR-Fusion, and FusionCatcher were run on fastq files.

Because of the large number (5,781,630) of predicted fusions from these four methods, manual inspection is impractical, if not impossible. We therefore developed a workflow (see detailed design principles and algorithmic descriptions in Supplementary Notes) using majority voting (a prediction is considered to have *k* votes if it is detected by *k* methods) to enable the effective and efficient detection of oncogenic fusions from 5190 patients. This workflow has eight critical considerations. First, mutual exclusivity among oncogenic fusions. Using 63 well-known oncogenic fusions (Supplementary Note 10 and Supplementary Data 19), we determined 1743 patients to harbor ≥1 of these fusions detected by ≥2 detection methods. Interestingly, only 4 (0.23%) patient samples harbor ≥2 fusions, which indicates that each patient tumor typically harbors no more than 1 oncogenic fusions. Second, harmonizing coordinate differences

among methods. By comparing predictions between methods, we determined that different methods can have ~10 base pairs differences in their predicted fusion coordinates (Supplementary Note 11a). Third, multiple calls of the same oncogenic fusion pairs. As demonstrated in this work, intronic versioning is observed in many fusion pairs. Clearly, each intronic version corresponds to a unique prediction. Depending on the signal strength (number of supporting reads), some methods may "miss" a low abundance version (thus a low vote count), although other high abundance versions are commonly detected. By focusing on the high abundance versions, we determined that >93% of oncogenic fusion versions have 3+ votes. Fourth, although there are >5.7 million predictions, only 0.3% (16,348) of predictions have 3+ votes. These data render manual inspection possible by focusing on high-vote predictions, with a false-negative rate <7%. Fifth, with the mutual exclusivity rule, we can establish a blacklist by collecting all predictions from 1743 fusion-positive patients that do not match any known fusions. Such a blacklist allows us to further filter common artifacts such as readthrough[16], with a negligible false-negative rate of 7%. Sixth, manual review of the remaining 7769 predictions with 3+ votes. As detailed in Supplementary Notes, priority was given to in-frame predictions with *n* ≥ 2 recurrences in our cohort, or with literature support, such as previously published childhood cancer studies (pan-cancer analyses[6,7] and a number of disease-focused analyses including ependymoma[51], EWS[52,53], RHB[54], low-grade glioma[55], high-grade glioma[56], T-cell acute lymphoblastic lymphoma[57], AML[58], as well as a recent clinical genomics report[43] and other literature[6–8,28,31,37,39,40,44,45,50–56,58–60]). Quality indications from these methods were also considered. This review takes about 5 h for an experienced scientist. Seventh, upon manual review, we used the comprehensive list of oncogenic fusions to run a second round of systematic detection. This step allows us to minimize the impact of a hard threshold of "≥3 votes". Upon this comprehensive identification, we re-analyzed and further confirmed the mutual exclusivity rule: only 7 (0.35%) of 2005 chimeric fusion-positive patients have 2+ oncogenic fusions. Eighth, determining the functional orientation of oncogenic fusions. When a patient tumor has balanced translocations, there might be reciprocal fusions, such as *ETV6-RUNX1* and *RUNX1-ETV6*. By collecting the number of patient samples with each orientation detected, we determined that all 52 fusions detected in 4+ patients have the clinically recognized orientation supported with higher frequency than the other orientation. Collectively, we detected 272 unique oncogenic fusion gene pairs that can generate chimeric proteins (Supplementary Data 33). We also reported promoter-hijacking fusions for 12 genes (Supplementary Data 30) in this cohort.

In some literature, the terminology of "fusion" is used interchangeably with "structural alteration/rearrangement". Because gene fusion is a much earlier concept following the discovery of *BCR-ABL1* fusion in Philadelphia chromosome-positive leukemia, here we propose to use "fusion" exclusively in transcriptome (or gene expression) setting and to use "structural alteration/arrangement" in genome (or DNA) setting, although their biological meaning could be identical and can be discerned from context. Clearly, not all gene fusions may carry biological functions like *BCR-ABL1* in Philadelphia chromosome leukemia. For example, chromosomal rearrangements that lead to the inactivation of tumor suppressor genes (e.g., *CDKN2A*, *RB1*, *ATRX*) can be detected as out-of-frame fusion transcripts but are beyond the scope of this work. In this work, "oncogenic fusion" indicates in-frame fusions that produce oncogenic proteins like *BCR-ABL1*.

The fusion detection was performed in an institutional (St. Jude) high-performance computing cluster with 227 nodes, 474 CPUs (12,128 Cores) and 194TB of RAM and 20 petabytes of useable parallel file system storage, connected through 40 Gigabit Ethernet links. Although Cicero, Arriba and STAR-Fusion take less than 2 days to finish for most of the samples, we noticed that the earlier FusionCatcher

version (v1.10) runs slowly for most samples. Therefore, we generated minibams by including known driver gene regions of childhood cancers (Supplementary Data 18 and 19) to validate the findings from Cicero, Arriba, and STAR-Fusion. Most of the jobs on these minibams can be finished in around 10 days. Due to the limited project storage space allocated for the laboratory, and the large storage space needed due to raw fastq and bam files as well as intermediate files, we analyzed the data in batches. The total download, re-download, run, re-run and analysis time of this cohort (using up to 500 jobs at any given time) took us about 1 year. Interestingly, a recent FusionCatcher version (v1.33) runs much faster for full bams (typically ~1 day) and we were able to finish the re-run of all four fusion detectors on all full bams/fastqs of our full cohort in less than 3 months for revision. All raw output from the four fusion detectors were deposited in Zenodo (https://doi.org/10.5281/zenodo.7510612)[61].

The commands used for the fusion detectors are as follows: (1) Arriba (v1.2.0): arriba -o good_fusions.tsv -O discarded_fusions.tsv -a genome_assembly -g annotation_gtf -b black_list -T -P r1.fq r2.fq; (2) Cicero (v0.3.0): Cicero.sh -b bamfile -g genomeVersion -r cicero_r-efdir -j juncitons.tab; (3) STAR-Fusion (v1.6.0): STAR-Fusion --left_fq r1.fq --right_fq r2.fq --genome_lib_dir genome_lib; and (4) Fusion-Catcher (v1.33): fusioncatcher -d DATADIR -i r1.fq,f2.fq. Here r1.fq and r2.fq indicate the fastq files of read1 and read2, respectively.

### Neo-Versioner

An in-house python script ("Neo-Versioner") was developed to determine the status of intronic versioning (Fig. 1a and Supplementary Fig. 1). First, for each gene pair (e.g., *CBFB-MYH11*), we checked the translation frame for all possible exon-exon combinations of the two involved genes to build a database of in-frame exon-exon combinations (Fig. 5c, both gray and red connections). Next, for each in-frame exon combination, we constructed a junction contig (60 nucleotides) using 30 nucleotides of involved exons from the N' gene and the C' gene, respectively. A database of 20-mers was then constructed from these contig sequences to facilitate the efficient extraction of RNAseq reads containing one of such 20-mers. Each candidate read was compared to all junction contigs. A junction contig is determined to be supported once if it is a substring of a read. To account for partial matching, we allowed a read to contain a matching of as few as 10 nucleotides from either N' or C' side, provided that the other side of the junction contig is fully matched to the read (Supplementary Fig. 1a). The above parameters assumed an error rate of <1% in short read Illumina sequencing that is justified by recent error profile studies on NGS[62,63].

A significant challenge in using public fusion detection methods is the harmonization of their output; in particular, there is no straightforward way to directly harmonize their read counts. Therefore, read counts from Neo-Versioner were used throughout the manuscript for all applicable analyses, such as the extent of alternative splicing.

### Calculating pseudo-binding affinity for splice sites

The binding affinity of the candidate splice site to splicing machinery is calculated using the well-established PWM method[64]. We downloaded human genes from UCSC Genome Browser (http://hgdownload.soe.ucsc.edu/goldenPath/hg19/database/ncbiRefSeqCurated.txt.gz), extracted protein-coding genes (RefSeq ID starts with "NM_") and their exon boundaries and constructed PWMs using 209,192 donors and 205,329 acceptors from these known protein-coding genes (Supplementary Data 20). For donor, 3 base pairs of 5' to the GT and 10 base pairs of 3' to the GT are used, totaling a 15 base pair motif[65]. For acceptor, 18 base pairs of 5' to the AG and 3 base pairs of 3' to the AG are used, totaling a 23 base pair motif[65]. The motifs are denoted as $M_{ij}$ where $i$ can be either of A, C, G or T and $j = 1,...,K$ where $K$ is 15 for donor and 23 for acceptor. $M_{ij}$ represents the observed occurrences of known splice sites at position $j$ for nucleotide $i$. Denote the candidate

DNA sequence as $S_j$, $j = 1,...,K$, it can be scored by the PWM using a log-likelihood ratio score method:

$$LLR = \sum_j \sum_i \log\left(\frac{M_{ij}}{B_i}\right) \times I(i,S_j) \qquad (1)$$

were $B_i$ is the genome-wide background frequency of nucleotides A, C, G and T. Here, we set $B_i = 0.3$ when $i$ is A or T and $B_i = 0.2$ when $i$ is C or G to account for the A/T richness in the human genome. $I(i, S_j)$ is an indicator function that takes value of 1 when $S_j = i$ and 0 otherwise.

To ensure the quality of our constructed motifs, we scored all splice sites of known human genes and confirmed most of the splice sites received positive scores (>80% donors have score >4; >80% acceptors have score >4.3). As a negative control, we extracted 1.12 million potential donor (GT) sites and 1.76 million potential acceptor (AG) sites that do not belong to known human genes from forward strand of chr19 (one of the shortest chromosomes to save computation time) and scored them. As it turned out, >90% of such false donors have score <4 and >90% of such false acceptors have score <4.3, validating the power of our PWM method in discriminating real splice sites from non-real sites (see Supplementary Fig. 10 for the score distribution of true and false splice donors and acceptors).

### Neo-splicer

As illustrated in Fig. 1a, the cancer cells may create neo-splice sites to allow the production of functional chimeric proteins if the natural splice sites were disrupted by rearrangements. However, as illustrated in Fig. 6, a neo-splice site may not necessarily lead to in-frame translation because multiple splice sites may be available for the cancer cells that will survive if there is one viable (in-frame) splicing isoform. To search for neo-splice sites that can result in in-frame translation, we developed an in-house script ("Neo-Splicer"; see Supplementary Fig. 1) in this work. To detect neo-splice sites, this method requires DNA breakpoints between the two genes along with the non-template insertion sequence (when applicable), because sometimes the neo-splice sites can be embedded in the non-template insertion sequences[66] flanking the rearrangement boundary. Given the ubiquitous nature of candidate splice sites (AG and GT; 1 in every 16 nucleotides expected by random chance), we detected putative splice sites by using PWM method as described in the section "Calculating pseudo-binding affinity for splice sites". Second, given the DNA breakpoints (40.5% (n = 813/2005) chance of detection in RNAseq data; Fig. 2j–l and Supplementary Data 8) of an oncogenic fusion, we enumerated all AG and GT dinucleotides between intact exons of involved genes, generated hypothetical exons, and checked corresponding translation frames. RNAseq reads were then compared with the above predictions to determine the neo-splice sites and corresponding isoforms used by the cancer cells (Supplementary Fig. 1b). We note that the cryptic neo-spliced exon and/or non-template insertion sequences can remain poorly mapped by standard mapper (here STAR v2.5.3a), such as the non-template sequences in *TCF3-HLF* fusion in HAL-01 and UoC-B1 (Fig. 6b, f and Supplementary Fig. 7d), especially when the neo-splice sites are within non-template sequences (fusion *C11orf96-MAML2* in patient SJEPD031093 in Supplementary Fig. 7a). These mapping challenges are resolved by Neo-Splicer using DNA contigs.

### Expression patterns of oncogenic fusions

Although N' genes, which contribute enhancer and promoter regions for the oncogenic fusions, are expected to be constitutively expressed in the host lineage of the corresponding tumor, the C' gene may not always be expressed (Fig. 3). We propose an EDS to measure such expression patterns. For this, we calculated the expression level of the (fusion portion) C' and N' genes as median sequencing depth ($E_c$ and $E_N$) in the corresponding RNAseq sample. The EDS score is then

defined as EDS = $E_C/E_N$ for each sample. For an index oncogenic fusion, the samples can be categorized into (1) positive for the index fusion; (2) positive for other fusions; and (3) negative for fusions. Discrepancy in EDS scores between category (1) and categories (2) and (3) would indicate potential differential expression of the C' gene. Because here we are interested in the relative expression ratio between C' gene and N' gene, the global RNAseq normalization procedures[67,68] are not needed, which renders our EDS analysis highly efficient. Such scores are similarly calculated in non-cancer samples from GTEx cohort (Supplementary Fig. 5). We calculated FPKM values as previously described[7] to study the expression level of C' genes in the promoter-hijacking-like category.

### Selection bias in fusion versioning
As illustrated in Fig. 5, the alternative exon (and therefore protein domain) usage due to intronic versioning can potentially lead to differential oncogenicity and therefore selection bias (although we expect equal oncogenicity for the different DNA breakpoints from the same intronic version where the same fusion protein is produced; Supplementary Note 13). For this, we calculated the observed patient prevalence ($N_i$) for all intronic versions of a given fusion. Next, we normalized the patient prevalence by the length of the corresponding intron ($L_i$). The RSB score is then defined as $RSB_{ij} = (N_i \times L_j)/(N_j \times L_i)$, where $i$ and $j$ indicate the two possible introns in evaluation, in either the N' gene or the C' gene. To evaluate statistical significance, we performed $\chi^2$ tests by comparing observed patient prevalence against expected patient prevalence under the null hypothesis that involved introns carrying equal selection pressure. Only intronic versioning (i.e., natural in-frame fusions) was considered in this analysis.

### Detecting DNA breakpoints from RNAseq
Given the known oncogenic fusion in an RNAseq sample, we attempted to detect DNA breakpoints as follows. Because the DNA breakpoints must happen downstream to the RNA breakpoint of N' gene and upstream to the RNA breakpoint of the C' gene, we enumerated the first and the last k-mer ($k = 20$) in each read and checked if these two k-mers can be mapped to the eligible regions of N' gene and C' gene, respectively. The match is extended as much as possible from the k-mer of N' gene to the 3' end of the read to determine the DNA breakpoint of N' gene, and then from the k-mer of C' gene as much as possible to the 5' end of the read (but not beyond the DNA breakpoint of N' gene) to determine the DNA breakpoint of C' gene. To benchmark the specificity of this method (as a sanity check), we collected whole-genome sequencing–based DNA breakpoints from published papers on applicable samples (Supplementary Data 8) as a gold standard. As it turned out, >90.6% RNA-based DNA breakpoint detections are within 5 base pairs of DNA-based detections, validating the high specificity of our method. However, we note that not all DNA breakpoints can be determined from RNAseq data (which results in false negatives as compared with DNA sequencing), due to varying RNAseq protocols (poly-T based mRNA-seq or total RNAseq that contains pre-spliced transcripts) or sampling fluctuations.

### Test of uniformity of DNA breakpoints in intron regions
The uniformity of distribution of DNA breakpoints in intron regions was assessed by using a two-dimensional extension of the Kolmogorov–Smirnov test that has found application in astronomy to study the clustering of planets[69] in a pseudo two-dimensional space.

### Splicing dominance score
To measure potential alternative splicing, we introduced an SDS (Fig. 4a). First, we calculated the read support ($X_i$) for all fusion versions $i$ (with a minimum of 3 supporting reads) detected in a sample (aligned with STAR v2.5.3a) with the index fusion. Denote the read support for the canonical splicing (exon closest to the DNA breakpoint) as $X_I$. Next, we defined the dominance score as SDS = $X_I/\sum X_i$. A higher SDS score would indicate a lack of alternative splicing.

To study whether alternative splicing in oncogenic fusions is an inherent property of host genes, we defined SDS scores for involved genes in samples without the index fusion (wildtype) in a similar fashion. First, we defined the fusion-target exon of N' gene as the most downstream exon among these fusion versions, and the fusion-target exon of C' gene as the most upstream exon among these fusion versions (Supplementary Fig. 6a). Then, we calculated the read supports ($Y_i$) for splicing that span the target exon of N' gene (or C' gene). Denote the read support for the canonical splicing (connecting exons closest to the DNA breakpoint) as $Y_I$. The dominance score is then defined as SDS = $Y_I/\sum Y_i$.

Samples of a matched cancer type were categorized into (1) positive for the index fusion; (2) positive for another fusion; and (3) negative for all fusions, to study the extent of alternative fusions and whether such property is found in corresponding wildtype genes in samples without the index fusion. This method is also applied to GTEx samples as normal control.

### Calculation of hazard ratio
Disease-free survival (DFS) was defined as the time since EOI1 to relapse, death, or last follow-up. Cox proportional hazard regression models were employed to estimate hazard ratios for univariable analysis of DFS in the context of fusion breakpoint and other established prognostic covariates. A $P$ value <0.05 was considered statistically significant.

### Cell lines
Cell line HAL-01 was purchased from DSMZ (catalog #ACC 610). Cell line UoC-B1 was a courtesy from William Evans and Jun J. Yang. For both cell lines, STR profiling, whole-genome and transcriptome sequencing were performed to confirm identification and DNA and RNA breakpoints. Both cell lines are negative for mycoplasma contamination using MycoAlert Mycoplasma Detection Kit (Lonza).

### Cell fitness/dependency assay
One million HAL-01 or UoC-B1 cells were transiently transfected with precomplexed ribonuclear proteins consisting of 150 pmol of chemically modified sgRNA (Synthego) and 50 pmol of SpCas9 protein (St. Jude Protein Production Core) via nucleofection (Lonza, 4D-Nucleofector™ X-unit) using solution P3 and program CA-137 in a small (20 ul) cuvette according to the manufacturer's recommended protocol. For deletion samples, a bridging ssODN donor (3 ug; IDT) was also included in the nucleofection. A portion of cells (~10% of well) was collected at the indicated day post nucleofection. Genomic DNA was harvested, amplified, and sequenced via deep sequencing using a two-step library generation method. Briefly, gene-specific primers with partial Illumina adapters were used to amplify the region of interest in step 1. Gene-specific amplicons were then indexed via nested PCR using primers that bind to the partial Illumina adapters in step 2. Primers and genome editing reagents are listed in Supplementary Data 14.

### NGS analysis of edited cell pools
Upon CRISPR editing, we performed targeted amplicon sequencing using Illumina MiSeq on the edited cell pools to quantify the induced indels across multiple observation timepoints. For exon targeting (g₁) in cell line HAL-01, the induced indels will lead to frameshift if the length is not 3n (3, 6, 9, etc.), which can be analyzed by CRIS.py that measures length of target amplicon reads[35]. However, the length measurement is not suitable for analyzing splice site disruption in the edited cell pools. We therefore developed dedicated in-house methods (cri-splicer) to analyze such data as below. This method incorporated our recent analysis[62] of error profiles in NGS data by filtering poor quality reads with >5% bases with Phred score <20.

For guide $g_2$ (targeting neo-donor) in cell line HAL-01, we expect the neo-donor site GT to be disrupted by the induced indel. Because it is possible that the indel can happen slightly off the desired GT dinucleotide, our algorithm is designed to account for the following three possible editing scenarios: (1) the indel falls into the 5′ coding exon of desired target GT, so that it is still exon targeting per se (Fig. 6e, "Coding" category); (2) the indel falls into the 3′ side of desired target GT so that it that can affect the binding affinity between splicing machinery and the donor motif; and (3) the indel directly disrupts the GT dinucleotide (Fig. 6e, "Loss" category). For scenario (1), the unedited donor motif (from GT to 10 bp downstream) must be intact, and the indel must locate to the 5′ end of this motif. We next checked the translation frame of the resultant mRNA by assuming this donor is utilized. To account for the potential decrease of binding affinity, we also calculated the PWM score (see "Calculating pseudo-binding affinity for splice sites" in Methods) for this donor motif from the mutant read. For scenario (2), the exonic sequence must be intact, and the indel must locate to the 3′ end of the exon. We also calculated the PWM score as described above. For scenario (3), neither the exonic boundary nor the unedited acceptor motif can be found in the mutant sequence. We scanned the mutant sequence for all GT dinucleotides, calculated their PWM scores and determined their translation frame status by assuming they can induce splicing. This procedure was similarly applied for guide $g_3$ (targeting neo-acceptor) in cell line HAL-01, except we used dinucleotide AG for acceptor and the PWM is trained from known acceptors of all human genes.

For negative controls $g_4$ and $g_5$ (that target upstream and downstream intronic regions in HAL-1), we counted the percentage of edited reads for 3n and non-3n indels as a negative control for guide $g_1$, even if no functional consequences are expected (Supplementary Fig. 8b, c).

A similar program was written for UoC-B1 editing, although in this cell line we simultaneously considered the reading frames of all three possible exons: α, β, and δ (Supplementary Fig. 9c).

We investigated the length of CRISPR-induced indels in our data. To account for potential sequencing errors[62], we limited our analysis to indels with more than 3 read support. In HAL-01, >95% of induced indels at day 3 have lengths between −9 and 9 (Supplementary Fig. 11a, b), which is consistent with the previous report[70]. Therefore, in this work, we defined "On-Target" editing as indels within 10 base pairs from the designed target position, so that indels with single read support can also be included. Notably, >80% of the induced indels are insertions (Supplementary Fig. 11a, b). For UoC-B1 double targeting, we studied both double focal indels and single large indels. As it turned out, double focal indels (Supplementary Fig. 11c) demonstrated a similar pattern to that of single guide targeting (Supplementary Fig. 11b). On the other hand, the single large deletions demonstrated lengths centered around −55 (Supplementary Fig. 11c), as expected by our guide RNA design (Fig. 6f).

### Indel calling

Considering the double focal indel and large deletion in UoC-B1 experiment, a dedicated script (indel-caller) was developed to call indels. Briefly, we used the unedited UoC-B1 DNA as a reference sequence for this locus for BLAST program[71]. We then blast each NGS read against this reference. Indels were then called by maximizing the perfect match from 5′ end and then from 3′ end. All remaining DNA segments were called "reference allele" and "mutant allele", respectively, for the indel, along with the position. Because this procedure will generate the same representation for both the large deletions and double focal indels, we performed a post-processing step to further call double focal indels. For this procedure, because the splice site between exons α and β is of critical importance, we checked the presence of a k-mer (CCCAG | GTATT, where the vertical line is the splice site between exons α and β) in the mutant allele of each called indel. An indel containing this k-mer was then split to call focal indels by

focusing on the DNA segments to the 5′ or to the 3′ of this k-mer, respectively. This method was used to generate the double focal indel length distribution in Supplementary Fig. 11 and the concordance analysis between double focal indels and single large deletions in Supplementary Fig. 9d.

### Reporting summary

Further information on research design is available in the Nature Portfolio Reporting Summary linked to this article.

## Data availability

All genomics datasets used for this work are from public resources detailed in Supplementary Data 2, including CBTN [https://portal.kidsfirstdrc.org], FredHutch [https://portal.gdc.cancer.gov], TARGET [https://ocg.cancer.gov/programs/target/data-matrix], G4K, PCGP, RTGC [https://platform.stjude.cloud/data/cohorts], Rhabdomyosarcoma (PMID:24436047) phs000720.v4.p1, Ewing sarcoma (PMID:25010205) phs000768.v2.p1 and phs000804.v1.p1, AML (PMID:27798625) [https://platform.stjude.cloud/data/cohorts, SJC-DS-1013], MDS (PMID:29146900) EGAD00001003782, AML (PMID:30760869) [https://ocg.cancer.gov/programs/target/data-matrix], Erythroleukemia (PMID:30926971) EGAS00001002537, AML (PMID:31350825) [https://platform.stjude.cloud/data/cohorts], B-ALL (PMID:31697823) HRA000119, tMN (PMID:33579957) EGAD00001006674, AML (PMID:34778799) EGAD00001006444, AML (PMID:35176137) EGAD00001008407. The total RNAseq data and WGS data generated in this study for HAL-01 and UoC-B1 cell lines are deposited at ENA under accession number PRJEB55308. The time series data post CRISPR editing for cell line HAL-01 and UoC-B1 are deposited in Zenodo [https://doi.org/10.5281/zenodo.7510612][61]. All data generated in this study are provided in Source Data file. All input data and plot scripts for figures can also be found in Zenodo [https://doi.org/10.5281/zenodo.7510612][61]. All these data are under restricted access and an Institutional Review Board (IRB) approval was obtained to request access to these data. Source Data are provided with this paper.

## Code availability

All in-house scripts for this work are deposited in Zenodo (https://doi.org/10.5281/zenodo.7510612)[61].

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

## Acknowledgements

X.M. thanks Makeda Porter and Michael Edmonson for the scientific editing of the manuscript. X.M. thanks Karol Szlachta for assistance in two-dimensional extension of the Kolmogorov–Smirnov test. We acknowledge the effort by St. Jude HPC facility (Chad Burdyshaw, Michael Brewer, and Mi Zhou) in enabling FusionCatcher v1.33 for our samples. Research reported in this publication was supported in part by the National Cancer Institute of the National Institutes of Health under Award Number R01CA273326 (to X.M.). S.M. is partly supported by Leukemia and Lymphoma Society (7025-21) and St. Baldrick Foundation (SAT-21-064-01-SBF-ACS). This work was also supported in part by the Fund for Innovation in Cancer Informatics (www.the-ici-fund.org, to X.M. and J.M.K.), Cancer Center Support Grant P30CA021765 (Developmental Fund to J.M.K. and X.M.) from the National Institutes of Health, and the American Lebanese Syrian Associated Charities (ALSAC). The content is solely the responsibility of the authors and does not necessarily represent the official views of the National Institutes of Health or other funding agencies.

## Author contributions

X.M. conceived the research. X.M. and Y.L. developed the mathematical models. Y.L. implemented the computational approaches. S.M.M., J.K. and R.B. performed CRISPR assay. L.D. monitored the time course post CRISPR editing. Y.L., Q.T., P.K., J.S., R.E.R, and T.I.S. downloaded tumor transcriptome data, performed mapping, and fusion detection. S.W. and G.W. downloaded GTEx data. Y.L. J.K., and R.B. analyzed NGS amplicon data post CRISPR editing. Y.L., J.K., R.B., L.D., Q.T., P.K., J.L.S., R.E.R., B.J.H., Y.-C.W., T.A., L.T., H.L.M., T.I.S., J.M., M.W., G.S., T.W., R.J.A., A.M.G., D.A.W., S.W., G.W., J.J.Y., W.E., M.L., J.E., J.Z., J.M.K., S.M., P.A.B., S.M.M., and X.M. analyzed data. X.M. drafted the manuscript. All authors read and approved the final manuscript.

## Competing interests

X.M. is a named inventor on a pending patent application based in part on the research disclosed in this manuscript. The authors declare that they have no competing interests.

## Additional information

[1]Department of Computational Biology, St. Jude Children's Research Hospital, Memphis, TN, USA. [2]Department of Cell and Molecular Biology and Center for Advanced Genome Editing, St. Jude Children's Research Hospital, Memphis, TN, USA. [3]Clinical Research Division, Fred Hutchinson Cancer Research Center, Seattle, WA, USA. [4]Department of Pediatrics and Helen Diller Family Comprehensive Cancer Center, University of California San Francisco, San Francisco, CA, USA. [5]Children's Oncology Group, Monrovia, CA, USA. [6]Department of Preventive Medicine, University of Southern California, Los Angeles, CA, USA.

[7]Department of Biostatistics and Bioinformatics, Moffitt Cancer Center, Tampa, FL, USA. [8]Department of Pathology, St. Jude Children's Research Hospital, Memphis, TN, USA. [9]Department of Pharmacy and Pharmaceutical Sciences, St. Jude Children's Research Hospital, Memphis, TN, USA. [10]Center for Applied Bioinformatics, St. Jude Children's Research Hospital, Memphis, TN, USA. [11]Ben Towne Center for Childhood Cancer Research, Seattle Children's Research Institute and the Department of Pediatrics, Seattle Children's Hospital, University of Washington, Seattle, WA, USA. [12]Bristol Myers Squibb, Princeton, NJ, USA. [13]Present address: Hopp Children's Cancer Center Heidelberg (KiTZ), Heidelberg, Germany. [14]Present address: Division of Pediatric Neurooncology, German Cancer Research Center (DKFZ), Heidelberg, Germany. ✉e-mail: Jeffery.Klco@stjude.org; smeshinc@fredhutch.org; Patrick.Brown@bms.com; Shondra.Miller@stjude.org; Xiaotu.Ma@stjude.org

