## [Peer review file · Nature Communications]

Editorial Note: The author responses in this file contain figures generated with the tool published in:
Zhou, X. et al. Exploration of Coding and Non-coding Variants in Cancer Using GenomePaint. *Cancer Cell*,
39(1):83-95.e4 (2021). <https://www.doi.org/10.1016/j.ccell.2020.12.011>

REVIEWER COMMENTS

Reviewer #1 (Remarks to the Author): Expert in paediatric cancer genomics

This manuscript is focused on a meta-data analysis of numerous publicly available pediatric cancer transcriptome sequencing datasets derived from an impressive number of 5,190 patients and aiming at elucidation of the etiology of oncogenic fusions. While the manuscript is well written (although not really telling a novel story) and the data science approaches seem to be solid and state-of-the-art, the results do not add any substantial novel insight into the biology or mechanistics of oncogenic fusions, but rather confirm previous knowledge in a large patient number.

1. The conclusion, "Together, these data indicate that random chance (or gene/intron length), and less frequently, local DNA properties can influence the formation of oncogenic fusions" is an interpretation that can easily be made based on the existing knowledge/data in the literature.
2. The diagnostic and clinical relevance of the data seems to be overestimated and it is unclear, how the necessity for the CRISPR-Cas9 experiments performed is extracted from the data.
3. Previously well-known phenomena (i.e. oncogenic fusion is leading to overexpression of an involved gene via the promotor of the other gene) are given a new name („promotor-hijacking“) without being a novel finding.
4. The observations regarding splicing are also expected. Why should there be no splicing of fusions genes as for any other gene? This is no surprise...
5. Sometimes single observations in the data (i.e. KMT2A fusion genes are associated with gene length/prevalence) are generalized without systematic validation
6. Some descriptions are misleading, i.e.: „Surprisingly, we obtained a statistically significant linear association ($R^2=0.82$; $P=0.002$; Fig. 2i) between gene length and patient prevalence“. This is not surprising, but rather to be expected.

Reviewer #2 (Remarks to the Author): Expert in gene fusion detection and computational cancer genomics

The manuscript "Etiology of oncogenic fusions in 5,190 childhood cancers and its clinical and therapeutic implication" by Liu et al. describes an effort to explore the characteristics of oncogenic fusions in childhood cancers including primarily leukemia and brain cancer. The authors examine fusion breakpoint positions, the expression of fusion partner genes in fusion-containing, fusion-lacking, and normal samples, and features contributing to alternatively spliced fusions, all with a focus on relevance to potential clinical applications in personalized medicine. In a spectacular demonstration of using CRISPR genome editing to target a particularly vulnerable type of oncogenic fusion involving "neo splicing" in cell lines, the authors clearly showed the essential nature of the neo splicing for cancer cell line viability, and addressed the complexity of targeting such fusions when more complex alternative splicing exists. The manuscript is very well written, quite comprehensive, and creative - I particularly like the 'circuit' style fusion figures. I appreciate that the authors have packaged up code along with data and made it all readily available. My recommendations and critiques are as follows:

Major

- The fusions studied in this work were restricted to oncogenic fusions relevant to childhood cancers. To identify the fusions, the authors leveraged four different fusion transcript prediction methods. From these complete lists of fusion predictions (not currently provided in supplemental materials, as far as I could tell), the subset relevant to childhood cancers was extracted and reported (provided in Supp. Table 3). The authors provided Supp. Figure 1 to show the fusions defined as childhood oncogenic fusions, which totals to just over 60 examples. But, there are over 260 fusion types (unique gene pairings) for the reported fusions in ST3. It isn't entirely clear how the full set of predicted fusions derived from the four prediction tools was distilled to this set of 260 that were analyzed. More transparency is needed here, ideally providing the full set of predictions in supplementary materials and defining the reasons for which the subset analyzed here were selected.

- One of the criteria for defining fusions of interest is that fusions produce an in-frame protein. Such in-frame fusion proteins represent a subset of all fusion transcripts relevant to cancer, and presumably childhood cancers. If the authors were to drop this requirement, are there other fusion transcripts (perhaps recurrent fusions) that might appear to be relevant to childhood leukemias among the initially unfiltered set?

- The TCF3--HLF fusion story is quite interesting. There is actually much known about TCF3--HLF published in the literature, and even though the paper is already quite long, it would be useful for the authors to add a few statements and reference relevant earlier work on this fusion. While TCF3--HLF as found here apparently requires newly found neo-splicing events to generate an in-frame protein, it is curious to me and wasn't clear from this manuscript that other previous studies on this fusion have found natural in-frame versions. I was very curious about this fusion as studied in this work and so explored the data myself with the publicly available RNA-seq data for the cell line UoC-B1 as leveraged here. I found evidence of direct fusions between TCF3 and HLF that would result in the frameshifted

protein but not requiring any neo-splicing events, which STAR-Fusion should have reported. This additional alternative splicing, while presumably not oncogenic, could be worth commenting on, particularly in respect to the relative expression levels of it as compared to the neo-spliced version. Similarly, I explored the HAL-01 cell line myself to directly explore the TCF3--HLF fusion in that sample. I didn't see in the manuscript what exact set of reads (ie. SRA accession) was used here, and so I downloaded one of many that are publicly available for this cell line (SRA: ERR3504860). Using STAR-Fusion and FusionInspector, I found evidence for more conventional in-frame fusions that did not require neosplicing, which surprised me, as these would presumably be oncogenic and not disrupted via genome editing. I encourage the authors to examine this closely and offer an explanation. I've included FusionInspector results for HAL-01 as reviewer supplemental info. The html file can be opened in a web browser, and the tsv file contains the predicted fusions.

- The methods section would benefit from additional details such that the process of identifying candidate fusions as performed in this study is more transparent and reproducible. This relates to my comment above regarding how fusions were initially filtered, but also extends to how the downstream analysis of neo-splicing events were discovered. Was neo-splicer somehow run on every fusion candidate that met some predefined criteria? Was finding or not finding neo-spliced exons one of the criteria that was initially used to define the set of filtered in-frame fusions, such that all initially defined frameshifted fusions were examined for candidate neo-splicing events that would restore the reading frame?

- The reported finding of total fusion gene length (summing the fusion gene partner lengths) being linearly correlated with patient prevalence is tenuous, and as the authors show, it does not hold for brain or solid tumors as examined here. The finding of a clear linear correlation between prevalence of KMT2A fusion partners and patient abundance according to the length of the gene fusion partner is quite interesting. However, the authors show just this one striking example. It would be worthwhile to explore all such fusions having multiple fusion partners to see if the linear correlation holds more generally.

- The proposed dependence of fusion prevalence on gene length becomes the basis of the authors relative selection bias score (RSB), used to examine the relative selection of exon-versioned fusions where evidence of alternative splicing exists. Normalizing the number of fusion breakpoints by intron length does make intuitive sense even if the direct evidence for doing so appears weak or limiting as shown. However, I do wonder if the RSB statistic is adding unnecessary complexity. Are there examples where the RSB statistic provides for more accurate estimates as opposed to using a simpler fusion version abundance ratio (lacking length normalization)? It would also be useful to know if fusion versions were limited to only the in-frame candidates primarily studied in this work, as considering out-of-frame alternative splicing would presumably impact the relative rankings of RSB statistics among fusion candidates; for example, theoretically there could be an out-of-frame fusion isoform fusion that dominates all in-frame fusions according to patient prevalence. We could make a similar argument for

the splicing dominance statistic with regard to considering out-of-frame fusion versions that exist but may not have been considered during this study. Clarification here is warranted.

- In the section on the "Landscape of childhood oncogenic fusions", it states "To test this hypothesis, we first validated the high accuracy (>91% RNA-based detections are within 5 base pairs of DNA-based detections; see Method; Supplementary Table 8; Extended Data Fig. 2a) of detecting DNA breakpoints by using RNA-seq data." In my experience, fusion transcripts as detected by RNA-seq data tend to have breakpoints at splice junctions, which can be quite distant from the point of genomic rearrangement that would be detected using DNA-seq data. This is consistent with Figure 5a, showing RNA breakpoints at splice sites, and DNA breakpoints scattered throughout the intron. So it isn't clear to me how the authors identified genomic breakpoints using RNA data in such cases. This requires more explanation in both the methods and the main text.

Minor:

- I appreciate that the authors put all code and related data on Zenodo. For tools developed here such as NeoSplicer, making the code, documentation, and test data available on GitHub would be far more appropriate. Also, more details in the very short manifest file describing the various scripts and data would also be helpful. I truly applaud the authors for their current efforts in making all these materials available, and I simply suggest making them slightly more accessible as companion materials. Finally, what's in the SV_Detection folder? It's not mentioned in the paper as far as I could tell.

- I expect there will be much interest in NeoSplicing as reported in this manuscript. If the authors can provide additional bam or GTF and fasta files for these fusions that would facilitate visualization using a tool such as IGV, that would make it easier for more in depth exploration. With bam formatting, it would be possible to examine the template-only insertion sequences in the viewer.

- In Figure 5a, I recommend labeling the N and C fusion genes. At first glance, the figure looked to me like the bottom part was supposed to represent a zoomed-in view of the upper part. It could be just me and the way I tend to make those types of 'zoomed in' figures, though. The N- and C- gene labels would have helped me personally.

- Line 67: In the statement "We demonstrate a novel mathematical model that can detect differential selection pressure, which is validated in CFBF-MYH11 positive acute myeloid leukemia (AML) to confer superior prognostic value on event free survival than other well-known clinical features." the 'novel mathematical model' seems a bit reaching. This is using the RSB metric, a length-normalized version of a relative patient abundance for introns - where intron length normalization seems intuitive but potentially unnecessary. In addition to clarifying whether out-of-frame fusion versions are considered, I'd suggest incorporating each of the fusion version comparisons tested in Fig. 5e, or as a companion supplemental figure so we can see how all the other combinations fare.

- For the HAL-01 cell line RNA-seq, please indicate the SRA accession for the data used.

- Including some details about how the computes were run, presumably in the cloud at St. Jude, would be helpful in the methods. Any time thousands of computationally intensive jobs are needed, it's useful for others to know about how such a feat was accomplished.

- For the SDS statistic, how were the RNA-seq alignments derived? Using STAR or another aligner?

- Please include commands executed for the various fusion finders in the supplement. If workflows were used (ie. WDL, CWL, nextflow, etc.), providing the workflows would be useful too.

- Fig 6d - the g4 negative control is shown, but not the g5 negative control. Is there a reason why g5 was excluded here?

- In ST2, it would be useful to include the counts of number of samples for each data set, as listed in the methods section. Also, it would be useful to have another supplementary table that indicates the sample name and corresponding data set association, which can be cross-referenced in the other data tables. Other metadata about the individual samples would be useful too including the number of reads in each. This way, one can determine from the fusion read counts what proportion of the total reads they represent.

- the MS-Excel ST3 table had a filter automatically invoked, showing only a small subset of the total data. Best to undo that filter before submission.

- in Figure 6, the guide numbers for cell line UoC-B1 reuse values g1 and g2. I'd recommend starting where the count left off with the HAL-01 guide numbering, using instead g6, g7 for clarity.

- The PWM scoring is reported to be in ranges of <2 to >6. It isn't immediately clear how such score numbers are being derived from the LLR equation. The LLR would usually give large negative values, where the least negative value reflects the highest probability. Please include more details on how the LLR converts to the score value.

- In extended figure 6, the row values don't always sum to 100 (usually 99), which I found curious.

- in Fig 6D, 'non-lethal' is used correctly to describe the negative control situation for in-frame and frameshift. In extended Figure 6, 'lethal' and 'non-lethal' are used for the negative control context. It might be better to use 'frameshift' and 'in-frame' instead, since both are non-lethal here as well.

- I'm curious why the RSB metric was chosen for pairwise comparisons as opposed to using an SDS like metric that accounts for the proportion by summing the total in the denominator. Would an SDS-like RSB metric outperform the RSB itself for ranking hazard ratios or be simpler for identifying best therapeutic candidates?

- In ST3 (and ideally, the original unfiltered starting fusion set), please include the programs that initially predicted them along with fusion evidence read support. For ST3, the NeoVersioner recomputed read support for predicted fusions would be useful for comparing to the originally defined read support quantities from the individual tools.

- in NeoSplicer, it isn't obvious how the non-template fusion sequences were defined as shown. The algorithm for doing this should be described as part of the NeoSplicer methods section.

- it is curious that many of the analysis tools used in this work were hand-crafted by the authors. For example, instead of using commonly leveraged alignment utilities, the authors used their own kmer-based read mapping schemes. Such analyses, like identifying DNA breakpoints from RNA-seq data in intronic regions, being dependent on custom utilities (instead of using trusted alignment methods like bowtie, STAR, bwa, etc) does make one question the accuracy of the methods being used. Showing a situation such as uniform insertions across intronic regions using custom methods, where we know intronic regions are repeat-rich and more difficult to accurately align reads to than exonic regions, could result from inaccurate / random placement of reads. I'm not suggesting that the authors methods are inaccurate, but rather suggesting that the authors would be best to demonstrate that their methods can accurately perform their function, such as by simulating data and ensuring accurate results using simulated data. This would be most important for NeoSplicer and the method inferring DNA breakpoints from RNA-seq data.

- Line 81: typo, should read "(C-terminus)"

- Lines 177, 517: RNA-seq does not provide 'absolute expression level'. RNA-seq provides relative expression levels (normalized by total read sequencing depth). Perhaps just change to 'expression level'.

- Line 247: typos, should be 'division with denominator three equals zero'

- Line 402: typo, rational

- Line 588: typo, pools

Reviewer #3 (Remarks to the Author): Expert in paediatric cancer genomics and gene fusions

This study by Liu et al. provided information on the etiology of 1,989 fusions identified in 5,190 childhood cancers. This report studied a wide spectrum of cancer, including leukemia, CNS, and solid tumors. The manuscript was well-written and provided exquisite detail. The authors identified 5 different molecular mechanisms by which oncogenic fusions may be generated, highlighting the large diversity in types of fusions. While different types of fusions have been described extensively in prior literature, Liu et al provided additional biological context into not only general classes of fusions but specific fusions as well. This work does demonstrate some clinical and therapeutic implications, which I would be interested in seeing highlighted more as childhood cancers tend to be underrepresented with regard to targeted therapies and clinical trials.

Major points:

- I do not see a caption for Supplementary Figure 1
- Extended Data Figure 3 (Lines 37-39) "On the other hand, FOXO1 appears to have higher expression than PAX3/PAX7 in normal brain tissues, consistent with the observation in brain tumors (Fig. 3b)." Why

is normal brain tissue from GTEx being used as the comparator for FOXO1 expression when this fusion is found predominately in muscle/soft tissue?

- For those fusions with alternative splicing, did the authors look at other transcript isoforms to see if fusions derived from alternative splicing may be due to multiple transcript isoforms for the same gene being expressed in the tissue type under study?
- The section on alternative splicing in fusion generation was intriguing since through clinical testing we have noted a number of fusions, including those involving RUNX1 and ABL1 (to name a couple), more frequently present with multiple fusion breakpoints in the same analyzed tumor sample. We have wondered whether alternative splicing has a role here and clinically which breakpoint is more informative to the providers or clinical outcomes. Did you find any samples in your data where different breakpoints representing the same fusion were seen in the same patient sample and if so, how did you determine which one was the predominant isoform?
- Given the title, I was hoping for additional clinical and therapeutic implications with evidence provided from this study. The authors hypothesize that certain fusion etiologies may more appropriate for targeted therapies or CRISPR methodology but only provide data for a single fusion event and EFS. While, the CRISPR approach to targeting some of the fusions, particularly those in which there isn't a targeted therapy available is interesting and promising, it is likely still a long off from clinical use. Additionally, those fusions that were hypothesized to be good drug targets lack a targeted therapeutic. Given that there is RNA-seq for all of these data, it may be interesting to look at patterns of expression to see if there is overexpression or evidence for a pathway deregulation which may be targeted with currently available therapeutics (i.e. MEK inhibitors for KIAA1549-BRAF), although this may be beyond the scope of the manuscript. Can the authors provide any additional outcomes data for the recurrent fusions described in this manuscript?

Minor points:

- Figure 1b and line 284 - The "NM1-PATZ1" fusion should be listed as "MN1-PATZ1"
- Ensure that gene names or fusions that are being discussed as the DNA or RNA are italicized
- The tool FusionCatcher is written in as both FusionCatcher and Fusion-catcher. Please update to maintain consistency
- Methods, Neo-Splicer (Line 501) – "where" should be "were"
- Extended Data Figure 2 caption (Line 17) – "show" should be "shown"
- Extended Data Figure 2 caption (Line 23) – "RUNX1-RUN1T1" should be "RUNX1-RUNX1T1"
- Supplementary Table 3 – How was the number of supporting reads calculated? How did you account for potentially discrepant coordinates from the four fusion callers?
- Figure 3b – "CBFB-MY11" should read "CBFB-MYH11"

Point-by-point response to reviewer comments

Reviewer #1 (Remarks to the Author): Expert in paediatric cancer genomics

This manuscript is focused on a meta-data analysis of numerous publicly available pediatric cancer transcriptome sequencing datasets derived from an impressive number of 5,190 patients and aiming at elucidation of the etiology of oncogenic fusions. While the manuscript is well written (although not really telling a novel story) and the data science approaches seem to be solid and state-of-the-art, the results do not add any substantial novel insight into the biology or mechanistics of oncogenic fusions, but rather confirm previous knowledge in a large patient number.

[Response] We appreciate the general positive comment from reviewer #1. It appears that this reviewer's main concern is "lack of novel insights", for which we respectfully disagree.

Indeed, thanks to the many large scale cancer genomic profiling works, numerous oncogenic fusions have been defined for childhood cancers in past decade. It has been noted that the prevalence of these fusions varies widely, but **no systematic study on this variation** is found in literature. It is exactly this void that our study is trying to fill in. To stress this significance, we added in section "Landscape of childhood oncogenic fusions": "Although it has been noted that the prevalence of oncogenic fusions varies widely^{33,34}, no systematic studies on potential mechanisms are found in literature."

Further, we added in Discussion to highlight the conceptual advancement of our analytical framework: "This work also highlights a novel analytical framework to investigate oncogenic fusions. By focusing on the large variability in patient prevalence of oncogenic fusions, we first established the null hypothesis that gene length or random chance can be a significant predictor in certain fusions. In fusion where such null hypothesis does not hold, we were able to find interesting patterns that reflect interesting biology (local DNA properties in *TCF3*) and clinical implications (*CBFB-MYH11*). Although majority of fusions utilize existing exon/intron structure by connecting them to form chimeric proteins, novel cryptic exons are created in ~3% fusion positive patients. The requirement of proper splicing sites and translation frames in these cases may explain their relative low prevalence. We expect this generic workflow to shed more biological and clinical insights when more samples are available for rare fusions."

1. The conclusion, "Together, these data indicate that random chance (or gene/intron length), and less frequently, local DNA properties can influence the formation of oncogenic fusions" is an interpretation that can easily be made based on the existing knowledge/data in the literature.

[Response] To provide a better context, we have added the rationale to study the factors contributing prevalence of oncogenic fusions: "Although it has been noted that the prevalence of oncogenic fusions varies widely^{33,34}, no systematic studies on potential mechanisms are found in literature."

We next stressed that the association between gene length and patient prevalence was next used as a null hypothesis to detect additional biological/clinical insights: "By assuming the association between gene length and patient prevalence as our null hypothesis, our extended analysis to brain tumor and solid tumor did not yield statistical significance (**Extended Data Fig. 2f-g**). This data may either reflect the diverse subtypes and corresponding smaller cohort sizes among brain tumor and solid tumor or indicate additional factors influencing the etiology of oncogenic fusions, which will be addressed in following sections."

Further, we added in Discussion to highlight the conceptual advancement of our analytical framework: “This work also highlights a novel analytical framework to investigate oncogenic fusions. By focusing on the large variability in patient prevalence of oncogenic fusions, we first established the null hypothesis that gene length or random chance can be a significant predictor in certain fusions. In fusion where such null hypothesis does not hold, we were able to find interesting patterns that reflect interesting biology (local DNA properties in *TCF3*) and clinical implications (*CBFB-MYH11*). Although majority of fusions utilize existing exon/intron structure by connecting them to form chimeric proteins, novel cryptic exons are created in ~3% fusion positive patients. The requirement of proper splicing sites and translation frames in these cases may explain their relative low prevalence. We expect this generic workflow to shed more biological and clinical insights when more samples are available for rare fusions.”

2. The diagnostic and clinical relevance of the data seems to be overestimated and it is unclear, how the necessity for the CRISPR-Cas9 experiments performed is extracted from the data.

[Response] We regret that the reviewer did not see the diagnostic and clinical relevance of our study on one of the most important type of somatic alterations—oncogenic fusions, as stressed in the initial submission: “In these cases, subtype-defining oncogenic fusions (e.g., BCR-ABL1 in Philadelphia chromosome positive patients^{1,2}) typically persist through the lifetime of a tumor^{8,9} and can serve as stable biomarkers for curative outcomes. Moreover, successes in targeted inhibition of oncogenic fusions (e.g., imatinib for BCR-ABL1¹¹) has inspired the notion of “**oncogene addiction**”¹² that posits on the **therapeutic potential of targeting oncogenic fusions.**”

With our systematic investigation on the prevalence, it naturally leads to a logic on where the fusion happens between exons with incompatible translation frames, and therefore the TCF3-HLF findings with “spectacular demonstration” of novel vulnerabilities (Reviewer #2). This logic is found in below sentence of original submission: “Indeed, close examination indicated that exon 16 of *TCF3* and exon 4 of *HLF* have incompatible translation frames (**Extended Data Fig. 5e**). Therefore, the neo splice sites and corresponding cryptic exons are created by the host cancer cell to compensate the translation problem.” This is the critical data that leads to our CRISPR editing strategy.

We added a statement in section “CRISPR targeting of neo splicing” to stress the clinical relevance of studying TCF3-HLF: “B-cell acute lymphoblastic leukemia (B-ALL) patients with TCF3-HLF fusion are currently considered incurable⁴³.”

3. Previously well-known phenomena (i.e. oncogenic fusion is leading to overexpression of an involved gene via the promotor of the other gene) are given a new name (“promotor-hijacking”) without being a novel finding.

[Response] We apologize for this confusion. To clarify, we added “Because these fusions generate novel chimeric proteins, this group of promoter hijacking-like fusions are fundamentally different from conventional promoter hijacking fusions (e.g., IGH-CRLF2/EPOR/DUX4) where no chimeric proteins are involved.”

4. The observations regarding splicing are also expected. Why should there be no splicing of fusions genes as for any other gene? This is no surprise...

[Response] Logically, a fusion oncogene may either have alternative splicing or not, therefore “no surprise”. However, our study provided a formal characterization on the exact list of fusions subject to

alternative splicing. We modified our text as “Together, our data indicated a clear involvement of alternative splicing in **certain, but not all**, oncogenic fusions, although such regulation is not specific to tumor and therefore is likely an intrinsic property of the host gene.”

Further, the lack of alternative splicing played a critical role in the success of our CRISPR-based targeting on TCF3-HLF fusion, which is now clarified as “**Indeed, RNA sequencing of CRISPR-edited HAL-01 cells confirmed lack of alternative splicing (Supplementary Table 13) in TCF3 (when exon 16 is used) so that the host cancer cells are completely dependent on cryptic exons via neo-splicing, which is in clear contrast with the weak E14-E4 alternative splicing in B-ALL with E15-E4 version of TCF3-HLF (SJALL018389 in Supplementary Table 7 and Supplementary Table 13).**”

As we stated, exons subject to alternative splicing may be dispensable for the tumor, and therefore should be avoided for drug targeting. This is an important concept we can draw for fusions with alternative splicing.

5. Sometimes single observations in the data (i.e. KMT2A fusion genes are associated with gene length/prevalence) are generalized without systematic validation.

[Response] Although we hope there is a single variable that can explain everything on the widely variable prevalence of oncogenic fusions, our study indicated it is unlikely. For example, CFBF-MYH11’s prevalence is significantly associated with protein domain usage. Likewise, in TCF3 fusions the local DNA property matters.

We updated the writing to stress that the association between gene length and patient prevalence was used as a null hypothesis to detect additional biological/clinical insights: “**By assuming the association between gene length and patient prevalence as our null hypothesis, our extended analysis to brain tumor and solid tumor did not yield statistical significance (Extended Data Fig. 2f-g).** This data may either reflect the diverse subtypes and corresponding smaller cohort sizes among brain tumor and solid tumor or indicate additional factors influencing the etiology of oncogenic fusions, **which will be addressed** in following sections.”

We also added in section “Discussion”: “**Notably, we only discovered KMT2A fusions with prevalence well predicted by gene length. For other fusions that lack the association between prevalence and gene length, our data strongly indicate alternative mechanisms, such as protein domain in CFBF-MYH11 (Fig. 5b) or the clustered DNA breakpoints in TCF3 fusions (Fig. 2i), are at play. It is warranted to validate our findings for specific fusions with larger sample sizes.**”

6. Some descriptions are misleading, i.e.: „Surprisingly, we obtained a statistically significant linear association (R-squared=0.82; P=0.002; Fig. 2i) between gene length and patient prevalence“. This is not surprising, but rather to be expected.

[Response] we updated “Surprisingly” to “**Indeed**”.

Reviewer #2 (Remarks to the Author): Expert in gene fusion detection and computational cancer genomics

The manuscript "Etiology of oncogenic fusions in 5,190 childhood cancers and its clinical and therapeutic implication" by Liu et al. describes an effort to explore the characteristics of oncogenic fusions in

childhood cancers including primarily leukemia and brain cancer. The authors examine fusion breakpoint positions, the expression of fusion partner genes in fusion-containing, fusion-lacking, and normal samples, and features contributing to alternatively spliced fusions, all with a focus on relevance to potential clinical applications in personalized medicine. In a spectacular demonstration of using CRISPR genome editing to target a particularly vulnerable type of oncogenic fusion involving "neo splicing" in cell lines, the authors clearly showed the essential nature of the neo splicing for cancer cell line viability, and addressed the complexity of targeting such fusions when more complex alternative splicing exists. The manuscript is very well written, quite comprehensive, and creative - I particularly like the 'circuit' style fusion figures. I appreciate that the authors have packaged up code along with data and made it all readily available. My recommendations and critiques are as follows:

[Response] We truly appreciate Reviewer #2's thorough review and insightful comments that significantly improved our presentation and the science.

Major

- The fusions studied in this work were restricted to oncogenic fusions relevant to childhood cancers. To identify the fusions, the authors leveraged four different fusion transcript prediction methods. From these complete lists of fusion predictions (not currently provided in supplemental materials, as far as I could tell), the subset relevant to childhood cancers was extracted and reported (provided in Supp. Table 3). The authors provided Supp. Figure 1 to show the fusions defined as childhood oncogenic fusions, which totals to just over 60 examples. But, there are over 260 fusion types (unique gene pairings) for the reported fusions in ST3. It isn't entirely clear how the full set of predicted fusions derived from the four prediction tools was distilled to this set of 260 that were analyzed. More transparency is needed here, ideally providing the full set of predictions in supplementary materials and defining the reasons for which the subset analyzed here were selected.

[Response] Thanks for the suggestion, we updated the "Fusion detection" section as follows: "Oncogenic fusions were detected by using state-of-the-art methods reported to have superior performance^{18,20}, including Cicero¹⁸, Arriba¹⁶, STAR-fusion¹⁷, and FusionCatcher¹⁹. Cicero was run on bam files aligned with STAR v2.5.3a while Arriba, STAR-Fusion, and FusionCatcher were run on fastq files. To ensure clinical relevance of detected oncogenic fusions, we curated a list of 315 driver genes (which included 142 and 77 significantly mutated genes from two non-overlapping pan-childhood cancer analyses^{6,7} and a number of disease-focused analyses including ependymoma²⁴, Ewing sarcoma^{23,26}, rhabdomyosarcoma²⁵, low grade glioma²², high grade glioma²⁷, T-cell acute lymphoblastic lymphoma⁶², acute myeloid leukemia³¹, as well as a recent clinical genomics report⁵⁵) for childhood cancers (Supplementary Table 18) reported in literature^{6-8,22-28,30,31,33,41,43,49,51,52,56,61} and used this gene list to extract candidate fusions predicted by these tools for manual review. This procedure resulted in 2,004 samples with 257 oncogenic fusion pairs (Supplementary Table 19). Upon manual review on remaining predictions, an additional 5 samples (Supplementary Table 19) were assigned as low confidence fusions in an ad hoc manner (for example, a rhabdomyosarcoma with *MEGF10-NUP210* fusion and a brain tumor sample with *NRCAM-PRKCB* fusion)."

Further, we added a caption to Supplementary Figure 1 (which was intended to be a print poster for oncologists) as follows to clarify how the 60 fusion pairs were obtained from the 260 list: "Recurrent (n≥3) oncogenic fusions identified in 5,019 childhood cancers. For each gene pair, coding exons (thick boxes) are colored white (frame 0), gray (frame 1), and black (frame 2). Intronic length is indicated by

numbers. Gray lines indicate theoretically in-frame fusions. Red lines indicate in-frame fusions observed in patients and its width indicate patient prevalence. RefSeq identifiers are indicated for each involving gene. Fusions involving neo-splicing or chimeric exons were not included (see paper).”

- One of the criteria for defining fusions of interest is that fusions produce an in-frame protein. Such in-frame fusion proteins represent a subset of all fusion transcripts relevant to cancer, and presumably childhood cancers. If the authors were to drop this requirement, are there other fusion transcripts (perhaps recurrent fusions) that might appear to be relevant to childhood leukemias among the initially unfiltered set?

[Response] Highly recurrent tumor suppressor genes (such as *CDKN2A* and *ATRX*) disrupted by chromosomal rearrangements can sometime be detected in RNA transcripts (but not always, such as in the case of whole 9p arm loss) and are beyond the scope of our study. We now clarified this in section “Fusion detection” as: “In some literature the terminology of “fusion” is used interchangeably with “structural alteration/rearrangement”. Because gene fusion is a much earlier concept following the discovery of *BCR-ABL1* fusion in Philadelphia chromosome positive leukemia, here we propose to use “fusion” exclusively in transcriptome (or gene expression) setting and to use “structural alteration/arrangement” in genome (or DNA) setting, although their biological meaning could be identical and can be discerned from context. Clearly not all gene fusions may carry biological functions like *BCR-ABL1* in Philadelphia chromosome leukemia. For example, chromosomal rearrangements that lead to inactivation of tumor suppressor genes (e.g., *CDKN2A*, *RB1*, *ATRX*) can be detected as out-of-frame fusion transcripts but are beyond the scope of this work. In this work, “oncogenic fusion” indicates in-frame fusions that produce oncogenic proteins like *BCR-ABL1*.”

- The *TCF3--HLF* fusion story is quite interesting. There is actually much known about *TCF3--HLF* published in the literature, and even though the paper is already quite long, it would be useful for the authors to add a few statements and reference relevant earlier work on this fusion.

[Response] We clarified the existing knowledge on *TCF3-HLF* as follows: “Consistent with previous report⁴², in our dataset we also discovered one case SJALL018389 (Supplementary Table 7) to harbor natural in-frame fusion between exon 15 of *TCF3* and exon 4 of *HLF*. In this sample, a weak alternative splicing between exon 14 of *TCF3* and exon 4 of *HLF* is observed. Analysis of published RNAseq^{43,44} data on *TCF3-HLF* cells with E15-E4 version under various drug treatments (JQ1, A-485)⁴⁴ further confirmed that this fusion version is subject to a weak alternative splicing regulation (**Supplementary Table 13**). Although it has been suggested that the cryptic exons function to make up the translation frame problem in *TCF3-HLF* by the cancer cells^{45,46}, there is no functional evidence available to date. In the next we sought to investigate the function of this cryptic exon and corresponding hypothetical neo splice sites through CRISPR-based genome editing.”

While *TCF3--HLF* as found here apparently requires newly found neo-splicing events to generate an in-frame protein, it is curious to me and wasn't clear from this manuscript that other previous studies on this fusion have found natural in-frame versions.

[Response] We appreciate this insightful comment. We have added following text to enrich our findings: “Consistent with previous report⁴², in our dataset we also discovered one case SJALL018389 (Supplementary Table 7) to harbor natural in-frame fusion between exon 15 of *TCF3* and exon 4 of *HLF*. In this sample, a weak alternative splicing between exon 14 of *TCF3* and exon 4 of *HLF* is observed. Analysis of published RNAseq^{43,44} data on *TCF3-HLF* cells with E15-E4 version under various drug

treatments (JQ1, A-485)⁴⁴ further confirmed that this fusion version is subject to a weak alternative splicing regulation (**Supplementary Table 13**). Although it has been suggested that the cryptic exons function to make up the translation frame problem in *TCF3-HLF* by the cancer cells^{45,46}, there is no functional evidence available to date. In the next we sought to investigate the function of this cryptic exon and corresponding hypothetical neo splice sites through CRISPR-based genome editing.”

I was very curious about this fusion as studied in this work and so explored the data myself with the publicly available RNA-seq data for the cell line UoC-B1 as leveraged here. I found evidence of direct fusions between TCF3 and HLF that would result in the frameshifted protein but not requiring any neo-splicing events, which STAR-Fusion should have reported. This additional alternative splicing, while presumably not oncogenic, could be worth commenting on, particularly in respect to the relative expression levels of it as compared to the neo-spliced version.

[Response] We decided to address this concern together with the next concern.

Similarly, I explored the HAL-01 cell line myself to directly explore the TCF3--HLF fusion in that sample. I didn't see in the manuscript what exact set of reads (ie. SRA accession) was used here, and so I downloaded one of many that are publicly available for this cell line (SRA: ERR3504860). Using STAR-Fusion and FusionInspector, I found evidence for more conventional in-frame fusions that did not require neosplicing, which surprised me, as these would presumably be oncogenic and not disrupted via genome editing. I encourage the authors to examine this closely and offer an explanation. I've included FusionInspector results for HAL-01 as reviewer supplemental info. The html file can be opened in a web browser, and the tsv file contains the predicted fusions.

[Response] We truly appreciate Reviewer #2's insight into this analysis. We decide to address the above two paragraph of comments together.

We initially were puzzled on how HAL-01 cell can generate natural in-frame fusions without using neo-splicing, because that is not compatible with our CRISPR data where effective killing was achieved--- which would be impossible if the cells were able to generate natural in-frame fusion.

We carefully re-analyzed the study design of PMID: 31735627 (and its earlier reference of PMID: 26214592) and figured out that the RNAseq data of SRA:ERR3504860 is from a patient 11a from PMID:26214592, where (Fig. 1b in that paper) it was clearly stated that 11a has a “Type II” (E15-E4, which is in-frame and utilizes natural exon structures) TCF3-HLF fusion. Because patient 11a's tumor is different from HAL-01, we added below in section “Neo splicing in oncogenic fusions”: “Consistent with previous report⁴², in our dataset we also discovered one case SJALL018389 (**Supplementary Table 7**) to harbor natural in-frame fusion between exon 15 of *TCF3* and exon 4 of *HLF*. In this sample, a weak alternative splicing between exon 14 of *TCF3* and exon 4 of *HLF* is observed. Analysis of published RNAseq^{43,44} data on *TCF3-HLF* cells with E15-E4 version under various drug treatments (JQ1, A-485)⁴⁴ further confirmed that this fusion version is subject to a weak alternative splicing regulation (**Supplementary Table 13**). Although it has been suggested that the cryptic exons function to make up the translation frame problem in *TCF3-HLF* by the cancer cells^{45,46}, there is no functional evidence available to date. In the next we sought to investigate the function of this cryptic exon and corresponding hypothetical neo splice sites through CRISPR-based genome editing.”

A screenshot of the new **Supplementary Table 13** is listed below:

Table S113. Fusion isoforms detection in 2 types of TCF3-HLF rearrangements. Listed are sample resource (Column A), DNA breakpoint types (Column B), sample (Column C), treatment (Column D), fusion types (Column E), breakpoints information in RNA (Column F), neo splice site (Column G-H), number of reads supporting wild type genes and fusion (Column I-K), number of reads supporting neo splice sites (Column L-M) and reference isoform used for wild type TCF3 and HLF (Column N-O).

sample resource	DNA breakpoints at TCF3	sample	Treatment	fusion_type	RNA breakpoint (hg19)	No. wildtype reads (N gene)	No. wildtype reads (C gene)	No. fusion reads	refseqA	refseqB
PDX_11a_m1660_DMSO	intron 15	ERR3504859	DMSO	In frame, conventional; E14-E14	TCF3.chr19.1619779..HLF.chr17.53398025.+	571	0	34	NM_003200	NM_002126
PDX_11a_m1660_DMSO	intron 15	ERR3504859	DMSO	In frame, conventional; E15-E14	TCF3.chr19.1619315..HLF.chr17.53398025.+	184	0	266	NM_003200	NM_002126
PDX_11a_m1660_JQ1	intron 15	ERR3504860	JQ1	In frame, conventional; E14-E14	TCF3.chr19.1619779..HLF.chr17.53398025.+	625	0	11	NM_003200	NM_002126
PDX_11a_m1660_JQ1	intron 15	ERR3504860	JQ1	In frame, conventional; E15-E14	TCF3.chr19.1619315..HLF.chr17.53398025.+	150	0	251	NM_003200	NM_002126
PDX_11a_m1660_A485	intron 15	ERR3504861	A-485	In frame, conventional; E14-E14	TCF3.chr19.1619779..HLF.chr17.53398025.+	328	0	15	NM_003200	NM_002126
PDX_11a_m1660_A485	intron 15	ERR3504861	A-485	In frame, conventional; E15-E14	TCF3.chr19.1619315..HLF.chr17.53398025.+	102	0	132	NM_003200	NM_002126
PDX_11a_m1699_DMSO	intron 15	ERR3504862	DMSO	In frame, conventional; E14-E14	TCF3.chr19.1619779..HLF.chr17.53398025.+	791	0	48	NM_003200	NM_002126
PDX_11a_m1699_DMSO	intron 15	ERR3504862	DMSO	In frame, conventional; E15-E14	TCF3.chr19.1619315..HLF.chr17.53398025.+	248	0	348	NM_003200	NM_002126
PDX_11a_m1699_JQ1	intron 15	ERR3504863	JQ1	In frame, conventional; E14-E14	TCF3.chr19.1619779..HLF.chr17.53398025.+	637	0	16	NM_003200	NM_002126
PDX_11a_m1699_JQ1	intron 15	ERR3504863	JQ1	In frame, conventional; E15-E14	TCF3.chr19.1619315..HLF.chr17.53398025.+	206	0	343	NM_003200	NM_002126
PDX_11a_m1699_A485	intron 15	ERR3504864	A-485	In frame, conventional; E14-E14	TCF3.chr19.1619779..HLF.chr17.53398025.+	371	0	25	NM_003200	NM_002126
PDX_11a_m1699_A485	intron 15	ERR3504864	A-485	In frame, conventional; E15-E14	TCF3.chr19.1619315..HLF.chr17.53398025.+	115	0	186	NM_003200	NM_002126
PDX_11a_m1700_DMSO	intron 15	ERR3504865	DMSO	In frame, conventional; E14-E14	TCF3.chr19.1619779..HLF.chr17.53398025.+	697	0	41	NM_003200	NM_002126
PDX_11a_m1700_DMSO	intron 15	ERR3504865	DMSO	In frame, conventional; E15-E14	TCF3.chr19.1619315..HLF.chr17.53398025.+	221	0	334	NM_003200	NM_002126
PDX_11a_m1700_JQ1	intron 15	ERR3504866	JQ1	In frame, conventional; E14-E14	TCF3.chr19.1619779..HLF.chr17.53398025.+	715	0	18	NM_003200	NM_002126
PDX_11a_m1700_JQ1	intron 15	ERR3504866	JQ1	In frame, conventional; E15-E14	TCF3.chr19.1619315..HLF.chr17.53398025.+	220	0	328	NM_003200	NM_002126
PDX_11a_m1700_A485	intron 15	ERR3504867	A-485	In frame, conventional; E14-E14	TCF3.chr19.1619779..HLF.chr17.53398025.+	348	0	28	NM_003200	NM_002126
PDX_11a_m1700_A485	intron 15	ERR3504867	A-485	In frame, conventional; E15-E14	TCF3.chr19.1619315..HLF.chr17.53398025.+	99	0	173	NM_003200	NM_002126

- The methods section would benefit from additional details such that the process of identifying candidate fusions as performed in this study is more transparent and reproducible. This relates to my comment above regarding how fusions were initially filtered, but also extends to how the downstream analysis of neo-splicing events were discovered. Was neo-splicer somehow run on every fusion candidate that met some predefined criteria? Was finding or not finding neo-spliced exons one of the criteria that was initially used to define the set of filtered in-frame fusions, such that all initially defined frameshifted fusions were examined for candidate neo-splicing events that would restore the reading frame?

[Response] This comment significantly improved our presentation. We added following details to make the study more reproducible in “Fusion detection” section: “Oncogenic fusions were detected by using state-of-the-art methods reported to have superior performance^{18,20}, including Cicero¹⁸, Arriba¹⁶, STAR-fusion¹⁷, and FusionCatcher¹⁹. Cicero was run on bam files aligned with STAR v2.5.3a while Arriba, STAR-fusion, and FusionCatcher were run on fastq files.

To ensure clinical relevance of detected oncogenic fusions, we curated a list of 315 driver genes (which included 142 and 77 significantly mutated genes from two non-overlapping pan-childhood cancer analyses^{6,7} and a number of disease-focused analyses including ependymoma²⁴, Ewing sarcoma^{23,26}, rhabdomyosarcoma²⁵, low grade glioma²², high grade glioma²⁷, T-cell acute lymphoblastic lymphoma⁶², acute myeloid leukemia³¹, as well as a recent clinical genomics report⁵⁵) for childhood cancers (**Supplementary Table 18**) reported in literature^{6-8,22-28,30,31,33,41,43,49,51,52,56,61} and used this gene list to extract candidate fusions predicted by these tools for manual review. This procedure resulted in 2,004 samples with 257 oncogenic fusion pairs (**Supplementary Table 19**). Upon manual review on remaining predictions, an additional 5 samples (**Supplementary Table 19**) were assigned as low confidence fusions in an ad hoc manner (for example, a rhabdomyosarcoma with *MEGF10-NUP210* fusion and a brain tumor sample with *NRCAM-PRKCB* fusion).

In some literature the terminology of “fusion” is used interchangeably with “structural alteration/rearrangement”. Because gene fusion is a much earlier concept following the discovery of *BCR-ABL1* fusion in Philadelphia chromosome positive leukemia, here we propose to use “fusion” exclusively in transcriptome (or gene expression) setting and to use “structural alteration/arrangement” in genome (or DNA) setting, although their biological meaning could be identical and can be discerned from context. Clearly not all gene fusions may carry biological functions like *BCR-ABL1* in Philadelphia chromosome leukemia. For example, chromosomal rearrangements that lead to inactivation of tumor suppressor genes (e.g., *CDKN2A*, *RB1*, *ATRX*) can be detected as out-of-frame fusion transcripts but are beyond the scope of this work. In this work, “oncogenic fusion” indicates in-frame fusions that produce oncogenic proteins like *BCR-ABL1*.

The fusion detection was performed in an institutional (St. Jude) high performance computing cluster with 227 nodes, 474 CPUs (12128 Cores) and 194TB of RAM and 20 petabytes of useable parallel file system storage, connected through 40 Gigabit Ethernet links. Although Cicero, Arriba and STAR-Fusion takes less than 2 days to finish for most of samples, we noticed that the earlier FusionCatcher version (v1.10) runs slowly for most samples. Therefore, we generated minibams by including known driver gene regions of childhood cancers (Supplementary Table 17) to validate the findings from Cicero, Arriba, and STAR-Fusion. Most of the jobs on these minibams can be finished around 10 days. Due to limited project storage space allocated for the laboratory, and the large storage space needed due to raw fastq and bam files as well as intermediate files, we analyzed the data in batches. The total download, re-download, run, re-run and analysis time of this cohort (using up to 500 jobs at any given time) took us about 1 year. Interestingly, a recent FusionCatcher version (v1.33) runs much faster for full bams (typically ~1 day) and we were able to finish the re-run of all four fusion detectors on all full bams/fastqs of our full cohort in less than 3 months for revision. All raw output from the four fusion detectors were deposited in zenodo (<https://doi.org/10.5281/zenodo.6421699>).

The commands used for the fusion detectors are as follows: (1) Arriba (v1.2.0): `arriba -o good_fusions.tsv -O discarded_fusions.tsv -a genome_assembly -g annotation_gtf -b black_list -T -P r1.fq r2.fq`; (2) Cicero (v0.3.0): `Cicero.sh -b bamfile -g genomeVersion -r cicero_refdir -j junctions.tab`; (3) STAR-Fusion (v1.6.0): `STAR-Fusion --left_fq r1.fq --right_fq r2.fq --genome_lib_dir genome_lib`; (4) FusionCatcher (v1.33): `fusioncatcher -d DATADIR -i r1.fq,r2.fq`. Here r1.fq and r2.fq indicates the fastq files of read1 and read2, respectively.”

Regarding the filtering, we added following details in “Model of fusion etiology and study design” section: “Candidate oncogenic fusions were detected using tools (Arriba¹⁶, STAR-Fusion¹⁷, CICERO¹⁸, and FusionCatcher¹⁹; **Methods**) reported to have superior performance^{18,20}. The detected candidate fusions were compared with previous genomics studies on childhood cancers^{6-8,15,21-33} to establish the comprehensive list of oncogenic fusions. To classify the detected oncogenic fusions into one of above four categories, we developed novel tools Neo-Versioner (to classify intronic versioning) and Neo-Splicer (to classify neo-splicing; see **Method** and **Extended Data Fig. 1**). If the fusion does not belong to either intronic versioning (i.e., no reads supporting natural exonic junctions between N' and C' genes) or neo splicing categories by our automated analysis, we manually review and classify the fusion into the categories of either chimeric exon or neo translational.”

- The reported finding of total fusion gene length (summing the fusion gene partner lengths) being linearly correlated with patient prevalence is tenuous, and as the authors show, it does not hold for brain or solid tumors as examined here. The finding of a clear linear correlation between prevalence of KMT2A fusion partners and patient abundance according to the length of the gene fusion partner is quite interesting. However, the authors show just this one striking example. It would be worthwhile to explore all such fusions having multiple fusion partners to see if the linear correlation holds more generally.

[Response] Although we hope there is a single variable that can explain everything on the widely variable prevalence of oncogenic fusions, our study indicated it is unlikely. For example, CFBF-MYH11's prevalence is significantly associated with protein domain usage. Likewise, in TCF3 fusions the local DNA property matters.

We updated the writing to stress that the association between gene length and patient prevalence was used as a null hypothesis to detect additional biological/clinical insights: “By assuming the association between gene length and patient prevalence as our null hypothesis, our extended analysis to brain tumor and solid tumor did not yield statistical significance (**Extended Data Fig. 2f-g**). This data may either reflect the diverse subtypes and corresponding smaller cohort sizes among brain tumor and solid tumor or indicate additional factors influencing the etiology of oncogenic fusions, which will be addressed in following sections.”

- The proposed dependence of fusion prevalence on gene length becomes the basis of the authors relative selection bias score (RSB), used to examine the relative selection of exon-versioned fusions where evidence of alternative splicing exists. Normalizing the number of fusion breakpoints by intron length does make intuitive sense even if the direct evidence for doing so appears weak or limiting as shown. However, I do wonder if the RSB statistic is adding unnecessary complexity. Are there examples where the RSB statistic provides for more accurate estimates as opposed to using a simpler fusion version abundance ratio (lacking length normalization)?

[Response] We added below statement to highlight the importance of length normalization in section “Selection bias in fusion versioning”: “To demonstrate the necessity of normalizing the patient prevalence using intronic length, we also generated data by excluding the length normalization. This resulted false positive findings (Supplementary Table 12), such as EWSR1-FLI1, CBFA2T3-GLIS2 and DEK-NUP214.”

It would also be useful to know if fusion versions were limited to only the in-frame candidates primarily studied in this work, as considering out-of-frame alternative splicing would presumably impact the relative rankings of RSB statistics among fusion candidates; for example, theoretically there could be an out-of-frame fusion isoform fusion that dominates all in-frame fusions according to patient prevalence. We could make a similar argument for the splicing dominance statistic with regard to considering out-of-frame fusion versions that exist but may not have been considered during this study. Clarification here is warranted.

[Response] We clarified this confusion in section “Selection bias in fusion versioning” as “Only intronic versioning (i.e., natural in-frame fusions) were considered in this analysis.”

- In the section on the "Landscape of childhood oncogenic fusions", it states "To test this hypothesis, we first validated the high accuracy (>91% RNA-based detections are within 5 base pairs of DNA-based detections; see Method; Supplementary Table 8; Extended Data Fig. 2a) of detecting DNA breakpoints by using RNA-seq data." In my experience, fusion transcripts as detected by RNA-seq data tend to have breakpoints at splice junctions, which can be quite distant from the point of genomic rearrangement that would be detected using DNA-seq data. This is consistent with Figure 5a, showing RNA breakpoints at splice sites, and DNA breakpoints scattered throughout the intron. So it isn't clear to me how the authors identified genomic breakpoints using RNA data in such cases. This requires more explanation in both the methods and the main text.

[Response] We added in section “Detecting DNA breakpoints from RNAseq” following statements to clarify: “To benchmark the specificity of this method, we collected whole-genome sequencing (WGS)-based DNA breakpoints from published papers on applicable samples (**Supplementary Table 8**) as gold standard. As it turned out, >91% RNA-based DNA breakpoint detections are within 5 base pairs of DNA-based detections, validating the high accuracy of our method. However, we note that not all DNA

breakpoints can be determined from RNAseq data, due to varying RNAseq protocols (poly-T based mRNA-seq or total RNAseq that contains pre-spliced transcripts) or sampling fluctuations.”

Minor:

- I appreciate that the authors put all code and related data on Zenodo. For tools developed here such as NeoSplicer, making the code, documentation, and test data available on GitHub would be far more appropriate. Also, more details in the very short manifest file describing the various scripts and data would also be helpful. I truly applaud the authors for their current efforts in making all these materials available, and I simply suggest making them slightly more accessible as companion materials. Finally, what's in the SV_Detection folder? It's not mentioned in the paper as far as I could tell.

[Response] We appreciate Reviewer #2's great interest in the codes. We have updated our manifest file to briefly introduce the folder structures and their basic purpose. For example, we now state that "SV_Detection" folder is "used to determine DNA breakpoints from RNAseq data".

Due the length of this manuscript, and the massive burden of executing the CRISPR experiments in additional 20 cell lines with neo-splicing and/or chimeric exon features, we are drafting a data portal manuscript that allow readers to interactively explore the findings in this work and run the analyses using our tools (which will be part of the well-received data portal of <https://pecan.stjude.cloud/proteinpaint/TP53>).

- I expect there will be much interest in NeoSplicing as reported in this manuscript. If the authors can provide additional bam or GTF and fasta files for these fusions that would facilitate visualization using a tool such as IGV, that would make it easier for more in-depth exploration. With bam formatting, it would be possible to examine the template-only insertion sequences in the viewer.

[Response] We have obtained approval from dbGaP for access of all datasets analyzed. Interested readers should be able to obtain access from dbGaP as well. On the other hand, we have generated and uploaded RNAseq data for HAL-01 and UoC-B1 cell lines, in response to Reviewer's critiques, in section "Data availability": "The total RNAseq data for HAL-01 and UoC-B1 cell lines are deposited at ENA (<https://www.ebi.ac.uk/ena/browser/view/PRJEB55308>)." We regret that we cannot make the patient data publicly available due to HIPAA regulation.

- In Figure 5a, I recommend labeling the N and C fusion genes. At first glance, the figure looked to me like the bottom part was supposed to represent a zoomed-in view of the upper part. It could be just me and the way I tend to make those types of 'zoomed in' figures, though. The N- and C- gene labels would have helped me personally.

[Response] Thanks, we now labeled the N and C genes in Figure 5a to avoid confusion.

- Line 67: In the statement "We demonstrate a novel mathematical model that can detect differential selection pressure, which is validated in CFBF-MYH11 positive acute myeloid leukemia (AML) to confer superior prognostic value on event free survival than other well-known clinical features." the 'novel mathematical model' seems a bit reaching. This is using the RSB metric, a length-normalized version of a relative patient abundance for introns - where intron length normalization seems intuitive but potentially unnecessary. In addition to clarifying whether out-of-frame fusion versions are considered,

I'd suggest incorporating each of the fusion version comparisons tested in Fig. 5e, or as a companion supplemental figure so we can see how all the other combinations fare.

[Response] We added below statement to highlight the importance of length normalization in section "Selection bias in fusion versioning": "To demonstrate the necessity of normalizing the patient prevalence using intronic length, we also generated data by excluding the length normalization. This resulted false positive findings (Supplementary Table 12), such as significant selection bias among versions would be detected for fusion EWSR1-FLI1, CBFA2T3-GLIS2 and DEK-NUP214."

We also clarified that only in-frame fusions are considered in section "Fusion detection" as: "In some literature the terminology of "fusion" is used interchangeably with "structural alteration/rearrangement". Because gene fusion is a much earlier concept following the discovery of *BCR-ABL1* fusion in Philadelphia chromosome positive leukemia, here we propose to use "fusion" exclusively in transcriptome (or gene expression) setting and to use "structural alteration/arrangement" in genome (or DNA) setting, although their biological meaning could be identical and can be discerned from context. Clearly not all gene fusions may carry biological functions like *BCR-ABL1* in Philadelphia chromosome leukemia. For example, chromosomal rearrangements that lead to inactivation of tumor suppressor genes (e.g., *CDKN2A*, *RB1*, *ATRX*) can be detected as out-of-frame fusion transcripts but are beyond the scope of this work. In this work, "oncogenic fusion" indicates in-frame fusions that produce oncogenic proteins like *BCR-ABL1*."

- For the HAL-01 cell line RNA-seq, please indicate the SRA accession for the data used.

[Response] We generated WGS and RNAseq data for the cell line. We updated the main text as following (in section "Data availability"): "The total RNAseq data for HAL-01 and UoC-B1 cell lines are deposited at ENA (<https://www.ebi.ac.uk/ena/browser/view/PRJEB55308>)."

- Including some details about how the computes were run, presumably in the cloud at St. Jude, would be helpful in the methods. Any time thousands of computationally intensive jobs are needed, it's useful for others to know about how such a feat was accomplished.

[Response] We updated the method section ("Fusion detection") to indicate the running time as following: "The fusion detection was performed in an institutional (St. Jude) high performance computing cluster with 227 nodes, 474 CPUs (12128 Cores) and 194TB of RAM and 20 petabytes of useable parallel file system storage, connected through 40 Gigabit Ethernet links. Although Cicero, Arriba and STAR-Fusion takes less than 2 days to finish for most of samples, we noticed that the earlier FusionCatcher version (v1.10) runs slowly for most samples. Therefore, we generated minibams by including known driver gene regions of childhood cancers (Supplementary Table 17) to validate the findings from Cicero, Arriba, and STAR-Fusion. Most of the jobs on these minibams can be finished around 10 days. Due to limited project storage space allocated for the laboratory, and the large storage space needed due to raw fastq and bam files as well as intermediate files, we analyzed the data in batches. The total download, re-download, run, re-run and analysis time of this cohort (using up to 500 jobs at any given time) took us about 1 year. Interestingly, a recent FusionCatcher version (v1.33) runs much faster for full bams (typically ~1 day) and we were able to finish the re-run of all four fusion detectors on all full bams/fastqs of our full cohort in less than 3 months for revision. All raw output from the four fusion detectors were deposited in zenodo (<https://doi.org/10.5281/zenodo.7033077>)."

- For the SDS statistic, how were the RNA-seq alignments derived? Using STAR or another aligner?

[Response] In section “Fusion detection”, we updated the description as “Oncogenic fusions were detected by using state-of-the-art methods reported to have superior performance^{18,20}, including Cicero (v0.3.0)¹⁸, Arriba (v1.2.0)¹⁶, STAR-fusion (v1.6.0)¹⁷, and FusionCatcher (v1.33)¹⁹. Cicero was run on bam files aligned with STAR v2.5.3a while Arriba, STAR-Fusion, and FusionCatcher were run on fastq files.”

We updated the SDS statistic calculation as: “To measure potential alternative splicing, we introduced a splicing dominance score (SDS; Fig. 4a). For this, we first calculated the read support (X_i) for all fusion versions i (with minimum of 3 supporting reads) detected in a sample (aligned with STAR v2.5.3a) with the index fusion.”

- Please include commands executed for the various fusion finders in the supplement. If workflows were used (ie. WDL, CWL, nextflow, etc.), providing the workflows would be useful too.

[Response] We clarified this information in section “Fusion detection”: “The commands used for the fusion detectors are as follows: (1) Arriba (v1.2.0): `arriba -o good_fusions.tsv -O discarded_fusions.tsv -a genome_assembly -g annotation_gtf -b black_list -T -P r1.fq r2.fq`; (2) Cicero (v0.3.0): `Cicero.sh -b bamfile -g genomeVersion -r cicero_refdir -j juncitons.tab`; (3) STAR-Fusion (v1.6.0): `STAR-Fusion --left_fq r1.fq --right_fq r2.fq --genome_lib_dir genome_lib`; (4) FusionCatcher (v1.33): `fusioncatcher -d DATADIR -i r1.fq,f2.fq`. Here r1.fq and r2.fq indicates the fastq files of read1 and read2, respectively.”

- Fig 6d - the g4 negative control is shown, but not the g5 negative control. Is there a reason why g5 was excluded here?

[Response] We have updated the caption of Fig. 6d to include g₅ data: “Negative control guide (g₄; see Extended Data Fig. 6b for a similar pattern in g₅) that targets upstream ...”

- In ST2, it would be useful to include the counts of number of samples for each data set, as listed in the methods section. Also, it would be useful to have another supplementary table that indicates the sample name and corresponding data set association, which can be cross-referenced in the other data tables. Other metadata about the individual samples would be useful too including the number of reads in each. This way, one can determine from the fusion read counts what proportion of the total reads they represent.

[Response] A new Supplementary Table ST2 is added (screenshot below). Sample and data set association is listed in Supplementary Table ST1. Read count from different tools are found in Supplementary Table ST3.

Table ST2. Data source of this study. Listed are data source tags (Column A) used in Table ST1, source publication or data portal URL (column D), short description of source study (Column C), and dataset ID from source URL when applicable (Column B) and number of patients involved (Column E).

Data Source Tag	Data Source Identifier (when applicable)	Short Description of Study	Source URL	#patient
CBTN		Children's Brain Tumor Network	https://portal.kidsfirstdrc.org	725
FredHutch	TARGET-AML	Pediatric AML	https://portal.gdc.cancer.gov/	1088
G4K	SJC-DS-1004	Genomes 4 Kids	https://platform.stjude.cloud/data/cohorts	253
PCGP	SJC-DS-1001	St. Jude-Washington University Pediatric Cancer Genome Project	https://platform.stjude.cloud/data/cohorts	777
RTCG	SJC-DS-1007	Real-time Clinical Genomics	https://platform.stjude.cloud/data/cohorts	1006
PMID_24436047		Rhabdomyosarcoma	https://pubmed.ncbi.nlm.nih.gov/24436047/	84
PMID_25010205		Ewing's Sarcoma	https://pubmed.ncbi.nlm.nih.gov/25010205/	62
PMID_25186949		Ewing's Sarcoma	https://pubmed.ncbi.nlm.nih.gov/25186949/	22
PMID_27798625		Pediatric AML	https://pubmed.ncbi.nlm.nih.gov/27798625/	23
PMID_29146900		Pediatric MDS	https://pubmed.ncbi.nlm.nih.gov/29146900/	14
PMID_30760869		Pediatric AML	https://pubmed.ncbi.nlm.nih.gov/30760869/	26
PMID_30926971		Acute Erythroleukemia	https://pubmed.ncbi.nlm.nih.gov/30926971/	8
PMID_31350825		Cytogenetically normal AML	https://pubmed.ncbi.nlm.nih.gov/31350825/	1
PMID_31697823		Therapy induced resistance in ALL	https://pubmed.ncbi.nlm.nih.gov/31697823/	101
PMID_33579957	SJC-DS-1011	Pediatric Therapy-related Myeloid Neoplasms	https://pubmed.ncbi.nlm.nih.gov/33579957/	43
PMID_34778799		Pediatric AML	https://pubmed.ncbi.nlm.nih.gov/34778799/	87
PMID_35176137		Discovery of UBTF as a novel AML subtype	https://pubmed.ncbi.nlm.nih.gov/35176137/	111
TARGET		NCI TARGET project	https://pubmed.ncbi.nlm.nih.gov/29489755/	759

- the MS-Excel ST3 table had a filter automatically invoked, showing only a small subset of the total data. Best to undo that filter before submission.

[Response] We now disabled filters in all supplementary tables.

- in Figure 6, the guide numbers for cell line UoC-B1 reuse values g₁ and g₂. I'd recommend starting where the count left off with the HAL-01 guide numbering, using instead g₆, g₇ for clarity.

[Response] We now use g₆ and g₇ for UoC-B1 line.

- The PWM scoring is reported to be in ranges of <2 to >6. It isn't immediately clear how such score numbers are being derived from the LLR equation. The LLR would usually give large negative values, where the least negative value reflects the highest probability. Please include more details on how the LLR converts to the score value.

[Response] In section "Calculating pseudo binding affinity for splice sites", we added below statements to clarify: "To ensure the quality of our constructed motifs, we scored all splice sites of known human genes and confirmed most of the splice sites received positive scores (>80% donors have score >4; >80% acceptors have score >4.3). As a negative control, we extracted 1.12 million potential donor (GT) sites and 1.76 million potential acceptor (AG) sites that do not belong to known human genes from forward strand of chr19 (one of the shortest chromosomes to save computation time) and scored them. As it turned out, >90% of such false donors have score <4 and >90% of such false acceptors have score <4.3, validating the power of our PWM method in discriminating real splice sites from non-real sites (see Supplementary Figure 2 for the score distribution of true and false splice donors and acceptors)."

- In extended figure 6, the row values don't always sum to 100 (usually 99), which I found curious.

[Response] We clarified this in caption of Extended Data Fig. 6 as "Due to rounding error, the row sums can be 100 or 99."

- in Fig 6D, 'non-lethal' is used correctly to describe the negative control situation for in-frame and

frameshift. In extended Figure 6, 'lethal' and 'non-lethal' are used for the negative control context. It might be better to use 'frameshift' and 'in-frame' instead, since both are non-lethal here as well.

[Response] Now the Extended Fig. 6b legend reads: “For panels **b** and **c**, instead of using lethal/non-lethal, a tag of frameshift/in-frame was given to each indel according to its length because the target region is genuine intronic (thus a negative control)”

- I'm curious why the RSB metric was chosen for pairwise comparisons as opposed to using an SDS like metric that accounts for the proportion by summing the total in the denominator. Would an SDS-like RSB metric outperform the RSB itself for ranking hazard ratios or be simpler for identifying best therapeutic candidates?

[Response] This comment significantly improved our presentation. We updated in section “Selection bias in intronic versioning” as: “Because intronic versioning can cause amino acid differences in the fusion protein which may in turn lead to potential functional difference, we hypothesized that intronic versioning could confer differential fitness to the host cells in some oncogenic fusions. To measure the effect size of potential selection bias between two intronic versions, we proposed a relative selection bias score (RSB) based on the observation that DNA breakpoints are generally distributed in introns in a near-uniform fashion and gene length can predict prevalence in patients (Fig. 2h-i). In this model, the patient prevalence of DNA breakpoints in a given intron should be proportional to its length if the resultant protein versions are functionally equivalent (i.e., confers the same positive selection pressure). The statistical significance of selection bias is measured by comparing (using a Chi-squared test) the observed patient prevalence in all intronic versions and corresponding expected patient prevalence under a null hypothesis that patient prevalence is proportional to intronic length.”

- In ST3 (and ideally, the original unfiltered starting fusion set), please include the programs that initially predicted them along with fusion evidence read support. For ST3, the NeoVersioner recomputed read support for predicted fusions would be useful for comparing to the originally defined read support quantities from the individual tools.

[Response] Please see patient level fusion detection along with the detection tools in Supplementary Table ST1. Read count from different fusion detection tools are provided in Supplementary Table ST3. Please refer to Supplementary Table ST7 for read count information from NeoVersioner.

- in NeoSplicer, it isn't obvious how the non-template fusion sequences were defined as shown. The algorithm for doing this should be described as part of the NeoSplicer methods section.

[Response] We clarified this in section “Neo-Splicer” as: “To detect novel splice sites, this method requires DNA breakpoints between the two genes along with the non-template insertion sequence if there is, since we note that sometime the novel splice sites can be embedded in the non-template insertion sequences⁶⁷ flanking the rearrangement boundary. Given the ubiquitous nature of candidate splice sites (AG and GT; 1 in every 16 nucleotides expected by random chance), we first detected putative splice sites by using Position Specific Weight Matrix method as described in section “Calculating pseudo binding affinity for splice sites”. Second, given the DNA breakpoints (39% (n=768/1,988) chance of detection in RNAseq data, Fig. 2j-l; Supplementary Table 8) of an oncogenic fusion, we enumerated all AG and GT dinucleotides between intact exons of involved genes, generated hypothetical exons, and checked corresponding translation frames. RNAseq reads were then compared with above predictions to determine the neo splice sites and corresponding isoforms used by the cancer cells (Extended Data

Fig. 1b). We note that the cryptic neo-spliced exon and/or non-template insertion sequences can remain poorly mapped by standard mapper (here STAR v2.5.3a), such as the non-template sequences in *TCF3-HLF* fusion in HAL-01 and UoC-B1 (**Fig. 6b,f; Extended Data Fig. 5d**), especially when the neo-splice sites are within non-template sequences (fusion *C11orf96-MAML2* in case SJEPD031093 in **Extended Data Fig. 5a**). These mapping challenges are resolved by Neo-Splicer using DNA contigs.”

- it is curious that many of the analysis tools used in this work were hand-crafted by the authors. For example, instead of using commonly leveraged alignment utilities, the authors used their own kmer-based read mapping schemes. Such analyses, like identifying DNA breakpoints from RNA-seq data in intronic regions, being dependent on custom utilities (instead of using trusted alignment methods like bowtie, STAR, bwa, etc) does make one question the accuracy of the methods being used. Showing a situation such as uniform insertions across intronic regions using custom methods, where we know intronic regions are repeat-rich and more difficult to accurately align reads to than exonic regions, could result from inaccurate / random placement of reads. I'm not suggesting that the authors methods are inaccurate, but rather suggesting that the authors would be best to demonstrate that their methods can accurately perform their function, such as by simulating data and ensuring accurate results using simulated data. This would be most important for NeoSplicer and the method inferring DNA breakpoints from RNA-seq data.

[Response] This comment greatly improved the clarity of our presentation. We updated the strength of our developed methods in section “Model of fusion etiology and study design” as: “To classify the detected oncogenic fusions into one of above four categories, we developed novel tools Neo-Versioner (to classify intronic versioning) and Neo-Splicer (to classify neo-splicing; see **Method** and **Extended Data Fig. 1**). If the fusion does not belong to either intronic versioning (i.e., no reads supporting natural exonic junctions between N' and C' genes) or neo splicing categories by our automated analysis, we manually review and classify the fusion into the categories of either chimeric exon or neo translational.”

Regarding the validation DNA breakpoints inference, we believe our real-world DNA-sequencing based validation (in section “Landscape of childhood oncogenic fusions” of original submission) can best serve this purpose than a simulated dataset: “This hypothesis implies that all eligible base pairs (under the constraints of splicing and translation frame; mostly intronic bases) in corresponding genes can contribute to functional oncogenic fusion and therefore DNA breakpoints should be uniformly distributed along the eligible introns. To test this hypothesis, we first validated the high accuracy (>91% RNA-based detections are within 5 base pairs of DNA-based detections; see **Method; Supplementary Table 8; Extended Data Fig. 2a**) of detecting DNA breakpoints by using RNAseq data.”

We also clarified our motivation for developing a k-mer based approach over standard mappers: “To detect novel splice sites, this method requires DNA contig around the rearrangement as input. We note that sometime the novel splice sites can be embedded in the non-template insertion sequences⁶⁵ flanking the rearrangement boundary. Given the ubiquitous nature of candidate splice sites (AG and GT; 1 in every 16 nucleotides expected by random chance), we first detected putative splice sites by using Position Specific Weight Matrix method as described in section “Calculating pseudo binding affinity for splice sites”. Second, given the DNA breakpoints (39% (n=768/1,988) chance of detection in RNAseq data, **Fig. 2j-l; Supplementary Table 8**) of an oncogenic fusion, we enumerated all AG and GT dinucleotides between intact exons of involved genes, generated hypothetical exons, and checked corresponding translation frames. RNAseq reads were then compared with above predictions to determine the neo splice sites and corresponding isoforms used by the cancer cells (**Extended Data Fig. 1b**). We note that the cryptic neo-spliced exon and/or non-template insertion sequences can remain

poorly mapped by standard mapper (here STAR v2.5.3a), such as the non-template sequences in *TCF3-HLF* fusion in HAL-01 and UoC-B1 (Fig. 6b,f; Extended Data Fig. 5d), especially when the neo-splice sites are within non-template sequences (fusion *C11orf96-MAML2* in case SJEPO31093 in Extended Data Fig. 5a). These mapping challenges are resolved by Neo-Splicer using DNA contigs.”

- Line 81: typo, should read "(C-terminus)"

[Response] Now it reads “Oncogenic fusions typically involve two genomic loci (genes) denoted as N’ gene (N-terminus) and C’ gene (C-terminus).”

- Lines 177, 517: RNA-seq does not provide 'absolute expression level'. RNA-seq provides relative expression levels (normalized by total read sequencing depth). Perhaps just change to 'expression level'.

[Response] Now it reads “By further calculating expression level measured” and “We calculated FPKM values as previously described⁷ to study the expression level of C’ genes in the promoter-hijacking-like category”

- Line 247: typos, should be 'division with denominator three equals zero'

[Response] Now it reads “the remainder of a division with denominator 3 equals 0”

- Line 402: typo, rationale

[Response] Now it reads “to enable rationale design of guide RNAs”

- Line 588: typo, pools

[Response] Now it reads “for analyzing splice site disruption in the edited cell pools”

Reviewer #3 (Remarks to the Author): Expert in paediatric cancer genomics and gene fusions

This study by Liu et al. provided information on the etiology of 1,989 fusions identified in 5,190 childhood cancers. This report studied a wide spectrum of cancer, including leukemia, CNS, and solid tumors. The manuscript was well-written and provided exquisite detail. The authors identified 5 different molecular mechanisms by which oncogenic fusions may be generated, highlighting the large diversity in types of fusions. While different types of fusions have been described extensively in prior literature, Liu et al provided additional biological context into not only general classes of fusions but specific fusions as well. This work does demonstrate some clinical and therapeutic implications, which I would be interested in seeing highlighted more as childhood cancers tend to be underrepresented with regard to targeted therapies and clinical trials.

[Response] We appreciate Reviewer #3’ encouraging comments. We added below statement to stress the fact that childhood cancers are underrepresented: “To address these questions, and with the consideration of childhood cancers are underrepresented in targeted therapies⁴, we comprehensively studied oncogenic fusions by using tumor transcriptome sequencing datasets of 5,190 patients from publicly available childhood cancer cohorts.”

Major points:

- I do not see a caption for Supplementary Figure 1

[Response] Thanks for catching this oversight. We have added the caption as following: “Recurrent ($n \geq 3$) oncogenic fusions identified in 5,019 childhood cancers. For each gene pair, coding exons (thick boxes) are colored white (frame 0), gray (frame 1), and black (frame 2). Intronic length is indicated by numbers. Gray lines indicate theoretically in-frame fusions. Red lines indicate in-frame fusions observed in patients and its width indicate patient prevalence. RefSeq identifiers are indicated for each involving gene. Fusions involving neo-splicing or chimeric exons were not included (see paper).”

- Extended Data Figure 3 (Lines 37-39) “On the other hand, FOXO1 appears to have higher expression than PAX3/PAX7 in normal brain tissues, consistent with the observation in brain tumors (Fig. 3b).” Why is normal brain tissue from GTEx being used as the comparator for FOXO1 expression when this fusion is found predominately in muscle/soft tissue?

[Response] We apologize for this typo. We updated the sentence as “On the other hand, FOXO1 appears to have higher expression than PAX3/PAX7 in normal muscle/soft tissues, consistent with the observation in rhabdomyosarcoma, a type of sarcoma made up of cells that normally develop into skeletal muscles (Fig. 3b).”

- For those fusions with alternative splicing, did the authors look at other transcript isoforms to see if fusions derived from alternative splicing may be due to multiple transcript isoforms for the same gene being expressed in the tissue type under study?

[Response] We updated the main text (section “Alternative splicing in oncogenic fusions”) as “For exons involved in alternative splicing, we also investigated whether they could match any known isoforms of the host gene. We detected only one recurrent alternative exon (exon 12 in NM_016320) of NUP98 that matches a second isoform NM_001365129 (Supplementary Table 11).”

- The section on alternative splicing in fusion generation was intriguing since through clinical testing we have noted a number of fusions, including those involving RUNX1 and ABL1 (to name a couple), more frequently present with multiple fusion breakpoints in the same analyzed tumor sample. We have wondered whether alternative splicing has a role here and clinically which breakpoint is more informative to the providers or clinical outcomes. Did you find any samples in your data where different breakpoints representing the same fusion were seen in the same patient sample and if so, how did you determine which one was the predominant isoform?

[Response] Indeed, we observed RUNX1 alternative splicing in the original submission: “As it turned out, alternative splicing in *ETV6-RUNX1* (identified in B-cell leukemia) is recapitulated in *RUNX1* gene in normal GTEx blood samples, and alternative splicing in *C11orf95-RELA* (identified in Ependymoma, a brain tumor) is recapitulated in *C11orf95* in normal GTEx brain samples (**Extended Data Fig. 4c**).”

We further clarified this in section “Discussion”: “Clearly, alternatively spliced exons and corresponding protein domains should be avoided in drug targeting due to their potentially dispensable nature. Our data also highlights the need to study the clinical implication of alternative splicing in oncogenic fusions. For example, in sample SJBALL030563_D1 (with fusion *ETV6-RUNX1*), we discovered 11 reads supporting junction E5-E3 and 7 reads for junction E5-E4 (Supplementary Table ST7). Although we determined the predominant isoform (E5-E3) by read supports, it would be interesting to study if one or both of chimeric proteins are translated and the tumorigenesis function of these chimeric proteins.”

- Given the title, I was hoping for additional clinical and therapeutic implications with evidence provided

from this study. The authors hypothesize that certain fusion etiologies may be more appropriate for targeted therapies or CRISPR methodology but only provide data for a single fusion event and EFS. While, the CRISPR approach to targeting some of the fusions, particularly those in which there isn't a targeted therapy available is interesting and promising, it is likely still a long way from clinical use. Additionally, those fusions that were hypothesized to be good drug targets lack a targeted therapeutic. Given that there is RNA-seq for all of these data, it may be interesting to look at patterns of expression to see if there is overexpression or evidence for a pathway deregulation which may be targeted with currently available therapeutics (i.e. MEK inhibitors for KIAA1549-BRAF), although this may be beyond the scope of the manuscript. Can the authors provide any additional outcomes data for the recurrent fusions described in this manuscript?

[Response] We appreciate Reviewer #3's curiosity into the targeting of oncogenic fusions. We are currently testing our CRISPR approach in additional 20 cell lines harboring neo-splicing or chimeric exons. However, the extent of that study would warrant a separate manuscript. We clarified this in Discussion: "However, we provide proof-of-concept data for TCF3-HLF fusion, a rare ALL subtype associated with a high rate of treatment failure⁴⁸ that clearly would benefit from such targeted therapy. It would be interesting to test if this strategy is broadly applicable to all oncogenic fusions with neo-splicing or chimeric exon feature. Furthermore, additional studies are needed to develop innovative targeting strategies for patients without such "easy-to-design" targeting strategies."

As agreed by Reviewer #3, this work is primarily focused on genetic vulnerabilities encoded in the DNA sequences and therefore overexpression and pathway deregulation are beyond the scope of this work.

Unfortunately, the outcome data are typically not well curated for most cancer genomics profiling data. We expect such analysis to be better streamlined in future cohorts with a dedicated study design.

Minor points:

- Figure 1b and line 284 - The "NM1-PATZ1" fusion should be listed as "MN1-PATZ1"

[Response] Now line 284 reads "where all four MN1-PATZ1 cases in". Fig. 1b also updated.

- Ensure that gene names or fusions that are being discussed as the DNA or RNA are italicized

[Response] All gene names are italicized in main text.

- The tool FusionCatcher is written in as both FusionCatcher and Fusion-catcher. Please update to maintain consistency

[Response] We now use "FusionCatcher" throughout the manuscript

- Methods, Neo-Splicer (Line 501) – "where" should be "were"

[Response] Now it reads "RNAseq reads were then compared"

- Extended Data Figure 2 caption (Line 17) – "show" should be "shown"

[Response] Now it reads "Shown in y-axis is the number of DNA breakpoints"

- Extended Data Figure 2 caption (Line 23) – “RUNX1-RUN1T1” should be “RUNX1-RUNX1T1”

[Response] Now it reads “for oncogenic fusion RUNX1-RUNX1T1”

- Supplementary Table 3 – How was the number of supporting reads calculated? How did you account for potentially discrepant coordinates from the four fusion callers?

[Response] We clarified this question as below: “A significant challenge in using public fusion detection tools is the harmonization of their output. For breakpoints, we used 10 bp as buffer distance to harmonize their output (Supplementary Table 3). However, there is no straightforward way to harmonize read counts from different tools. Therefore, Neo-Versioner was used throughout the manuscript for all applicable analysis.”

- Figure 3b – “CBFB-MY11” should read “CBFB-MYH11”

[Response] Now it reads “CBFB-MYH11”

REVIEWER COMMENTS

Reviewer #1 (Remarks to the Author):

The manuscript has been substantially improved and in the current form is acceptable for publication.

Reviewer #2 (Remarks to the Author):

While I appreciate several of the edits to the revised manuscript, the revisions unfortunately do little to address many of my earlier concerns. Most notably, a major failing of the work is the arbitrary decision to restrict all analyses to in-frame coding fusions. This ignores fusion variants to protein-coding genes that would lead to altered or truncated coding regions and important promoter-hijacking fusions. Such fusions are highly relevant to pediatric cancers. For example, there is no mention of oncogenic fusion CRLF2::P2RY8, which is highly relevant to pediatric ALL, involves two protein-coding genes, but does not yield an in-frame fusion protein. Also, important promoter-hijacking oncogenic fusions relevant to pediatric cancer that are actually mentioned in the paper, including IGH::CRLF2/EPOR/DUX4, are not actually studied as part of this work, yet would be especially relevant in analyses involving fusion impact on expression as in Figure 3, related to the expression dominance score. Other analyses including the relative selection bias score continue to fail to consider transcript variant breakpoints that do not lead to in-frame coding regions. Because of this, there could very well be actual promoter hijacking fusion variants that yield the dominant fusion variant that are entirely disregarded when a lower expressed in-frame variant exists.

Another major shortcoming of the manuscript is that it continues to be heavily biased towards fusions selected by the authors as relevant to pediatric cancer based on having a gene partner or partners that have previously been defined as oncogenes – now made explicitly clear by the recent revision. The authors continue to not make available as supplementary material the complete list of fusion candidates identified by running the various fusion-finding algorithms, but instead provide their preselected list of fusions they deem as relevant for this analysis. Hence, we continue to find no actual reporting of prevalence of pediatric cancer relevant fusions mentioned above as we would expect of a comprehensive survey.

Often, the number of fusions analyzed in any specific analysis total few in number, and this is due to the need for sufficient patient prevalence as part of the analysis. This is understandable, but also unfortunate, because it reduces the impact of the work. The abstract is written in a grandiose style with

little to no specific details reported. It is only when reading the main text that one discovers that only in-frame coding fusions were examined, up to 20 fusions were explored for impacts on splicing, and a single fusion was found to have clinically relevant prognostics. It would be best to include specifics in the abstract. Instead of 'a subset of oncogenic fusions', indicate which ones and/or how many. Also, related to this last point in the main text, it isn't indicated how many other fusions were tested using hazard ratios and not found significant.

The finding that patient prevalence of KMT2A fusion partners is well correlated with gene length is quite interesting and clearly significant. However, no specific evidence is given that other oncogenic fusions behave similarly. There may also be KMT2A fusion partners identified as candidate fusions in the initial survey but not selected by the authors for this particular analysis, but this is not clear given that the full set of initial fusion candidates was not reported. As mentioned in my earlier review, the correlation between gene length in leukemia fusions (p -value = 0.05) is tenuous, and there was no significant association found for the other tumor types analyzed. I suspect the KMT2A fusions contributed greatly towards the marginally significant finding with a $p \leq 0.05$ threshold. Removing KMT2A fusions from that analysis would possibly make the finding non-significant. If the authors would make the data for these analyses available as supplementary materials, it would make it possible for others to more easily directly explore them.

Reporting that DNA fusion breakpoints are found mostly to occur uniformly in relevant introns makes sense. Reporting that RNA-based fusion breakpoints are generally found within 5 bases of a DNA breakpoint does not. While I haven't examined the authors references for this finding, based on my own experience and based on basic splicing biology coupled with uniform DNA breakpoint data, it just simply does not make sense whatsoever. Uniform DNA breakpoints across long intronic regions cannot be within 5 bases on average from a splice junction. This is clear from the authors own figure 5a. I could be misinterpreting something here, because this is too bizarre to have a disagreement over. The data for Figure 2a should be made available, and it should be examined to see if there are specific fusions that are biasing results, such as RUNX1/T1 where breakpoints are not found as uniform and the fusion is one of the most prevalent.

The selection bias measurement has a null hypothesis stated as having patient prevalence proportional to intronic length. This seems intuitive, but I see no evidence presented for this null hypothesis to be valid. The authors provide evidence that DNA breakpoints are often found uniformly distributed at sites of fusions, and there is unconvincing evidence that (except for KMT2A fusions) that gene length is associated with patient prevalence, and so I suspect the authors are attempting to make a jump here in using these other limited data to support this null hypothesis. Even so, based on my earlier recommendation, the authors included results based on not normalizing for intron length, and I found these results to be quite interesting, and rather than contributing false positives or false negatives, instead highlighting those fusions where the statistics are more likely to be robust indicators of relevant findings. Ultimately, while length normalization is intuitive, the authors have no compelling evidence that it is required, even though attempts are made to do so.

Reviewer #3 (Remarks to the Author):

Thank you for the thoughtful edits in response to all reviewers. The authors have sufficiently responded to my comments.

Reviewer #1 (Remarks to the Author):

The manuscript has been substantially improved and in the current form is acceptable for publication.

Reviewer #2 (Remarks to the Author):

2.1a While I appreciate several of the edits to the revised manuscript, the revisions unfortunately do little to address many of my earlier concerns. Most notably, a major failing of the work is the arbitrary decision to restrict all analyses to in-frame coding fusions. This ignores fusion variants to protein-coding genes that would lead to altered or truncated coding regions and important promoter-hijacking fusions. Such fusions are highly relevant to pediatric cancers.

[Response] We regret that Reviewer #2 still wants us to include truncated coding (tumor suppressor) genes after our expanded description on why tumor suppressor genes are ignored, in our first rebuttal. To stress that our rationale is carefully thought, we added **Supplementary Notes** in this revision, which will be collectively addressed in rebuttal section 2.1c.

2.1b For example, there is no mention of oncogenic fusion CRLF2::P2RY8, which is highly relevant to pediatric ALL, involves two protein-coding genes, but does not yield an in-frame fusion protein. Also, important promoter-hijacking oncogenic fusions relevant to pediatric cancer that are actually mentioned in the paper, including IGH::CRLF2/EPOR/DUX4, are not actually studied as part of this work, yet would be especially relevant in analyses involving fusion impact on expression as in Figure 3, related to the expression dominance score.

[Response] This “protein truncating” and “promoter-hijacking” question is collectively addressed in section 2.1c. We also added a **Supplementary Note 8b** to detail the reason why RNAseq data alone is not sufficient to study promoter hijacking as below:

8.b) Oncogenic fusions in the category of promoter/enhancer hijacking. In this category, a rearrangement can bring a strong promoter/enhancer to an otherwise silenced proto-oncogene and lead to its aberrant expression. When the novel promoter is far, the proto-oncogene may start its transcription from its own transcription start site, thereby leaving no split reads or discordant read pairs in RNAseq data for bioinformatic detection (**Fig. SN2a**). On the other hand, the transcripts may contain part of the novel promoter sequences when the novel promoter is closer (**Fig. SN2b**). Moreover, it is also possible that a point mutation in the native promoter can convert it to a strong active promoter (such as *TAL1* in pediatric T-ALL¹⁵) and lead to aberrant expression (**Fig. SN2c**)— biologically this scenario is not promoter/enhancing *per se*. Clearly, without DNA (preferentially whole genome) sequencing data, scenarios a) and c) cannot be resolved by transcriptome sequencing and, a forced analysis will result in biased conclusions that does not meet our scientific rigor. Instead, such patterns are best studied in our ongoing project on the signatures of rearrangements (SVs) using whole genome sequencing in >1,500 pediatric cancer patients. Nevertheless, we provided results on known oncogenic fusions (*CRLF2*, *DUX4*, *EPOR*, *BCL11B*, **Table SN2a-e**) in promoter/enhancer-hijacking category to address Reviewer #2’s request though we did not perform systematic discovery.

Fig. SN2. Promoter/enhancer-hijacking. In hypothetical scenario (a), the blue chromosome (and a strong enhancer/promoter highlighted by blue oval) was brought proximity to the orange proto-oncogene and lead to its aberrant expression. Transcription only involves the proto-oncogene due to the space between the blue promoter and orange gene. In (b), the blue promoter contacts the transcription start site of orange gene, so that the transcription involves both the proto-oncogene and a small part of the blue chromosome. In (c), a point mutation in the promoter region may convert it to a strong active promoter to initiate the proto-oncogene without a fusion event (such as *TAL1* enhancer mutation in Mansour et al (2014)¹⁵. Split reads or discordant read pairs are expected for scenario (b) but not scenario (a) or (c). DNA (preferentially whole genome) sequencing are needed to ascertain the fusion status. Nevertheless, the aberrant high expression of such proto-oncogene typically can help ascertain the tumor subtype.

2.1c Other analyses including the relative selection bias score continue to fail to consider transcript variant breakpoints that do not lead to in-frame coding regions. Because of this, there could very well be actual promoter hijacking fusion variants that yield the dominant fusion variant that are entirely disregarded when a lower expressed in-frame variant exists.

[Response] Although it is tempting to detect all cancer related alterations from RNAseq data, we regret that Reviewer #2 significantly over-estimated the power of RNAseq alone. In fact, in clinical testing (e.g., <https://pubmed.ncbi.nlm.nih.gov/34301788/>), DNA sequencing, in particular whole genome sequencing (WGS), is always preferred to include. This is because gene loss (such as *CDKN2A* deletion in leukemias) can be only definitively called in WGS while in RNAseq data this can remain as an inference by low/no expression when split read is absent. Although tumor suppressor gene can be lost due to gene truncation (as requested repeatedly by Reviewer #2), whole gene loss due to focal deletion is also frequently observed (e.g., *CDKN2A*). In this regard, detection of tumor suppressor gene loss in RNAseq data is NOT guaranteed, which is in clear contrast with activating mutations such as *KRAS* G12D or oncogenic fusions (that must be expressed to exert their biological functions) as investigated in this work. Therefore, in this work we intentionally skipped tumor suppressor genes using RNAseq data because we do not want to provide a potentially biased picture of such events that compromise our scientific rigor. Similarly, for promoter hijacking the detection of DNA breakpoints are not guaranteed in RNAseq (see rebuttal in 2.1b). In this revision we drafted a comprehensive background on the rationale of our study design as below. However, most of the information in this background is common knowledge in cancer genomics and we therefore put them into **Supplementary Notes** rather than in the main text/online methods.

1. Clinically recognized oncogenic fusions that generate chimeric proteins

Since the discovery of *BCR-ABL1* fusion oncoprotein in Philadelphia-chromosome leukemia¹, classical cytogenetics method has revealed many additional cancer subtypes such as $t(1;19)(q23;q13)$ ² that was later determined to generate *TCF3-PBX1*³, $t(17;19)$ ⁴ that was later determined to generate *TCF3-HLF*⁵ and $t(11;22)(q24;q12)$ that was determined to generate *EWSR1-FLI1*⁶. With the advent of next generation sequencing technologies, oncogenic fusions are increasingly discovered in the past decade via simultaneously interrogating the genome (DNA sequencing) and the transcriptome (RNA sequencing) of tumors from patient cohorts of similar diagnosis, where DNA and RNA data cross validate each other. This include *C11orf95-RELA* (*C11orf95* recently renamed as *ZFTA*) in pediatric ependymomas (EPD)⁷, *KIAA1549-BRAF* in pediatric low-grade glioma (LGG)⁸. Supporting evidence from both DNA (on the rearrangement) and RNA (on the generation of chimeric protein) is a critical feature of these findings.

2. Clinically recognized oncogenic fusions that leads to aberrant expression of proto-oncogenes

In addition to the above “conventional” oncogenic fusions where a chimeric fusion oncoprotein is generated, there is another category of “promoter-hijacking” fusions. In this category, a constitutively active promoter or enhancer region is brought to a proto-oncogene (via chromosomal rearrangement) that is typically silenced in corresponding lineage of the cancer cells. Such rearrangement leads to aberrant expression of the proto-oncogene. Prominent examples of this category include *CRLF2/DUX4/EPOR* aberrant expression via rearrangement to immunoglobulin heavy chain (*IGH*) region B-ALL^{9,10}, *TAL1/TAL2* aberrant expression via rearrangement to T-cell receptor region (*TCR*) in T-ALL¹¹, *GFI1* aberrant expression via intra-chromosomal rearrangements to active enhancers in medulloblastoma¹², *CRLF2* aberrant expression via intra-chromosomal rearrangement to *P2RY8* promoter¹³, as well as our newly discovered *BCL11B* aberrant expression in lineage-ambiguous leukemia¹⁴. Because no chimeric proteins are generated, this fusion category is typically termed “promoter/enhancer-hijacking”. Interestingly, other mutational mechanisms can also lead to such aberrant expression of proto-oncogenes. For example, the seminal work by Thomas Look and colleagues¹⁵ has demonstrated that small insertions/deletions in enhancer regions of proto-oncogene *TAL1* can be sufficient to lead to its aberrant expression in pediatric T-ALL. Although corresponding tumors do not have chromosomal rearrangements (or fusion events) involving *TAL1*, we still consider these tumors as *TAL1* category.

3. Functional evidence of clinically recognized oncogenic fusions

Although experimentally challenging, putative oncogenic fusions such as *ZFTA-RELA* (also known as *C11orf95-RELA*) have recently been shown to be sufficient to drive pediatric ependymoma¹⁶. On the other hand, the success of imatinib on *BCR-ABL1*¹⁷, and the genetic knockout of oncogenic fusions such as *TCF3-HLF* in this work, have demonstrated that these oncogenic fusions, or more precisely the fusion oncoproteins they encode, plays an essential role to the survival of host cancer cells, which forms the basis of the hypothesis “oncogene addiction” that posits on the therapeutic value of targeting these oncogenic fusions.

4. Clinically recognized oncogenic fusions being invariable to clonal evolution and initiating driver

Comparison of tumors collected at initial diagnosis and at relapse for pediatric leukemia^{13,18} has reinforced the notion that subtype-defining oncogenic fusions are cancer initiating events¹⁹. In these studies, the oncogenic fusions are always conserved between diagnosis and relapse tumors, although other subclonal mutations (e.g., *CDKN2A* loss and *NT5C2* gain-of-function

mutations) can be either eradicated or *de novo* acquired from diagnosis to relapse¹³. The clonal nature (i.e., being present in all cancer cells) of oncogenic fusions thus renders them ideal therapeutic targets.

5. Oncogene versus tumor suppressor gene (TSG)

In addition to the many oncogenic fusions mentioned above, extensive genome sequencing efforts in the past decade have led to the discovery of many additional significantly mutated genes also known as cancer drivers, in both adult^{20,21} and childhood cancers^{9,22}. In observation of these many cancer driver genes, Bert Vogelstein and colleagues²³ have pioneered the concept of classifying cancer driver genes into “tumor suppressor genes (TSG)” and “oncogenes”, where a “TSG” is a gene that, when *inactivated* by mutation, increases the selective growth advantage of the cell in which it resides, while an “oncogene” is a gene that, when *activated* by mutation, increases the selective growth advantage of the cell in which it resides. Under this broad concept, the oncogenic fusions mentioned above belong to the category of “oncogene” because corresponding fusion oncoproteins are hyperactive. On the other hand, the well-known TSGs including *CDKN2A* and *RB1*^{9,22} typically demonstrate inactivating (also known as loss-of-function) mutations, including gain of stop codon, protein-frame shifting, splice site altering, whole gene loss due to large deletion, or partial gene truncation due to focal deletion. A model of functional consequences on TSGs and oncogenes from diverse mutation types are illustrated in Fig. SN1 a-f.

Fig. SN1a. Diverse mutation types to disrupt a tumor suppressor gene (TSG; a) and less diverse mutation types for hyperactivation (b). In a TSG, a mutation (#1) that disrupts promoter or enhancer can lead to expression loss, a mutation (#2) that disrupts translation start codon ATG, a mutation (#3) that disrupt the gene structure via an intronic breakpoint, a mutation (#4) that disrupts the splice sites, and a mutation (#5) that disrupt the protein codon can all lead to loss of function (total gene deletion not illustrated). On the other hand, there are limited ways to make an oncogene hyperactive (panel b), which include a stronger promoter/enhancer via mutation #1, a stronger amino acid via mutation #3, or forming a chimeric protein via rearrangement mutation #2.

Fig. SN1b. Example oncogene *KRAS*. Chromosome coordinates (chr12) are shown on top and gene structure shown on bottom. *KRAS* mutations detected from pediatric cancers were shown as numbers that also indicate protein amino acid change. For example, there are 63 tumor specimens having mutations resulted in G12D, 22 tumor specimens having mutations resulted in A146T. Clearly, these protein coding mutations are in-frame and therefore can generate a mutant protein. Data in Fig. SN1b-f adapted (Oct 15, 2022) from <https://pecan.stjude.cloud/>.

Fig. SN1c. Example oncogene *ABL1*. In addition to the few point mutations detected from thousands of pediatric cancers, a prominent observation in *ABL1* is fusions, including 41 *BCR-ABL1*, 4 *ETV6-ABL1*, and a few other rare fusions such as *RCSD1-ABL1*, all shown as half-white-half-black circles. Breakpoints aligned with exon boundaries represent RNA breakpoints (i.e., splice junctions) while breakpoints not aligned with exon boundaries represent DNA breakpoints (black arrows). While DNA breakpoints can sometime be detected from RNAseq data (in total RNAseq protocol where pre-mRNA are included, see section SN 9 “Predicting DNA breakpoints from RNAseq data” in this Supplementary Notes), whole genome DNA sequencing typically ensure ascertainment of DNA breakpoints. On the other hand, DNA sequencing data typically cannot give definitive clue on RNA breakpoints (i.e., splice junctions) due to possibility of alternative splicing, such as *KMT2A* rearrangements in Fig. 4d of this work.

Fig. SN1d. Example oncogene *MYCN*. Beside P44L mutation in 29 tumor samples, *MYCN* does not have other highly recurrent alterations except copy number gain (red horizontal color bars in bottom) that are enriched in solid tumor (ST) neuroblastoma (NBL) that is detected in 19% of tumors (n=148). Further, *MYCN* amplification is also detected in Wilms tumor (ST, WLM) with frequency 11% (n=14). Note the common amplified region of *MYCN* and sometime the amplification can extend to far flanking regions of *MYCN*.

Fig. SN1e. Example tumor suppressor gene *RB1*. *RB1* gene has diverse mutation types in pediatric cancers, including stop gain mutations such as W78* in 8 specimens, R320* in 5 specimens. We also observed frameshifting mutations such as A74fs, L64fs, L317fs, D856fs. Moreover, the half-white-half-black circles indicate enrichment of structural rearrangements such as to gene *RCBTB2* in 5 patients and another 9 patients to another region in chr13. Further, focal deletions were detected in pediatric T-ALL (6%) and B-ALL (2%) that removed last several exons of *RB1*.

Fig. SN1f. Example tumor suppressor gene *CDKN2A*. Unlike TSG *RB1*, *CDKN2A* is enriched with copy number loss in pediatric T-ALL (54%) and B-ALL (11%). Although in some tumors the detection can be so focal that only few exons are affected (in this case it is possible to detect a truncating “fusion” from RNAseq data), in many tumors the size of deletion can be as big as arm level so that no truncating “fusion” transcripts are expected in RNAseq data.

6. Clinically recognized fusion-negative samples

Although oncogenic fusions have been routinely used for clinical subtyping, not all human cancers are fusion positive. For example, in pediatric B-ALL it has long been known that fusion-negative subtypes exist, including hyperdiploid (that with >50 chromosomes) and hypodiploid (those with <45 chromosomes) B-ALL²⁴. In pediatric neuroblastoma, extensive efforts in the study of whole genome, exome, and transcriptome sequencing data have not identified clinically

meaningful oncogenic fusions for most samples, except the well-known high *MYCN* amplification in ~20% of patients^{9,25,26}. In malignant rhabdoid tumours, *SMARCB1* homozygous loss is the only hallmark of nearly all patient tumors²⁷. Similarly, *RB1* homozygous loss is the only hallmark of nearly all retinoblastoma tumors²⁸. Clearly, candidate fusions detected in tumors of fusion-negative subtypes such as hyperdiploid B-ALL, neuroblastoma, rhabdoid or retinoblastoma tumors are more likely passenger events, if not artefacts, and scrutiny is warranted before accepting them as a true oncogenic fusion, as will be discussed in section **SN 11g** on mutual exclusivity pattern among oncogenic fusions. This data highlights the critical need of knowledge on well-defined tumor subtypes to ensure scientific rigor in reporting novel oncogenic fusions. In fact, clinically-relevant novel fusion-negative subtypes continue to be discovered, such as the novel subtype of *UBTF*-ITD in pediatric AML among the known clinical fusion-negative subtypes of *NPM1* and *CEBPA*²⁹.

7. Remarks on clinically recognized oncogenic fusions

The above data highlights a few characteristics of clinically recognized oncogenic fusions such as *BCR-ABL1*: 1) to date all of these fusions are in-frame and activating (i.e., TSG does not belong to the category of oncogenic fusions); 2) promoter/enhancer-hijacking can be regarded as a different category of oncogenic fusion because they do not generate chimeric proteins; 3) these fusions are subtype-defining so that typically we see no more than one fusion per tumor, also known as mutual exclusivity rule²⁹ that will be discussed in section **SN 11g**; 4) despite extensive clonal evolution during the course of the life span of a tumor, subtype-defining oncogenic fusions typically remain intact; 5) like *ZFTA-RELA* (also known as *C11orf95-RELA*) and *TCF3-HLF*, these fusions are expected to be functionally sufficient and necessary to the host cancer cells; 6) not all human cancers are expected to have oncogenic fusions. Interestingly, to date all clinically recognized oncogenic fusions in pediatric cancers have supporting evidence from both DNA and RNA sequencing data whenever both data types are available, highlighting a critical bioinformatic pattern during technical evaluation of candidate oncogenic fusions.

8. Study design of this work

The above molecular mechanistic insights lead us to following strategy in this study design.

8.a) Tumor suppressor genes. Due to the diverse mutation types (including substitutions (SNVs), small insertion/deletions (Indels), copy number loss (CNVs), or structural alterations (SVs)) that can all lead to loss-of-function, we always rely on DNA sequencing (especially whole genome sequencing) to definitively ascertain the mutation status for TSGs. Although occasionally truncating mutations can be detected in RNAseq, we deem a whole-genome sequencing cohort would better serve the goal of comprehensively and unbiasedly studying etiology. In fact, we are currently drafting a manuscript on the signatures of rearrangements (SVs) using whole genome sequencing (WGS) in >1,500 pediatric cancer patients. With this consideration, we decided to NOT include tumor suppressor gene in this study, which is designed to focused on oncogenic fusions like *BCR-ABL1*. However, in response to Reviewer #2's request, we analyzed highly frequent *CDKN2A* and *NBAS* truncating fusions in section **SN 12**.

8.b) Oncogenic fusions in the category of promoter/enhancer hijacking. In this category, a rearrangement can bring a strong promoter/enhancer to an otherwise silenced proto-oncogene and lead to its aberrant expression. When the novel promoter is far, the proto-oncogene may start its transcription from its own transcription start site, thereby leaving no split reads or discordant read pairs in RNAseq data for bioinformatic detection (**Fig. SN2a**). On the other hand,

the transcripts may contain part of the novel promoter sequences when the novel promoter is closer (**Fig. SN2b**). Moreover, it is also possible that a point mutation in the native promoter can convert it to a strong active promoter (such as *TAL1* in pediatric T-ALL¹⁵) and lead to aberrant expression (**Fig. SN2c**)— biologically this scenario is not promoter/enhancer *per se*. Clearly, without DNA (preferentially whole genome) sequencing data, scenarios a) and c) cannot be resolved by transcriptome sequencing and, a forced analysis will result in biased conclusions that does not meet the scientific rigor requirement of this journal. Instead, such patterns are best studied in our ongoing project on the signatures of rearrangements (SVs) using whole genome sequencing in >1,500 pediatric cancer patients. Nevertheless, we provided results on known oncogenic fusions (*CRLF2*, *DUX4*, *EPOR*, *BCL11B*, **Table SN2a-e**) in promoter/enhancer-hijacking category to address Reviewer #2's request though we did not perform systematic discovery.

Fig. SN2. Promoter/enhancer-hijacking. In hypothetical scenario (a), the blue chromosome (and a strong enhancer/promoter highlighted by blue oval) was brought proximity to the orange proto-oncogene and lead to its aberrant expression. Transcription only involves the proto-oncogene due to the space between the blue promoter and orange gene. In (b), the blue promoter contacts the transcription start site of orange gene, so that the transcription involves both the proto-oncogene and a small part of the blue chromosome. In (c), a point mutation in the promoter region may convert it to a strong active promoter to initiate the proto-oncogene without a fusion event (such as *TAL1* enhancer mutation in Mansour et al (2014)¹⁵). Split reads or discordant read pairs are expected for scenario (b) but not scenario (a) or (c). DNA (preferentially whole genome) sequencing are needed to ascertain the fusion status. Nevertheless, the aberrant high expression of such proto-oncogene typically can help ascertain the tumor subtype.

8.c) Conventional oncogenic fusions that generate chimeric proteins. As shown in our Fig. 1a, oncogenic fusions that generate chimeric proteins are obligated to have split read or discordant read pair signals in RNAseq data, either polyT protocol or total RNA protocol. It is this exact category that our large cohort of 5,190 RNAseq datasets can be used to generate scientifically rigor discoveries.

2.2a Another major shortcoming of the manuscript is that it continues to be heavily biased towards fusions selected by the authors as relevant to pediatric cancer based on having a gene partner or partners that have previously been defined as oncogenes – now made explicitly clear by the recent revision.

[Response] We appreciate this critique from Reviewer #2's. It significantly improved our work. Indeed, manual review of raw predictions from software tools (with good sensitivity) has been our routine

practice, even though there are thousands of samples. With many years of experience in such tasks, we can analyze many samples quickly with a high accuracy. However, we do realize that this practice is difficult to be transferred to the research community in general, thus compromising its reproducibility. Also, human error can happen during the manual review process. Therefore, in this revision we converted our many years of manual review experience into a highly efficient and effective computational pipeline, as detailed in **Method** section of main text (and Supplementary Notes). This method takes advantage of existing knowledge on some known oncogenic fusions to develop effective filters to greatly reduce the number of predictions from >5.7 million to 7,769 so that unbiased manual inspection is feasible for most researchers. Of note, in our first submission we detected fusions in 1,988 patients, while in this new method we detected fusions in 2,005 patients. The gain of 17 additional detections (0.8%) indicates not only the robustness of our previous manual review but also the power of the novel method.

Because of the large number (5,781,630) of predicted fusions from these four methods, manual inspection is impractical, if not impossible. We therefore developed a novel workflow (see detailed design principles and algorithmic descriptions in **Supplementary Notes**) using majority voting (a prediction is considered to have k votes if it is detected by k methods) to enable the effective and efficient detection of oncogenic fusions from 5,190 patients. This workflow has eight critical considerations. First, mutual exclusivity among oncogenic fusions. Using 63 well-known oncogenic fusions (Supplementary Note 10, **Supplementary Table 19**), we determined 1,743 patients to harbor ≥ 1 of these fusions detected by ≥ 2 detection methods. Interestingly, only 4 (0.23%) patient samples harbor ≥ 2 fusions, which indicates that each patient tumor typically harbors no more than 1 oncogenic fusions. Second, harmonizing coordinate differences among methods. By comparing predictions between methods, we determined that different methods can have ~ 10 nt differences in their predicted fusion coordinates (**Supplementary Notes 11a**). Third, multiple calls of the same oncogenic fusion pairs. As demonstrated in this work, intronic versioning is observed in many fusion pairs. Clearly, each intronic version corresponds to a unique prediction. Depending on the signal strength (number of supporting reads), some methods may “miss” a low abundance version (thus a low vote count) although other high abundance versions are commonly detected. By focusing on the high abundance versions, we determined that >93% of oncogenic fusion versions have 3+ votes. Fourth, although there are >5.7 million predictions, only 0.3% (16,348) predictions have 3+ votes. This data renders manual inspection possible by focusing on high-vote predictions, with a false negative rate <7%. Fifth, with the mutual exclusivity rule, we can establish blacklist by collecting all predictions from 1,743 fusion-positive patients that do not match any known fusions. Such blacklist allows us to further filter common artefacts such as readthrough¹⁶, with a negligible false negative rate of 7%. Sixth, manual review of the remaining 7,769 predictions with 3+ votes. As detailed in **Supplementary Notes**, priority was given to in-frame predictions with $n \geq 2$ recurrence in our cohort, or with literature support, such as previously published childhood cancer studies (pan-cancer analyses^{6,7} and a number of disease-focused analyses including ependymoma⁵¹, Ewing sarcoma^{52,53}, rhabdomyosarcoma⁵⁴, low grade glioma⁵⁵, high grade glioma⁵⁶, T-cell acute lymphoblastic lymphoma⁵⁷, acute myeloid leukemia⁵⁸, as well as a recent clinical genomics report⁴³ and other literature^{6-8,28,31,37,39,40,44,45,50-56,58-60}). Quality indication from these methods were also considered. This review takes about 5 hours for an experience scientist. Seventh, upon manual review, we used the comprehensive list of oncogenic fusions to run a second round of systematic detection (rescue) as in the first consideration (those detected by 2 methods are included). This step allows us to minimize the impact of hard threshold of “ ≥ 3 ”

votes". Upon this comprehensive identification, we re-analyzed and further confirmed the mutual exclusivity rule: only 7 (0.35%) of 2,005 chimeric fusion-positive patients have 2+ oncogenic fusions. Eighth, determining the functional orientation of oncogenic fusions. When a patient tumor has balanced translocations, there might be reciprocal fusions, such as *ETV6-RUNX1* and *RUNX1-ETV6*. By collecting the number of patient samples with each orientation detected, we determined that all 52 fusions detected in 4+ patients have the clinically recognized orientation supported with higher frequency than the other orientation. Collectively, we detected 272 unique oncogenic fusion gene pairs that can generate chimeric proteins (**Table SN3c**). We also reported promoter-hijacking fusions for 12 genes (**Table SN2e**) in this cohort.

2.2b The authors continue to not make available as supplementary material the complete list of fusion candidates identified by running the various fusion-finding algorithms, but instead provide their preselected list of fusions they deem as relevant for this analysis. Hence, we continue to find no actual reporting of prevalence of pediatric cancer relevant fusions mentioned above as we would expect of a comprehensive survey.

[Response] We regret that Reviewer #2 failed to find the complete list of fusions that is provided in our Revision (screenshot from revised manuscript and rebuttal letter shown below). In this revision, the zenodo link has been updated to <https://doi.org/10.5281/zenodo.7510612>.

531 bams (typically ~1 day) and we were able to finish the re-run of all four fusion detectors on all full
532 bams/fastqs of our full cohort in less than 3 months for revision. All raw output from the four fusion
533 detectors were deposited in [zenodo \(https://doi.org/10.5281/zenodo.7033077\)](https://doi.org/10.5281/zenodo.7033077).

2.3a Often, the number of fusions analyzed in any specific analysis total few in number, and this is due to the need for sufficient patient prevalence as part of the analysis. This is understandable, but also unfortunate, because it reduces the impact of the work. The abstract is written in a grandiose style with little to no specific details reported. It is only when reading the main text that one discovers that only in-frame coding fusions were examined, up to 20 fusions were explored for impacts on splicing, and a single fusion was found to have clinically relevant prognostics. It would be best to include specifics in the abstract. Instead of 'a subset of oncogenic fusions', indicate which ones and/or how many.

[Response] We respectfully disagree with the comment that "the number ... total few in number." It is well known that prevalence of cancer driver genes has a "long tail" nature in almost all human cancers such as prostate cancer (<https://pubmed.ncbi.nlm.nih.gov/29610475/>), where only a few genes drive the development of tumors in many patients, and many more cancer genes only affect a few patients. Similarly, in this work, among the 272 gene pairs identified, only a few fusion pairs have sufficient number of samples for statistical analysis. The rare gene pairs can only be studied by using specifically designed patient cohorts. We have revised the abstract as following.

Oncogenic fusions formed through chromosomal rearrangements are hallmarks of childhood cancer that define cancer subtype, predict outcome, persist through treatment, and can be ideal therapeutic targets. However, mechanistic understanding of the etiology of oncogenic fusions remains elusive. Here we report a comprehensive detection of 272 oncogenic fusion gene pairs by using tumor transcriptome sequencing data from 5,190 childhood cancer patients. We identified diverse factors, including translation frame, protein domain, splicing, and gene length, that shape the formation of oncogenic fusions. Our mathematical modeling revealed a strong

link between differential selection pressure and clinical outcome in *CBFB-MYH11*. We discovered 4 oncogenic fusions, including *RUNX1-RUNX1T1*, *TCF3-PBX1*, *CBFA2T3-GLIS2*, and *KMT2A-AFDN*, with promoter-hijacking-like features that may offer novel strategies for therapeutic targeting. We uncovered extensive alternative splicing in oncogenic fusions including *KMT2A-MLLT3*, *KMT2A-MLLT10*, *C11orf95-RELA*, *NUP98-NSD1*, *KMT2A-AFDN* and *ETV6-RUNX1*. Strikingly, we discovered novel splice sites in 18 oncogenic fusion gene pairs and demonstrated that such splice sites confer novel therapeutic vulnerability for etiology-based genome editing. Our study reveals general principles on the etiology of oncogenic fusions in childhood cancer and suggests profound clinical implications including novel etiology-based risk stratification and genome-editing-based therapeutics.

2.3b Also, related to this last point in the main text, it isn't indicated how many other fusions were tested using hazard ratios and not found significant.

[Response] In Discussion, we added below statement as a limitation of our study:

Fourth, we detected clear selection bias in intronic versioning and established a strong link of such selection bias with clinical outcomes for *CBFB-MYH11*. This data indicated a differential oncogenicity of corresponding fusions due to the inclusion/exclusion of protein domains. However, no other fusion pairs can be studied in this work due to lack of publicly available clinical outcome data for corresponding cases for additional candidates with selection bias (*ETV6-RUNX1*, *KIAA1549-BRAF*, *NUT214-ABL1*, and *BCR-ABL1*). It would be interesting to test these candidate pairs in future dedicated studies.

2.4a The finding that patient prevalence of *KMT2A* fusion partners is well correlated with gene length is quite interesting and clearly significant. However, no specific evidence is given that other oncogenic fusions behave similarly.

[Response] We have added this analysis to two additional genes *ETV6* and *PAX5* (that have ≥ 5 fusion partner genes) that allowed statistical analysis. In section "Landscape of childhood oncogenic fusions" of main text, we added below:

By limiting the analysis to leukemia with oncogenic fusions with ≥ 5 fusion partners (*KMT2A*²⁴, *ETV6*, and *PAX5*), we obtained a statistically significant linear association for *KMT2A* either when fusions with recurrence >3 were considered (R-squared=0.82; $P=0.002$; Fig. 2i) or when fusions with recurrence >1 were considered (R-squared=0.86; $P=1.5 \times 10^{-5}$; Extended Data Fig. 2i) but not for *ETV6* and *PAX5* ($P>0.1$; Extended Data Fig. 2j-k). The above observations are also observed if only involved introns are considered (Extended Data Fig. 2l-r). Excluding *KMT2A* fusions resulted an insignificant association in leukemia ($P=0.22$; data not shown). The overall insignificant association between gene length and patient prevalence in oncogenic fusions (except *KMT2A*) indicated that additional molecular factors (such as protein domain, frame, and splicing) may play a major role in the formation or selection of oncogenic fusions, as will be demonstrated next.

We also added these data in Extended Data Fig. 2:

Extended Data Fig. 2 Impact of gene length on prevalence of oncogenic fusions. (a). DNA breakpoints can be detected from RNAseq data (see raw data in Supplementary Table 8). Shown in y-axis is the number of DNA breakpoints as a function of distance between DNA breakpoints detected from RNA and DNA sequencing data (n=43 samples with both data

available) shown in x-axis (in base pairs). Note that each sample (oncogenic fusion) has two DNA breakpoints (totaling 86 breakpoints), one from N' gene and another from C' gene. Majority of detections from RNA data perfectly match that detected from DNA data. DNA breakpoints are uniformly distributed in *ETV6-RUNX1* (b), *NUP98-NSD1* (c), and demonstrated clustered distribution in promoter region of *RUNX1T1* gene for oncogenic fusion *RUNX1-RUNX1T1* (d), in addition to those shown in Fig. 2j-i. Also shown are linear regression between gene length and patient prevalence for fusions observed in leukemia (e), brain tumor (f), solid tumor (g) and all childhood cancer (h). In panels b-d, Bonferroni correction (Q values) is applied to the raw P values to account for multiple testing. For *KMT2A* (i), *ETV6* (j), and *PAX5* (k), there are sufficient number (≥ 5) of different fusion partner genes to study the association between patient prevalence and length of the partner genes. Only *KMT2A* partner genes resulted a significant association with R-squared of 0.86. Similar analysis was performed by only using the length of involved introns for leukemia (l), brain (m), solid (n), or all cancer types (o), as well as for partner genes for *KMT2A* (p), *ETV6* (q), and *PAX5* (r). Only *KMT2A* partner genes demonstrate significant association between intron length and patient prevalence (R-squared of 0.84). Removing *KMT2A* fusions from panels e and l resulted in similar insignificant P values (data not shown).

2.4b There may also be *KMT2A* fusion partners identified as candidate fusions in the initial survey but not selected by the authors for this particular analysis, but this is not clear given that the full set of initial fusion candidates was not reported.

[Response] We have added in section “**Landscape of childhood oncogenic fusions**” to highlight systematic analysis of *ETV6*, *PAX5*, and *KMT2A*:

We analyzed oncogenic fusions for leukemia, solid tumor, and brain tumor, separately. A marginally significant linear association (**Fig. 2h; Extended Data Fig. 2e**; R-squared=0.12; P=0.058) was observed between patient prevalence and total length of the involved gene pairs, and no significance were observed in brain and solid tumors (**Extended Data Fig. 2f-g**). By limiting the analysis to leukemia with oncogenic fusions with ≥ 5 fusion partners (*KMT2A*²⁴, *ETV6*, and *PAX5*), we obtained a statistically significant linear association for *KMT2A* either when fusions with recurrence >3 were considered (R-squared=0.82; P=0.002; **Fig. 2i**) or when fusions with recurrence >1 were considered (R-squared=0.86; P=1.5 $\times 10^{-5}$; **Extended Data Fig. 2i**) but not for *ETV6* and *PAX5* (P >0.1 ; **Extended Data Fig. 2j-k**). The above observations are also observed if only involved introns are considered (**Extended Data Fig. 2l-r**). Excluding *KMT2A* fusions resulted an insignificant association in leukemia (P=0.22; data not shown). The overall insignificant association between gene length and patient prevalence in oncogenic fusions (except *KMT2A*) indicated that additional molecular factors (such as protein domain, frame, and splicing) may play a major role in the formation or selection of oncogenic fusions, as will be demonstrated next.

2.4c As mentioned in my earlier review, the correlation between gene length in leukemia fusions (p-value = 0.05) is tenuous, and there was no significant association found for the other tumor types analyzed. I suspect the *KMT2A* fusions contributed greatly towards the marginally significant finding with a p ≤ 0.05 threshold. Removing *KMT2A* fusions from that analysis would possibly make the finding non-significant.

[Response] We have added following analysis in section “**Landscape of childhood oncogenic fusions**” to highlight the contribution of *KMT2A* in leukemia:

We analyzed oncogenic fusions for leukemia, solid tumor, and brain tumor, separately. A marginally significant linear association (**Fig. 2h; Extended Data Fig. 2e**; R-squared=0.12; P=0.058) was observed between patient prevalence and total length of the involved gene pairs, and no significance were observed in brain and solid tumors (**Extended Data Fig. 2f-g**). By limiting the analysis to leukemia with oncogenic fusions with ≥ 5 fusion partners (*KMT2A*²⁴, *ETV6*, and *PAX5*), we obtained a statistically significant linear association for *KMT2A* either when fusions with recurrence >3 were considered (R-squared=0.82; P=0.002; **Fig. 2i**) or when fusions with recurrence >1 were considered (R-squared=0.86; P=1.5 $\times 10^{-5}$; **Extended Data Fig. 2i**) but not for *ETV6* and *PAX5* (P >0.1 ; **Extended Data Fig. 2j-k**). The above observations are also observed if only involved introns are considered (**Extended Data Fig. 2l-r**). Excluding *KMT2A* fusions resulted an insignificant association in leukemia (P=0.22; data not shown). The overall insignificant association between gene length and patient prevalence in oncogenic fusions (except *KMT2A*) indicated that additional molecular factors (such as protein domain, frame, and splicing) may play a major role in the formation or selection of oncogenic fusions, as will be demonstrated next.

2.4c If the authors would make the data for these analyses available as supplementary materials, it would make it possible for others to more easily directly explore them.

[Response] We added Supplementary Tables (**Table ST21** for raw data of **Fig. 2** and **Extended Data Fig. 2** and **Table ST22** for **Fig. 5b**) to detail such intermediate results for readers to directly explore. This is no

need of such data for other figures/panels.

2.5a Reporting that DNA fusion breakpoints are found mostly to occur uniformly in relevant introns makes sense. Reporting that RNA-based fusion breakpoints are generally found within 5 bases of a DNA breakpoint does not. While I haven't examined the authors references for this finding, based on my own experience and based on basic splicing biology coupled with uniform DNA breakpoint data, it just simply does not make sense whatsoever. Uniform DNA breakpoints across long intronic regions cannot be within 5 bases on average from a splice junction. This is clear from the authors own figure 5a. I could be misinterpreting something here, because this is too bizarre to have a disagreement over.

[Response] We regret that Reviewer #2 misinterpreted the data, where the result on "5 bases" is a sanity check to ensure our DNA breakpoint detection from RNAseq data is reliable. We have added a section "Detecting DNA breakpoints from RNAseq data" in **Supplementary Note 9** as below to detail the rationale of our analysis.

In this work, we attempted to detect DNA breakpoints (here termed d-event) from RNAseq data to interrogate the uniformity of DNA breakpoints in relative intronic regions. As shown in the model of **Fig. SN 5**, although RNA splicing breakpoints (here termed m-event; commonly referred to as "fusion") are guaranteed to be observed in mRNA species that have undergone splicing, d-events are only occasionally observed in total RNA sequencing but NOT in poly(T)-based mRNA sequencing. In this work, we compared our d-event detections in RNA sequencing datasets against the ground truth d-events defined in DNA (whole genome) sequencing datasets and demonstrated that 91% of our detections are within 5-bp of ground truth. This accuracy enables us to reach reliable conclusion that DNA breakpoints are roughly uniformly distributed in relative introns of oncogenic fusions.

Fig. SN 5. Detecting DNA and RNA breakpoints from next generation sequencing data. (a) During gene expression, genetic information encoded in wildtype human chromosomes (gray, thin line indicates intron/intergenic region, thick boxes indicate exons) are first transcribed into pre-spliced transcripts (pre-mRNA; black), which is in turn spliced (to remove introns and retain exons) to generate mature RNA species (mRNA, green). (b) This Central Dogma also applies cancer genome (here blue and orange indicate two different chromosomal regions joined together by rearrangement). In this model, an intronic rearrangement (here termed as "d-event" to stress it is observed from DNA) happened between the two involved genes shown in blue and orange in the cancer genome. This d-event is observable in pre-mRNA; however, this d-event typically is not observed in mRNA because the intronic regions are spliced out. On

the other hand, in mRNA, the rearrangement is manifested as splicing junctions (here termed “m-event” to indicate it is observed in mRNA; commonly referred to as “fusion” events in bioinformatics) which typically are not directly observed in the DNA of cancer genome although biological inference is possible (with alternative splicing being the confounding factor in consideration). (c) In next generation sequencing, we can perform DNA sequencing (such as whole genome sequencing (WGS), targeted capture or Sanger sequencing, etc.) and RNA sequencing. Earlier RNA sequencing practices typically utilize poly(T) protocol, which can only interrogate mRNA species and therefore can NOT be used to detect d-events. Recent RNA sequencing practices typically utilize total-RNA protocol, which can simultaneously interrogate mRNA species and pre-mRNA species and enables simultaneous detection of d-event (if the total RNA contains sufficient pre-mRNA species and therefore is not guaranteed) and m-event.

2.5b The data for Figure 2a should be made available, and it should be examined to see if there are specific fusions that are biasing results, such as RUNX1/T1 where breakpoints are not found as uniform and the fusion is one of the most prevalent.

[Response] We believe this reviewer was referring our **Extended Data Fig. 2a** (but not Figure 2a) on the concordance between predicted DNA breakpoints from RNAseq data and those from DNAseq data. We have provided the raw data in **Supplementary Table 8** and we have updated the **Extended Data Fig. 2a** length as following:

Extended Data Fig. 2 Impact of gene length on prevalence of oncogenic fusions. (a). DNA breakpoints can be detected from RNAseq data (see raw data in **Supplementary Table 8**). Shown in y-axis is the number of DNA breakpoints as a function of distance between DNA breakpoints detected from RNA and DNA sequencing data (n=43 samples with both data available) shown in x-axis (in base pairs). Note that each sample (oncogenic fusion) has two DNA breakpoints (totaling 86 breakpoints), one from N' gene and another from C' gene. Majority of detections from RNA data perfectly match that detected from DNA data. DNA breakpoints are uniformly distributed in **ETV6-RUNX1** (b), **NUP98-NSD1** (c), and demonstrated clustered distribution in promoter region of **RUNX1T1** gene for oncogenic fusion **RUNX1-RUNX1T1** (d), in addition to those shown in **Fig. 2j-i**. Also shown are linear regression between gene length and patient prevalence for fusions observed in leukemia (e), brain tumor (f), solid tumor (g) and all childhood cancer (h). In panels **b-d**, Bonferroni correction (Q values) is applied to the raw P values to account for multiple testing. For **KMT2A** (i), **ETV6** (j), and **PAX5** (k), there are sufficient number (≥ 5) of different fusion partner genes to study the association between patient prevalence and length of the partner genes. Only **KMT2A** partner genes resulted a significant association with R-squared of 0.86. Similar analysis were performed by only using the length of involved introns for leukemia (l), brain (m), solid (n), or all cancer types (o), as well as for partner genes for **KMT2A** (p), **ETV6** (q), and **PAX5** (r). Only **KMT2A** partner genes demonstrate significant association between intron length and patient prevalence (R-squared of 0.84). Removing **KMT2A** fusions from panels e and l resulted in similar insignificant P values (data not shown).

2.6a The selection bias measurement has a null hypothesis stated as having patient prevalence proportional to intronic length. This seems intuitive, but I see no evidence presented for this null hypothesis to be valid. The authors provide evidence that DNA breakpoints are often found uniformly distributed at sites of fusions, and there is unconvincing evidence that (except for KMT2A fusions) that gene length is associated with patient prevalence, and so I suspect the authors are attempting to make a jump here in using these other limited data to support this null hypothesis.

[Response] We apologize for the confusion. We modified the justification in section “**Selection bias in intronic versioning**” to stress that our model only relies on the uniform distribution of DNA breakpoints (but not gene length).

Because intronic versioning can cause amino acid differences in the fusion protein which may in turn lead to potential functional difference, we hypothesized that intronic versioning could confer differential fitness to the host cells in some oncogenic fusions (**Supplementary Note 13**). To measure the effect size of potential selection bias between two intronic versions, we proposed a relative selection bias score (RSB; **Methods**) based on the observation that DNA breakpoints are distributed in relevant introns in a near-uniform fashion in 95% fusions (**Fig. 2j-l**; **Extended Data Fig. 2b-d**; **Supplementary Table 8**). In this model, the patient prevalence of DNA breakpoints falling in an intron should be proportional to its length if the resultant protein versions are functionally equivalent (i.e., confers the same positive selection pressure). The statistical significance of selection bias is measured by comparing (using a Chi-squared test) the observed patient prevalence in all intronic versions and corresponding expected patient prevalence under a null hypothesis that patient prevalence is proportional to intronic length. When the involved exon (**Fig. 5a**, star) encodes functionally important protein domain and thus lead to higher positive selection pressure, its corresponding intron will have disproportionately high patient prevalence.

Further, we added a section in **Supplementary Note 13** to illustrate the rationale of our analysis of selection bias:

13. Model of selection bias

Our study indicated that many fusion gene pairs have intronic versioning, where slightly different oncoproteins can be generated in different patient tumors. A natural question is whether such difference may carry biological significance such as oncogenicity. For this, we performed a theoretical analysis (Fig. SN5). Here, we introduced the concept of DNA rearrangement events before and after selection. First, DNA rearrangement events can happen either in a random fashion (Fig. SN5a) or in a non-random fashion (Fig. SN5b). Second, random DNA rearrangement events may be subjected to unbiased (Fig. SN5c) or biased (Fig. SN5d) selection, which will generate unbiased or biased patient prevalence among different intronic versions, respectively. Clearly, biased DNA rearrangement events (Fig. SN5b,e) can confound the statistical analysis and we therefore dropped *TCF3-PBX1* (Fig. 2I) and *RUNX1-RUNX1T1* (Extended Data Fig. 2d) from this analysis. Fusions with alternative splicing can also confound this analysis and are dropped (Fig. 5b).

Fig. SN5. Selection bias. DNA rearrangement events can happen in a random fashion (a; such as Fig. 2j-k) or in a non-random fashion due to an (unknown, such as the *TCF3-PBX1* example in Fig. 2I) molecular mechanism (b). In either scenario, two oncoproteins (A and B) are generated, regardless of the exact DNA breakpoints, due to splicing. Next, selection come to play. First, the oncogenicity of proteins A and B might be same. In this case, selection would be neutral between A and B versions, and the original un-biased rearrangements are mirrored in un-biased patient prevalence (c). Second, the oncogenicity of

proteins A and B might be different. In this case, selection bias will be reflected in biased patient prevalence (where version A has less frequency than expected; d). Clearly, selection bias cannot be analyzed when DNA rearrangement is biased (panels e and b) and when alternative splicing is observed (Fig. 5b).

2.6b Even so, based on my earlier recommendation, the authors included results based on not normalizing for intron length, and I found these results to be quite interesting, and rather than contributing false positives or false negatives, instead highlighting those fusions where the statistics are more likely to be robust indicators of relevant findings. Ultimately, while length normalization is intuitive, the authors have no compelling evidence that it is required, even though attempts are made to do so.

[Response] Please see our rebuttal in section 2.6c. Normalization is required under this rationale. We appreciate this comment that significantly improved our presentation.

Reviewer #3 (Remarks to the Author):

Thank you for the thoughtful edits in response to all reviewers. The authors have sufficiently responded to my comments.

REVIEWERS' COMMENTS

Reviewer #2 (Remarks to the Author):

Thank you for the detailed response to my critiques. I appreciate the very thorough responses and the revisions to the manuscript, which address my earlier concerns. Very well done! and I look forward to seeing the revised manuscript in press.

Point-by-point response to reviewer comments

(Round 1)

Reviewer #1 (Remarks to the Author): Expert in paediatric cancer genomics

This manuscript is focused on a meta-data analysis of numerous publicly available pediatric cancer transcriptome sequencing datasets derived from an impressive number of 5,190 patients and aiming at elucidation of the etiology of oncogenic fusions. While the manuscript is well written (although not really telling a novel story) and the data science approaches seem to be solid and state-of-the-art, the results do not add any substantial novel insight into the biology or mechanistics of oncogenic fusions, but rather confirm previous knowledge in a large patient number.

[Response] We appreciate the general positive comment from reviewer #1. It appears that this reviewer's main concern is "lack of novel insights", for which we respectfully disagree.

Indeed, thanks to the many large scale cancer genomic profiling works, numerous oncogenic fusions have been defined for childhood cancers in past decade. It has been noted that the prevalence of these fusions varies widely, but **no systematic study on this variation** is found in literature. It is exactly this void that our study is trying to fill in. To stress this significance, we added in section "Landscape of childhood oncogenic fusions": "Although it has been noted that the prevalence of oncogenic fusions varies widely^{1,2}, no systematic studies on potential mechanisms are found in literature."

Further, we added in Discussion to highlight the conceptual advancement of our analytical framework: "This work also highlights a novel analytical framework to investigate oncogenic fusions. By focusing on the large variability in patient prevalence of oncogenic fusions, we first established the null hypothesis that gene length or random chance can be a significant predictor in certain fusions. In fusion where such null hypothesis does not hold, we were able to find interesting patterns that reflect interesting biology (local DNA properties in *TCF3*) and clinical implications (*CBFB-MYH11*). Although majority of fusions utilize existing exon/intron structure by connecting them to form chimeric proteins, novel cryptic exons are created in ~3% fusion positive patients. The requirement of proper splicing sites and translation frames in these cases may explain their relative low prevalence. We expect this generic workflow to shed more biological and clinical insights when more samples are available for rare fusions."

1. The conclusion, "Together, these data indicate that random chance (or gene/intron length), and less frequently, local DNA properties can influence the formation of oncogenic fusions" is an interpretation that can easily be made based on the existing knowledge/data in the literature.

[Response] To provide a better context, we have added the rationale to study the factors contributing prevalence of oncogenic fusions: "Although it has been noted that the prevalence of oncogenic fusions varies widely^{1,2}, no systematic studies on potential mechanisms are found in literature."

We next stressed that the association between gene length and patient prevalence was next used as a null hypothesis to detect additional biological/clinical insights: "By assuming the association between gene length and patient prevalence as our null hypothesis, our extended analysis to brain tumor and solid tumor did not yield statistical significance (**Extended Data Fig. 2f-g**). This data may either reflect the diverse subtypes and corresponding smaller cohort sizes among brain tumor and solid tumor or

indicate additional factors influencing the etiology of oncogenic fusions, which will be addressed in following sections.”

Further, we added in Discussion to highlight the conceptual advancement of our analytical framework: “This work also highlights a novel analytical framework to investigate oncogenic fusions. By focusing on the large variability in patient prevalence of oncogenic fusions, we first established the null hypothesis that gene length or random chance can be a significant predictor in certain fusions. In fusion where such null hypothesis does not hold, we were able to find interesting patterns that reflect interesting biology (local DNA properties in *TCF3*) and clinical implications (*CBFB-MYH11*). Although majority of fusions utilize existing exon/intron structure by connecting them to form chimeric proteins, novel cryptic exons are created in ~3% fusion positive patients. The requirement of proper splicing sites and translation frames in these cases may explain their relative low prevalence. We expect this generic workflow to shed more biological and clinical insights when more samples are available for rare fusions.”

2. The diagnostic and clinical relevance of the data seems to be overestimated and it is unclear, how the necessity for the CRISPR-Cas9 experiments performed is extracted from the data.

[Response] We regret that the reviewer did not see the diagnostic and clinical relevance of our study on one of the most important type of somatic alterations—oncogenic fusions, as stressed in the initial submission: “In these cases, subtype-defining oncogenic fusions (e.g., BCR-ABL1 in Philadelphia chromosome positive patients^{3,4}) typically persist through the lifetime of a tumor^{5,6} and can serve as stable biomarkers for curative outcomes. Moreover, successes in targeted inhibition of oncogenic fusions (e.g., imatinib for BCR-ABL1⁷) has inspired the notion of “**oncogene addiction**”⁸ that posits on the **therapeutic potential of targeting oncogenic fusions.**”

With our systematic investigation on the prevalence, it naturally leads to a logic on where the fusion happens between exons with incompatible translation frames, and therefore the TCF3-HLF findings with “spectacular demonstration” of novel vulnerabilities (Reviewer #2). This logic is found in below sentence of original submission: “Indeed, close examination indicated that exon 16 of *TCF3* and exon 4 of *HLF* have incompatible translation frames (**Extended Data Fig. 5e**). Therefore, the neo splice sites and corresponding cryptic exons are created by the host cancer cell to compensate the translation problem.” This is the critical data that leads to our CRISPR editing strategy.

We added a statement in section “CRISPR targeting of neo splicing” to stress the clinical relevance of studying TCF3-HLF: “B-cell acute lymphoblastic leukemia (B-ALL) patients with TCF3-HLF fusion are currently considered incurable⁹.”

3. Previously well-known phenomena (i.e. oncogenic fusion is leading to overexpression of an involved gene via the promotor of the other gene) are given a new name (“promotor-hijacking”) without being a novel finding.

[Response] We apologize for this confusion. To clarify, we added “Because these fusions generate novel chimeric proteins, this group of promoter hijacking-like fusions are fundamentally different from conventional promoter hijacking fusions (e.g., IGH-CRLF2/EPOR/DUX4) where no chimeric proteins are involved.”

4. The observations regarding splicing are also expected. Why should there be no splicing of fusion genes as for any other gene? This is no surprise...

[Response] Logically, a fusion oncogene may either have alternative splicing or not, therefore “no surprise”. However, our study provided a formal characterization on the exact list of fusions subject to alternative splicing. We modified our text as “Together, our data indicated a clear involvement of alternative splicing in **certain, but not all**, oncogenic fusions, although such regulation is not specific to tumor and therefore is likely an intrinsic property of the host gene.”

Further, the lack of alternative splicing played a critical role in the success of our CRISPR-based targeting on TCF3-HLF fusion, which is now clarified as “**Indeed, RNA sequencing of CRISPR-edited HAL-01 cells confirmed lack of alternative splicing (Supplementary Table 13) in TCF3 (when exon 16 is used) so that the host cancer cells are completely dependent on cryptic exons via neo-splicing, which is in clear contrast with the weak E14-E4 alternative splicing in B-ALL with E15-E4 version of TCF3-HLF (SJALL018389 in Supplementary Table 7 and Supplementary Table 13).**”

As we stated, exons subject to alternative splicing may be dispensable for the tumor, and therefore should be avoided for drug targeting. This is an important concept we can draw for fusions with alternative splicing.

5. Sometimes single observations in the data (i.e. KMT2A fusion genes are associated with gene length/prevalence) are generalized without systematic validation.

[Response] Although we hope there is a single variable that can explain everything on the widely variable prevalence of oncogenic fusions, our study indicated it is unlikely. For example, CFBF-MYH11’s prevalence is significantly associated with protein domain usage. Likewise, in TCF3 fusions the local DNA property matters.

We updated the writing to stress that the association between gene length and patient prevalence was used as a null hypothesis to detect additional biological/clinical insights: “**By assuming the association between gene length and patient prevalence as our null hypothesis, our extended analysis to brain tumor and solid tumor did not yield statistical significance (Extended Data Fig. 2f-g). This data may either reflect the diverse subtypes and corresponding smaller cohort sizes among brain tumor and solid tumor or indicate additional factors influencing the etiology of oncogenic fusions, which will be addressed** in following sections.”

We also added in section “Discussion”: “**Notably, we only discovered KMT2A fusions with prevalence well predicted by gene length. For other fusions that lack the association between prevalence and gene length, our data strongly indicate alternative mechanisms, such as protein domain in CFBF-MYH11 (Fig. 5b) or the clustered DNA breakpoints in TCF3 fusions (Fig. 2I), are at play. It is warranted to validate our findings for specific fusions with larger sample sizes.**”

6. Some descriptions are misleading, i.e.: „Surprisingly, we obtained a statistically significant linear association (R-squared=0.82; P=0.002; Fig. 2i) between gene length and patient prevalence“. This is not surprising, but rather to be expected.

[Response] we updated “Surprisingly” to “Indeed”.

Reviewer #2 (Remarks to the Author): Expert in gene fusion detection and computational cancer genomics

The manuscript "Etiology of oncogenic fusions in 5,190 childhood cancers and its clinical and therapeutic implication" by Liu et al. describes an effort to explore the characteristics of oncogenic fusions in childhood cancers including primarily leukemia and brain cancer. The authors examine fusion breakpoint positions, the expression of fusion partner genes in fusion-containing, fusion-lacking, and normal samples, and features contributing to alternatively spliced fusions, all with a focus on relevance to potential clinical applications in personalized medicine. In a spectacular demonstration of using CRISPR genome editing to target a particularly vulnerable type of oncogenic fusion involving "neo splicing" in cell lines, the authors clearly showed the essential nature of the neo splicing for cancer cell line viability, and addressed the complexity of targeting such fusions when more complex alternative splicing exists. The manuscript is very well written, quite comprehensive, and creative - I particularly like the 'circuit' style fusion figures. I appreciate that the authors have packaged up code along with data and made it all readily available. My recommendations and critiques are as follows:

[Response] We truly appreciate Reviewer #2's thorough review and insightful comments that significantly improved our presentation and the science.

Major

- The fusions studied in this work were restricted to oncogenic fusions relevant to childhood cancers. To identify the fusions, the authors leveraged four different fusion transcript prediction methods. From these complete lists of fusion predictions (not currently provided in supplemental materials, as far as I could tell), the subset relevant to childhood cancers was extracted and reported (provided in Supp. Table 3). The authors provided Supp. Figure 1 to show the fusions defined as childhood oncogenic fusions, which totals to just over 60 examples. But, there are over 260 fusion types (unique gene pairings) for the reported fusions in ST3. It isn't entirely clear how the full set of predicted fusions derived from the four prediction tools was distilled to this set of 260 that were analyzed. More transparency is needed here, ideally providing the full set of predictions in supplementary materials and defining the reasons for which the subset analyzed here were selected.

[Response] Thanks for the suggestion, we updated the “Fusion detection” section as follows: “Oncogenic fusions were detected by using state-of-the-art methods reported to have superior performance^{10,11}, including Cicero¹¹, Arriba¹², STAR-fusion¹³, and FusionCatcher¹⁴. Cicero was run on bam files aligned with STAR v2.5.3a while Arriba, STAR-Fusion, and FusionCatcher were run on fastq files. To ensure clinical relevance of detected oncogenic fusions, we curated a list of 315 driver genes (which included 142 and 77 significantly mutated genes from two non-overlapping pan-childhood cancer analyses^{15,16} and a number of disease-focused analyses including ependymoma¹⁷, Ewing sarcoma^{18,19}, rhabdomyosarcoma²⁰, low grade glioma²¹, high grade glioma²², T-cell acute lymphoblastic lymphoma²³, acute myeloid leukemia²⁴, as well as a recent clinical genomics report²⁵) for childhood cancers (**Supplementary Table 18**) reported in literature^{5,9,15-22,24,26-34} and used this gene list to extract candidate fusions predicted by these tools for manual review. This procedure resulted in 2,004 samples with 257

oncogenic fusion pairs (**Supplementary Table 19**). Upon manual review on remaining predictions, an additional 5 samples (**Supplementary Table 19**) were assigned as low confidence fusions in an ad hoc manner (for example, a rhabdomyosarcoma with *MEGF10-NUP210* fusion and a brain tumor sample with *NRCAM-PRKCB* fusion).”

Further, we added a caption to Supplementary Figure 1 (which was intended to be a print poster for oncologists) as follows to clarify how the 60 fusion pairs were obtained from the 260 list: “Recurrent ($n \geq 3$) oncogenic fusions identified in 5,019 childhood cancers. For each gene pair, coding exons (thick boxes) are colored white (frame 0), gray (frame 1), and black (frame 2). Intronic length is indicated by numbers. Gray lines indicate theoretically in-frame fusions. Red lines indicate in-frame fusions observed in patients and its width indicate patient prevalence. RefSeq identifiers are indicated for each involving gene. Fusions involving neo-splicing or chimeric exons were not included (see paper).”

- One of the criteria for defining fusions of interest is that fusions produce an in-frame protein. Such in-frame fusion proteins represent a subset of all fusion transcripts relevant to cancer, and presumably childhood cancers. If the authors were to drop this requirement, are there other fusion transcripts (perhaps recurrent fusions) that might appear to be relevant to childhood leukemias among the initially unfiltered set?

[Response] Highly recurrent tumor suppressor genes (such as *CDKN2A* and *ATRX*) disrupted by chromosomal rearrangements can sometime be detected in RNA transcripts (but not always, such as in the case of whole 9p arm loss) and are beyond the scope of our study. We now clarified this in section “Fusion detection” as: “In some literature the terminology of “fusion” is used interchangeably with “structural alteration/rearrangement”. Because gene fusion is a much earlier concept following the discovery of *BCR-ABL1* fusion in Philadelphia chromosome positive leukemia, here we propose to use “fusion” exclusively in transcriptome (or gene expression) setting and to use “structural alteration/arrangement” in genome (or DNA) setting, although their biological meaning could be identical and can be discerned from context. Clearly not all gene fusions may carry biological functions like *BCR-ABL1* in Philadelphia chromosome leukemia. For example, chromosomal rearrangements that lead to inactivation of tumor suppressor genes (e.g., *CDKN2A*, *RB1*, *ATRX*) can be detected as out-of-frame fusion transcripts but are beyond the scope of this work. In this work, “oncogenic fusion” indicates in-frame fusions that produce oncogenic proteins like *BCR-ABL1*.”

- The *TCF3--HLF* fusion story is quite interesting. There is actually much known about *TCF3--HLF* published in the literature, and even though the paper is already quite long, it would be useful for the authors to add a few statements and reference relevant earlier work on this fusion.

[Response] We clarified the existing knowledge on *TCF3-HLF* as follows: “Consistent with previous report³⁵, in our dataset we also discovered one case SJALL018389 (Supplementary Table 7) to harbor natural in-frame fusion between exon 15 of *TCF3* and exon 4 of *HLF*. In this sample, a weak alternative splicing between exon 14 of *TCF3* and exon 4 of *HLF* is observed. Analysis of published RNAseq^{9,36} data on *TCF3-HLF* cells with E15-E4 version under various drug treatments (JQ1, A-485)³⁶ further confirmed that this fusion version is subject to a weak alternative splicing regulation (**Supplementary Table 13**). Although it has been suggested that the cryptic exons function to make up the translation frame

problem in *TCF3-HLF* by the cancer cells^{37,38}, there is no functional evidence available to date. In the next we sought to investigate the function of this cryptic exon and corresponding hypothetical neo splice sites through CRISPR-based genome editing.”

While *TCF3--HLF* as found here apparently requires newly found neo-splicing events to generate an in-frame protein, it is curious to me and wasn't clear from this manuscript that other previous studies on this fusion have found natural in-frame versions.

[Response] We appreciate this insightful comment. We have added following text to enrich our findings:

“Consistent with previous report³⁵, in our dataset we also discovered one case SJALL018389 (Supplementary Table 7) to harbor natural in-frame fusion between exon 15 of *TCF3* and exon 4 of *HLF*. In this sample, a weak alternative splicing between exon 14 of *TCF3* and exon 4 of *HLF* is observed. Analysis of published RNAseq^{9,36} data on *TCF3-HLF* cells with E15-E4 version under various drug treatments (JQ1, A-485)³⁶ further confirmed that this fusion version is subject to a weak alternative splicing regulation (**Supplementary Table 13**). Although it has been suggested that the cryptic exons function to make up the translation frame problem in *TCF3-HLF* by the cancer cells^{37,38}, there is no functional evidence available to date. In the next we sought to investigate the function of this cryptic exon and corresponding hypothetical neo splice sites through CRISPR-based genome editing.”

I was very curious about this fusion as studied in this work and so explored the data myself with the publicly available RNA-seq data for the cell line UoC-B1 as leveraged here. I found evidence of direct fusions between *TCF3* and *HLF* that would result in the frameshifted protein but not requiring any neo-splicing events, which STAR-Fusion should have reported. This additional alternative splicing, while presumably not oncogenic, could be worth commenting on, particularly in respect to the relative expression levels of it as compared to the neo-spliced version.

[Response] We decided to address this concern together with the next concern.

Similarly, I explored the HAL-01 cell line myself to directly explore the *TCF3--HLF* fusion in that sample. I didn't see in the manuscript what exact set of reads (ie. SRA accession) was used here, and so I downloaded one of many that are publicly available for this cell line (SRA: ERR3504860). Using STAR-Fusion and FusionInspector, I found evidence for more conventional in-frame fusions that did not require neosplicing, which surprised me, as these would presumably be oncogenic and not disrupted via genome editing. I encourage the authors to examine this closely and offer an explanation. I've included FusionInspector results for HAL-01 as reviewer supplemental info. The html file can be opened in a web browser, and the tsv file contains the predicted fusions.

[Response] We truly appreciate Reviewer #2's insight into this analysis. We decide to address the above two paragraph of comments together.

We initially were puzzled on how HAL-01 cell can generate natural in-frame fusions without using neo-splicing, because that is not compatible with our CRISPR data where effective killing was achieved--- which would be impossible if the cells were able to generate natural in-frame fusion.

We carefully re-analyzed the study design of PMID: 31735627 (and its earlier reference of PMID: 26214592) and figured out that the RNAseq data of SRA:ERR3504860 is from a patient 11a from PMID:26214592, where (Fig. 1b in that paper) it was clearly stated that 11a has a “Type II” (E15-E4, which is in-frame and utilizes natural exon structures) TCF3-HLF fusion. Because patient 11a’s tumor is different from HAL-01, we added below in section “Neo splicing in oncogenic fusions”: “Consistent with previous report³⁵, in our dataset we also discovered one case SJALL018389 (Supplementary Table 7) to harbor natural in-frame fusion between exon 15 of *TCF3* and exon 4 of *HLF*. In this sample, a weak alternative splicing between exon 14 of *TCF3* and exon 4 of *HLF* is observed. Analysis of published RNAseq^{9,36} data on *TCF3-HLF* cells with E15-E4 version under various drug treatments (JQ1, A-485)³⁶ further confirmed that this fusion version is subject to a weak alternative splicing regulation (Supplementary Table 13). Although it has been suggested that the cryptic exons function to make up the translation frame problem in *TCF3-HLF* by the cancer cells^{37,38}, there is no functional evidence available to date. In the next we sought to investigate the function of this cryptic exon and corresponding hypothetical neo splice sites through CRISPR-based genome editing.”

A screenshot of the new **Supplementary Table 13** is listed below:

1 **Table S13. Fusion isoforms detection in 2 types of TCF3-HLF rearrangements.** Listed are sample resource (Column A), DNA breakpoint types (Column B), sample (Column C), treatment (Column D), fusion type(s) (Column E), breakpoints information in RNA (Column F), neo splice site (Column G-H), number of reads supporting wild type genes and fusion (Column I-K), number of reads supporting neo splice sites (Column L-M) and reference isoform used for wild type TCF3 and HLF (Column N-O).

sample resource	DNA breakpoints at TCF3	sample	Treatment	fusion_type	RNA breakpoint (hg19)	No. wildtype reads (W gene)	No. wildtype reads (C gene)	No. fusion reads	refseqA	refseqB
PDX_11a_m1660_DMSO	intron 15	ERR3504859	DMSO	In frame, conventional; E14-E14	TCF3.chr19.1619779-.HLF.chr17.53398025+	571	0	34	NM_003200	NM_002126
PDX_11a_m1660_DMSO	intron 15	ERR3504859	DMSO	In frame, conventional; E15-E14	TCF3.chr19.1619315-.HLF.chr17.53398025+	184	0	266	NM_003200	NM_002126
PDX_11a_m1660_JQ1	intron 15	ERR3504860	JQ1	In frame, conventional; E14-E14	TCF3.chr19.1619779-.HLF.chr17.53398025+	625	0	11	NM_003200	NM_002126
PDX_11a_m1660_JQ1	intron 15	ERR3504860	JQ1	In frame, conventional; E15-E14	TCF3.chr19.1619315-.HLF.chr17.53398025+	150	0	251	NM_003200	NM_002126
PDX_11a_m1660_A-485	intron 15	ERR3504861	A-485	In frame, conventional; E14-E14	TCF3.chr19.1619779-.HLF.chr17.53398025+	328	0	15	NM_003200	NM_002126
PDX_11a_m1660_A-485	intron 15	ERR3504861	A-485	In frame, conventional; E15-E14	TCF3.chr19.1619315-.HLF.chr17.53398025+	102	0	132	NM_003200	NM_002126
PDX_11a_m1699_DMSO	intron 15	ERR3504862	DMSO	In frame, conventional; E14-E14	TCF3.chr19.1619779-.HLF.chr17.53398025+	791	0	48	NM_003200	NM_002126
PDX_11a_m1699_DMSO	intron 15	ERR3504862	DMSO	In frame, conventional; E15-E14	TCF3.chr19.1619315-.HLF.chr17.53398025+	248	0	348	NM_003200	NM_002126
PDX_11a_m1699_JQ1	intron 15	ERR3504863	JQ1	In frame, conventional; E14-E14	TCF3.chr19.1619779-.HLF.chr17.53398025+	637	0	16	NM_003200	NM_002126
PDX_11a_m1699_JQ1	intron 15	ERR3504863	JQ1	In frame, conventional; E15-E14	TCF3.chr19.1619315-.HLF.chr17.53398025+	206	0	343	NM_003200	NM_002126
PDX_11a_m1699_A-485	intron 15	ERR3504864	A-485	In frame, conventional; E14-E14	TCF3.chr19.1619779-.HLF.chr17.53398025+	371	0	25	NM_003200	NM_002126
PDX_11a_m1699_A-485	intron 15	ERR3504864	A-485	In frame, conventional; E15-E14	TCF3.chr19.1619315-.HLF.chr17.53398025+	115	0	186	NM_003200	NM_002126
PDX_11a_m1700_DMSO	intron 15	ERR3504865	DMSO	In frame, conventional; E14-E14	TCF3.chr19.1619779-.HLF.chr17.53398025+	697	0	41	NM_003200	NM_002126
PDX_11a_m1700_DMSO	intron 15	ERR3504865	DMSO	In frame, conventional; E15-E14	TCF3.chr19.1619315-.HLF.chr17.53398025+	221	0	334	NM_003200	NM_002126
PDX_11a_m1700_JQ1	intron 15	ERR3504866	JQ1	In frame, conventional; E14-E14	TCF3.chr19.1619779-.HLF.chr17.53398025+	715	0	18	NM_003200	NM_002126
PDX_11a_m1700_JQ1	intron 15	ERR3504866	JQ1	In frame, conventional; E15-E14	TCF3.chr19.1619315-.HLF.chr17.53398025+	220	0	328	NM_003200	NM_002126
PDX_11a_m1700_A-485	intron 15	ERR3504867	A-485	In frame, conventional; E14-E14	TCF3.chr19.1619779-.HLF.chr17.53398025+	348	0	28	NM_003200	NM_002126
PDX_11a_m1700_A-485	intron 15	ERR3504867	A-485	In frame, conventional; E15-E14	TCF3.chr19.1619315-.HLF.chr17.53398025+	99	0	173	NM_003200	NM_002126

- The methods section would benefit from additional details such that the process of identifying candidate fusions as performed in this study is more transparent and reproducible. This relates to my comment above regarding how fusions were initially filtered, but also extends to how the downstream analysis of neo-splicing events were discovered. Was neo-splicer somehow run on every fusion candidate that met some predefined criteria? Was finding or not finding neo-spliced exons one of the criteria that was initially used to define the set of filtered in-frame fusions, such that all initially defined frameshifted fusions were examined for candidate neo-splicing events that would restore the reading frame?

[Response] This comment significantly improved our presentation. We added following details to make the study more reproducible in “Fusion detection” section: “Oncogenic fusions were detected by using state-of-the-art methods reported to have superior performance^{10,11}, including Cicero¹¹, Arriba¹², STAR-fusion¹³, and FusionCatcher¹⁴. Cicero was run on bam files aligned with STAR v2.5.3a while Arriba, STAR-fusion, and FusionCatcher were run on fastq files.”

To ensure clinical relevance of detected oncogenic fusions, we curated a list of 315 driver genes (which included 142 and 77 significantly mutated genes from two non-overlapping pan-childhood cancer analyses^{15,16} and a number of disease-focused analyses including ependymoma¹⁷, Ewing sarcoma^{18,19}, rhabdomyosarcoma²⁰, low grade glioma²¹, high grade glioma²², T-cell acute lymphoblastic lymphoma²³, acute myeloid leukemia²⁴, as well as a recent clinical genomics report²⁵) for childhood cancers (**Supplementary Table 18**) reported in literature^{5,9,15-22,24,26-34} and used this gene list to extract candidate fusions predicted by these tools for manual review. This procedure resulted in 2,004 samples with 257 oncogenic fusion pairs (**Supplementary Table 19**). Upon manual review on remaining predictions, an additional 5 samples (**Supplementary Table 19**) were assigned as low confidence fusions in an ad hoc manner (for example, a rhabdomyosarcoma with *MEGF10-NUP210* fusion and a brain tumor sample with *NRCAM-PRKCB* fusion).

In some literature the terminology of “fusion” is used interchangeably with “structural alteration/rearrangement”. Because gene fusion is a much earlier concept following the discovery of *BCR-ABL1* fusion in Philadelphia chromosome positive leukemia, here we propose to use “fusion” exclusively in transcriptome (or gene expression) setting and to use “structural alteration/arrangement” in genome (or DNA) setting, although their biological meaning could be identical and can be discerned from context. Clearly not all gene fusions may carry biological functions like *BCR-ABL1* in Philadelphia chromosome leukemia. For example, chromosomal rearrangements that lead to inactivation of tumor suppressor genes (e.g., *CDKN2A*, *RB1*, *ATRX*) can be detected as out-of-frame fusion transcripts but are beyond the scope of this work. In this work, “oncogenic fusion” indicates in-frame fusions that produce oncogenic proteins like *BCR-ABL1*.

The fusion detection was performed in an institutional (St. Jude) high performance computing cluster with 227 nodes, 474 CPUs (12128 Cores) and 194TB of RAM and 20 petabytes of useable parallel file system storage, connected through 40 Gigabit Ethernet links. Although Cicero, Arriba and STAR-Fusion takes less than 2 days to finish for most of samples, we noticed that the earlier FusionCatcher version (v1.10) runs slowly for most samples. Therefore, we generated minibams by including known driver gene regions of childhood cancers (**Supplementary Table 17**) to validate the findings from Cicero, Arriba, and STAR-Fusion. Most of the jobs on these minibams can be finished around 10 days. Due to limited project storage space allocated for the laboratory, and the large storage space needed due to raw fastq and bam files as well as intermediate files, we analyzed the data in batches. The total download, re-download, run, re-run and analysis time of this cohort (using up to 500 jobs at any given time) took us about 1 year. Interestingly, a recent FusionCatcher version (v1.33) runs much faster for full bams (typically ~1 day) and we were able to finish the re-run of all four fusion detectors on all full bams/fastqs of our full cohort in less than 3 months for revision. All raw output from the four fusion detectors were deposited in zenodo (<https://doi.org/10.5281/zenodo.6421699>).

The commands used for the fusion detectors are as follows: (1) Arriba (v1.2.0): `arriba -o good_fusions.tsv -O discarded_fusions.tsv -a genome_assembly -g annotation_gtf -b black_list -T -P r1.fq r2.fq`; (2) Cicero (v0.3.0): `Cicero.sh -b bamfile -g genomeVersion -r cicero_refdir -j junctions.tab`; (3) STAR-Fusion (v1.6.0): `STAR-Fusion --left_fq r1.fq --right_fq r2.fq --genome_lib_dir genome_lib`; (4) FusionCatcher (v1.33): `fusioncatcher -d DATADIR -i r1.fq,f2.fq`. Here r1.fq and r2.fq indicates the fastq files of read1 and read2, respectively.”

Regarding the filtering, we added following details in “Model of fusion etiology and study design” section: “Candidate oncogenic fusions were detected using tools (Arriba¹², STAR-Fusion¹³, CICERO¹¹, and FusionCatcher¹⁴; **Methods**) reported to have superior performance^{10,11}. The detected candidate fusions were compared with previous genomics studies on childhood cancers^{5,15-22,24,26,27,29,39-42} to establish the comprehensive list of oncogenic fusions. To classify the detected oncogenic fusions into one of above four categories, we developed novel tools Neo-Versioner (to classify intronic versioning) and Neo-Splicer (to classify neo-splicing; see **Method** and **Extended Data Fig. 1**). If the fusion does not belong to either intronic versioning (i.e., no reads supporting natural exonic junctions between N’ and C’ genes) or neo splicing categories by our automated analysis, we manually review and classify the fusion into the categories of either chimeric exon or neo translational.”

- The reported finding of total fusion gene length (summing the fusion gene partner lengths) being linearly correlated with patient prevalence is tenuous, and as the authors show, it does not hold for brain or solid tumors as examined here. The finding of a clear linear correlation between prevalence of KMT2A fusion partners and patient abundance according to the length of the gene fusion partner is quite interesting. However, the authors show just this one striking example. It would be worthwhile to explore all such fusions having multiple fusion partners to see if the linear correlation holds more generally.

[Response] Although we hope there is a single variable that can explain everything on the widely variable prevalence of oncogenic fusions, our study indicated it is unlikely. For example, CBFB-MYH11’s prevalence is significantly associated with protein domain usage. Likewise, in TCF3 fusions the local DNA property matters.

We updated the writing to stress that the association between gene length and patient prevalence was used as a null hypothesis to detect additional biological/clinical insights: “By assuming the association between gene length and patient prevalence as our null hypothesis, our extended analysis to brain tumor and solid tumor did not yield statistical significance (**Extended Data Fig. 2f-g**). This data may either reflect the diverse subtypes and corresponding smaller cohort sizes among brain tumor and solid tumor or indicate additional factors influencing the etiology of oncogenic fusions, which will be addressed in following sections.”

- The proposed dependence of fusion prevalence on gene length becomes the basis of the authors relative selection bias score (RSB), used to examine the relative selection of exon-versioned fusions where evidence of alternative splicing exists. Normalizing the number of fusion breakpoints by intron length does make intuitive sense even if the direct evidence for doing so appears weak or limiting as

shown. However, I do wonder if the RSB statistic is adding unnecessary complexity. Are there examples where the RSB statistic provides for more accurate estimates as opposed to using a simpler fusion version abundance ratio (lacking length normalization)?

[Response] We added below statement to highlight the importance of length normalization in section "Selection bias in fusion versioning": "To demonstrate the necessity of normalizing the patient prevalence using intronic length, we also generated data by excluding the length normalization. This resulted false positive findings (Supplementary Table 12), such as EWSR1-FLI1, CBFA2T3-GLIS2 and DEK-NUP214."

It would also be useful to know if fusion versions were limited to only the in-frame candidates primarily studied in this work, as considering out-of-frame alternative splicing would presumably impact the relative rankings of RSB statistics among fusion candidates; for example, theoretically there could be an out-of-frame fusion isoform fusion that dominates all in-frame fusions according to patient prevalence. We could make a similar argument for the splicing dominance statistic with regard to considering out-of-frame fusion versions that exist but may not have been considered during this study. Clarification here is warranted.

[Response] We clarified this confusion in section "Selection bias in fusion versioning" as "Only intronic versioning (i.e., natural in-frame fusions) were considered in this analysis."

- In the section on the "Landscape of childhood oncogenic fusions", it states "To test this hypothesis, we first validated the high accuracy (>91% RNA-based detections are within 5 base pairs of DNA-based detections; see Method; Supplementary Table 8; Extended Data Fig. 2a) of detecting DNA breakpoints by using RNA-seq data." In my experience, fusion transcripts as detected by RNA-seq data tend to have breakpoints at splice junctions, which can be quite distant from the point of genomic rearrangement that would be detected using DNA-seq data. This is consistent with Figure 5a, showing RNA breakpoints at splice sites, and DNA breakpoints scattered throughout the intron. So it isn't clear to me how the authors identified genomic breakpoints using RNA data in such cases. This requires more explanation in both the methods and the main text.

[Response] We added in section "Detecting DNA breakpoints from RNAseq" following statements to clarify: "To benchmark the specificity of this method, we collected whole-genome sequencing (WGS)-based DNA breakpoints from published papers on applicable samples (Supplementary Table 8) as gold standard. As it turned out, >91% RNA-based DNA breakpoint detections are within 5 base pairs of DNA-based detections, validating the high accuracy of our method. However, we note that not all DNA breakpoints can be determined from RNAseq data, due to varying RNAseq protocols (poly-T based mRNA-seq or total RNAseq that contains pre-spliced transcripts) or sampling fluctuations."

Minor:

- I appreciate that the authors put all code and related data on Zenodo. For tools developed here such as NeoSplicer, making the code, documentation, and test data available on GitHub would be far more appropriate. Also, more details in the very short manifest file describing the various scripts and data would also be helpful. I truly applaud the authors for their current efforts in making all these materials available, and I simply suggest making them slightly more accessible as companion materials. Finally, what's in the SV_Detection folder? It's not mentioned in the paper as far as I could tell.

[Response] We appreciate Reviewer #2's great interest in the codes. We have updated our manifest file to briefly introduce the folder structures and their basic purpose. For example, we now state that "SV_Detection" folder is "used to determine DNA breakpoints from RNAseq data".

Due the length of this manuscript, and the massive burden of executing the CRISPR experiments in additional 20 cell lines with neo-splicing and/or chimeric exon features, we are drafting a data portal manuscript that allow readers to interactively explore the findings in this work and run the analyses using our tools (which will be part of the well-received data portal of <https://pecan.stjude.cloud/proteinpaint/TP53>)⁴³.

- I expect there will be much interest in NeoSplicing as reported in this manuscript. If the authors can provide additional bam or GTF and fasta files for these fusions that would facilitate visualization using a tool such as IGV, that would make it easier for more in-depth exploration. With bam formatting, it would be possible to examine the template-only insertion sequences in the viewer.

[Response] We have obtained approval from dbGaP for access of all datasets analyzed. Interested readers should be able to obtain access from dbGaP as well. On the other hand, we have generated and uploaded RNAseq data for HAL-01 and UoC-B1 cell lines, in response to Reviewer's critiques, in section "Data availability": "The total RNAseq data for HAL-01 and UoC-B1 cell lines are deposited at ENA (<https://www.ebi.ac.uk/ena/browser/view/PRJEB55308>)." We regret that we cannot make the patient data publicly available due to HIPAA regulation.

- In Figure 5a, I recommend labeling the N and C fusion genes. At first glance, the figure looked to me like the bottom part was supposed to represent a zoomed-in view of the upper part. It could be just me and the way I tend to make those types of 'zoomed in' figures, though. The N- and C- gene labels would have helped me personally.

[Response] Thanks, we now labeled the N and C genes in Figure 5a to avoid confusion.

- Line 67: In the statement "We demonstrate a novel mathematical model that can detect differential selection pressure, which is validated in CFBF-MYH11 positive acute myeloid leukemia (AML) to confer superior prognostic value on event free survival than other well-known clinical features." the 'novel mathematical model' seems a bit reaching. This is using the RSB metric, a length-normalized version of a relative patient abundance for introns - where intron length normalization seems intuitive but potentially unnecessary. In addition to clarifying whether out-of-frame fusion versions are considered, I'd suggest incorporating each of the fusion version comparisons tested in Fig. 5e, or as a companion supplemental figure so we can see how all the other combinations fare.

[Response] We added below statement to highlight the importance of length normalization in section "Selection bias in fusion versioning": "To demonstrate the necessity of normalizing the patient prevalence using intronic length, we also generated data by excluding the length normalization. This resulted false positive findings (Supplementary Table 12), such as significant selection bias among versions would be detected for fusion EWSR1-FLI1, CBFA2T3-GLIS2 and DEK-NUP214."

We also clarified that only in-frame fusions are considered in section "Fusion detection" as: "In some literature the terminology of "fusion" is used interchangeably with "structural alteration/rearrangement". Because gene fusion is a much earlier concept following the discovery of

BCR-ABL1 fusion in Philadelphia chromosome positive leukemia, here we propose to use “fusion” exclusively in transcriptome (or gene expression) setting and to use “structural alteration/arrangement” in genome (or DNA) setting, although their biological meaning could be identical and can be discerned from context. Clearly not all gene fusions may carry biological functions like *BCR-ABL1* in Philadelphia chromosome leukemia. For example, chromosomal rearrangements that lead to inactivation of tumor suppressor genes (e.g., *CDKN2A*, *RB1*, *ATRX*) can be detected as out-of-frame fusion transcripts but are beyond the scope of this work. In this work, “oncogenic fusion” indicates in-frame fusions that produce oncogenic proteins like *BCR-ABL1*.”

- For the HAL-01 cell line RNA-seq, please indicate the SRA accession for the data used.

[Response] We generated WGS and RNAseq data for the cell line. We updated the main text as following (in section “Data availability”): “The total RNAseq data for HAL-01 and UoC-B1 cell lines are deposited at ENA (<https://www.ebi.ac.uk/ena/browser/view/PRJEB55308>).”

- Including some details about how the computes were run, presumably in the cloud at St. Jude, would be helpful in the methods. Any time thousands of computationally intensive jobs are needed, it's useful for others to know about how such a feat was accomplished.

[Response] We updated the method section (“Fusion detection”) to indicate the running time as following: “The fusion detection was performed in an institutional (St. Jude) high performance computing cluster with 227 nodes, 474 CPUs (12128 Cores) and 194TB of RAM and 20 petabytes of useable parallel file system storage, connected through 40 Gigabit Ethernet links. Although Cicero, Arriba and STAR-Fusion takes less than 2 days to finish for most of samples, we noticed that the earlier FusionCatcher version (v1.10) runs slowly for most samples. Therefore, we generated minibams by including known driver gene regions of childhood cancers (Supplementary Table 17) to validate the findings from Cicero, Arriba, and STAR-Fusion. Most of the jobs on these minibams can be finished around 10 days. Due to limited project storage space allocated for the laboratory, and the large storage space needed due to raw fastq and bam files as well as intermediate files, we analyzed the data in batches. The total download, re-download, run, re-run and analysis time of this cohort (using up to 500 jobs at any given time) took us about 1 year. Interestingly, a recent FusionCatcher version (v1.33) runs much faster for full bams (typically ~1 day) and we were able to finish the re-run of all four fusion detectors on all full bams/fastqs of our full cohort in less than 3 months for revision. All raw output from the four fusion detectors were deposited in zenodo (<https://doi.org/10.5281/zenodo.7033077>).”

- For the SDS statistic, how were the RNA-seq alignments derived? Using STAR or another aligner?

[Response] In section “Fusion detection”, we updated the description as “Oncogenic fusions were detected by using state-of-the-art methods reported to have superior performance^{10,11}, including Cicero (v0.3.0)¹¹, Arriba (v1.2.0)¹², STAR-fusion (v1.6.0)¹³, and FusionCatcher (v1.33)¹⁴. Cicero was run on bam files aligned with STAR v2.5.3a while Arriba, STAR-Fusion, and FusionCatcher were run on fastq files.”

We updated the SDS statistic calculation as: “To measure potential alternative splicing, we introduced a splicing dominance score (SDS; Fig. 4a). For this, we first calculated the read support (X_i) for all fusion

versions *i* (with minimum of 3 supporting reads) detected in a sample (aligned with STAR v2.5.3a) with the index fusion.”

- Please include commands executed for the various fusion finders in the supplement. If workflows were used (ie. WDL, CWL, nextflow, etc.), providing the workflows would be useful too.

[Response] We clarified this information in section “Fusion detection”: “The commands used for the fusion detectors are as follows: (1) Arriba (v1.2.0): `arriba -o good_fusions.tsv -O discarded_fusions.tsv -a genome_assembly -g annotation_gtf -b black_list -T -P r1.fq r2.fq`; (2) Cicero (v0.3.0): `Cicero.sh -b bamfile -g genomeVersion -r cicero_refdir -j juncitons.tab`; (3) STAR-Fusion (v1.6.0): `STAR-Fusion --left_fq r1.fq --right_fq r2.fq --genome_lib_dir genome_lib`; (4) FusionCatcher (v1.33): `fusioncatcher -d DATADIR -i r1.fq,r2.fq`. Here r1.fq and r2.fq indicates the fastq files of read1 and read2, respectively.”

- Fig 6d - the g4 negative control is shown, but not the g5 negative control. Is there a reason why g5 was excluded here?

[Response] We have updated the caption of Fig. 6d to include g₅ data: “Negative control guide (g₄; see Extended Data Fig. 6b for a similar pattern in g₅) that targets upstream ...”

- In ST2, it would be useful to include the counts of number of samples for each data set, as listed in the methods section. Also, it would be useful to have another supplementary table that indicates the sample name and corresponding data set association, which can be cross-referenced in the other data tables. Other metadata about the individual samples would be useful too including the number of reads in each. This way, one can determine from the fusion read counts what proportion of the total reads they represent.

[Response] A new Supplementary Table ST2 is added (screenshot below). Sample and data set association is listed in Supplementary Table ST1. Read count from different tools are found in Supplementary Table ST3.

Table ST2. Data source of this study. Listed are data source tags (Column A) used in Table ST1, source publication or data portal URL (column D), short description of source study (Column C), and dataset ID from source URL when applicable (Column B) and number of patients involved (Column E).

Data Source Tag	Data Source Identifier (when applicable)	Short Description of Study	Source URL	#patient
CBTN		Children’s Brain Tumor Network	https://portal.kidsfirstdrc.org	725
FredHutch	TARGET-AML	Pediatric AML	https://portal.gdc.cancer.gov/	1088
G4K	SJC-DS-1004	Genomes 4 Kids	https://platform.stjude.cloud/data/cohorts	253
PCGP	SJC-DS-1001	St. Jude-Washington University Pediatric Cancer Genome Project	https://platform.stjude.cloud/data/cohorts	777
RTCG	SJC-DS-1007	Real-time Clinical Genomics	https://platform.stjude.cloud/data/cohorts	1006
PMID_24436047		Rhabdomyosarcoma	https://pubmed.ncbi.nlm.nih.gov/24436047/	84
PMID_25010205		Ewing’s Sarcoma	https://pubmed.ncbi.nlm.nih.gov/25010205/	62
PMID_25186949		Ewing’s Sarcoma	https://pubmed.ncbi.nlm.nih.gov/25186949/	22
PMID_27798625		Pediatric AML	https://pubmed.ncbi.nlm.nih.gov/27798625/	23
PMID_29146900		Pediatric MDS	https://pubmed.ncbi.nlm.nih.gov/29146900/	14
PMID_30760869		Pediatric AML	https://pubmed.ncbi.nlm.nih.gov/30760869/	26
PMID_30926971		Acute Erythroleukemia	https://pubmed.ncbi.nlm.nih.gov/30926971/	8
PMID_31350825		Cytogenetically normal AML	https://pubmed.ncbi.nlm.nih.gov/31350825/	1
PMID_31697823		Therapy induced resistance in ALL	https://pubmed.ncbi.nlm.nih.gov/31697823/	101
PMID_33579957	SJC-DS-1011	Pediatric Therapy-related Myeloid Neoplasms	https://pubmed.ncbi.nlm.nih.gov/33579957/	43
PMID_34778799		Pediatric AML	https://pubmed.ncbi.nlm.nih.gov/34778799/	87
PMID_35176137		Discovery of UBTF as a novel AML subtype	https://pubmed.ncbi.nlm.nih.gov/35176137/	111
TARGET		NCI TARGET project	https://pubmed.ncbi.nlm.nih.gov/29489755/	759

- the MS-Excel ST3 table had a filter automatically invoked, showing only a small subset of the total data. Best to undo that filter before submission.

[Response] We now disabled filters in all supplementary tables.

- in Figure 6, the guide numbers for cell line UoC-B1 reuse values g1 and g2. I'd recommend starting where the count left off with the HAL-01 guide numbering, using instead g6, g7 for clarity.

[Response] We now use g₆ and g₇ for UoC-B1 line.

- The PWM scoring is reported to be in ranges of <2 to >6. It isn't immediately clear how such score numbers are being derived from the LLR equation. The LLR would usually give large negative values, where the least negative value reflects the highest probability. Please include more details on how the LLR converts to the score value.

[Response] In section "Calculating pseudo binding affinity for splice sites", we added below statements to clarify: "To ensure the quality of our constructed motifs, we scored all splice sites of known human genes and confirmed most of the splice sites received positive scores (>80% donors have score >4; >80% acceptors have score >4.3). As a negative control, we extracted 1.12 million potential donor (GT) sites and 1.76 million potential acceptor (AG) sites that do not belong to known human genes from forward strand of chr19 (one of the shortest chromosomes to save computation time) and scored them. As it turned out, >90% of such false donors have score <4 and >90% of such false acceptors have score <4.3, validating the power of our PWM method in discriminating real splice sites from non-real sites (see Supplementary Figure 2 for the score distribution of true and false splice donors and acceptors)."

- In extended figure 6, the row values don't always sum to 100 (usually 99), which I found curious.

[Response] We clarified this in caption of Extended Data Fig. 6 as "Due to rounding error, the row sums can be 100 or 99."

- in Fig 6D, 'non-lethal' is used correctly to describe the negative control situation for in-frame and frameshift. In extended Figure 6, 'lethal' and 'non-lethal' are used for the negative control context. It might be better to use 'frameshift' and 'in-frame' instead, since both are non-lethal here as well.

[Response] Now the Extended Fig. 6b legend reads: "For panels b and c, instead of using lethal/non-lethal, a tag of frameshift/in-frame was given to each indel according to its length because the target region is genuine intronic (thus a negative control)"

- I'm curious why the RSB metric was chosen for pairwise comparisons as opposed to using an SDS like metric that accounts for the proportion by summing the total in the denominator. Would an SDS-like RSB metric outperform the RSB itself for ranking hazard ratios or be simpler for identifying best therapeutic candidates?

[Response] This comment significantly improved our presentation. We updated in section "Selection bias in intronic versioning" as: "Because intronic versioning can cause amino acid differences in the

fusion protein which may in turn lead to potential functional difference, we hypothesized that intronic versioning could confer differential fitness to the host cells in some oncogenic fusions. **To measure the effect size of potential selection bias between two intronic versions**, we proposed a relative selection bias score (RSB) based on the observation that DNA breakpoints are generally distributed in introns in a near-uniform fashion and gene length can predict prevalence in patients (**Fig. 2h-i**). In this model, the patient prevalence of DNA breakpoints in a given intron should be proportional to its length if the resultant protein versions are functionally equivalent (i.e., confers the same positive selection pressure). **The statistical significance of selection bias is measured by comparing (using a Chi-squared test) the observed patient prevalence in all intronic versions and corresponding expected patient prevalence under a null hypothesis that patient prevalence is proportional to intronic length.**"

- In ST3 (and ideally, the original unfiltered starting fusion set), please include the programs that initially predicted them along with fusion evidence read support. For ST3, the NeoVersioner recomputed read support for predicted fusions would be useful for comparing to the originally defined read support quantities from the individual tools.

[Response] Please see patient level fusion detection along with the detection tools in Supplementary Table ST1. Read count from different fusion detection tools are provided in Supplementary Table ST3. Please refer to Supplementary Table ST7 for read count information from NeoVersioner.

- in NeoSplicer, it isn't obvious how the non-template fusion sequences were defined as shown. The algorithm for doing this should be described as part of the NeoSplicer methods section.

[Response] We clarified this in section "Neo-Splicer" as: **"To detect novel splice sites, this method requires DNA breakpoints between the two genes along with the non-template insertion sequence if there is, since we note that sometime the novel splice sites can be embedded in the non-template insertion sequences⁴⁴ flanking the rearrangement boundary.** Given the ubiquitous nature of candidate splice sites (AG and GT; 1 in every 16 nucleotides expected by random chance), we first detected putative splice sites by using Position Specific Weight Matrix method as described in section **"Calculating pseudo binding affinity for splice sites"**. Second, given the DNA breakpoints (39% (n=768/1,988) chance of detection in RNAseq data, **Fig. 2j-l; Supplementary Table 8**) of an oncogenic fusion, we enumerated all AG and GT dinucleotides between intact exons of involved genes, generated hypothetical exons, and checked corresponding translation frames. RNAseq reads **were** then compared with above predictions to determine the neo splice sites and corresponding isoforms used by the cancer cells (**Extended Data Fig. 1b**). **We note that the cryptic neo-spliced exon and/or non-template insertion sequences can remain poorly mapped by standard mapper (here STAR v2.5.3a), such as the non-template sequences in TCF3-HLF fusion in HAL-01 and UoC-B1 (Fig. 6b,f; Extended Data Fig. 5d), especially when the neo-splice sites are within non-template sequences (fusion C11orf96-MAML2 in case SJEPD031093 in Extended Data Fig. 5a). These mapping challenges are resolved by Neo-Splicer using DNA contigs."**

- it is curious that many of the analysis tools used in this work were hand-crafted by the authors. For example, instead of using commonly leveraged alignment utilities, the authors used their own kmer-based read mapping schemes. Such analyses, like identifying DNA breakpoints from RNA-seq data in intronic regions, being dependent on custom utilities (instead of using trusted alignment methods like bowtie, STAR, bwa, etc) does make one question the accuracy of the methods being used. Showing a situation such as uniform insertions across intronic regions using custom methods, where we know intronic regions are repeat-rich and more difficult to accurately align reads to than exonic regions, could result from inaccurate / random placement of reads. I'm not suggesting that the authors methods are

inaccurate, but rather suggesting that the authors would be best to demonstrate that their methods can accurately perform their function, such as by simulating data and ensuring accurate results using simulated data. This would be most important for NeoSplicer and the method inferring DNA breakpoints from RNA-seq data.

[Response] This comment greatly improved the clarity of our presentation. We updated the strength of our developed methods in section “Model of fusion etiology and study design” as: “To classify the detected oncogenic fusions into one of above four categories, we developed novel tools Neo-Versioner (to classify intronic versioning) and Neo-Splicer (to classify neo-splicing; see **Method** and **Extended Data Fig. 1**). If the fusion does not belong to either intronic versioning (i.e., no reads supporting natural exonic junctions between N' and C' genes) or neo splicing categories by our automated analysis, we manually review and classify the fusion into the categories of either chimeric exon or neo translational.”

Regarding the validation DNA breakpoints inference, we believe our real-world DNA-sequencing based validation (in section “Landscape of childhood oncogenic fusions” of original submission) can best serve this purpose than a simulated dataset: “This hypothesis implies that all eligible base pairs (under the constraints of splicing and translation frame; mostly intronic bases) in corresponding genes can contribute to functional oncogenic fusion and therefore DNA breakpoints should be uniformly distributed along the eligible introns. To test this hypothesis, we first validated the high accuracy (>91% RNA-based detections are within 5 base pairs of DNA-based detections; see **Method**; **Supplementary Table 8**; **Extended Data Fig. 2a**) of detecting DNA breakpoints by using RNAseq data.”

We also clarified our motivation for developing a k-mer based approach over standard mappers: “To detect novel splice sites, this method requires DNA contig around the rearrangement as input. We note that sometime the novel splice sites can be embedded in the non-template insertion sequences⁶⁵ flanking the rearrangement boundary. Given the ubiquitous nature of candidate splice sites (AG and GT; 1 in every 16 nucleotides expected by random chance), we first detected putative splice sites by using Position Specific Weight Matrix method as described in section “Calculating pseudo binding affinity for splice sites”. Second, given the DNA breakpoints (39% (n=768/1,988) chance of detection in RNAseq data, **Fig. 2j-l**; **Supplementary Table 8**) of an oncogenic fusion, we enumerated all AG and GT dinucleotides between intact exons of involved genes, generated hypothetical exons, and checked corresponding translation frames. RNAseq reads were then compared with above predictions to determine the neo splice sites and corresponding isoforms used by the cancer cells (**Extended Data Fig. 1b**). We note that the cryptic neo-spliced exon and/or non-template insertion sequences can remain poorly mapped by standard mapper (here STAR v2.5.3a), such as the non-template sequences in *TCF3-HLF* fusion in HAL-01 and UoC-B1 (**Fig. 6b,f**; **Extended Data Fig. 5d**), especially when the neo-splice sites are within non-template sequences (fusion *C11orf96-MAML2* in case SJEPD031093 in **Extended Data Fig. 5a**). These mapping challenges are resolved by Neo-Splicer using DNA contigs.”

- Line 81: typo, should read "(C-terminus)"

[Response] Now it reads “Oncogenic fusions typically involve two genomic loci (genes) denoted as N’ gene (N-terminus) and C’ gene (C-terminus).”

- Lines 177, 517: RNA-seq does not provide 'absolute expression level'. RNA-seq provides relative expression levels (normalized by total read sequencing depth). Perhaps just change to 'expression level'.

[Response] Now it reads “By further calculating expression level measured” and “We calculated FPKM values as previously described⁷ to study the expression level of C’ genes in the promoter-hijacking-like category”

- Line 247: typos, should be 'division with denominator three equals zero'

[Response] Now it reads “the remainder of a division with denominator 3 equals 0”

- Line 402: typo, rational

[Response] Now it reads “to enable rationale design of guide RNAs”

- Line 588: typo, pools

[Response] Now it reads “for analyzing splice site disruption in the edited cell pools”

Reviewer #3 (Remarks to the Author): Expert in paediatric cancer genomics and gene fusions

This study by Liu et al. provided information on the etiology of 1,989 fusions identified in 5,190 childhood cancers. This report studied a wide spectrum of cancer, including leukemia, CNS, and solid tumors. The manuscript was well-written and provided exquisite detail. The authors identified 5 different molecular mechanisms by which oncogenic fusions may be generated, highlighting the large diversity in types of fusions. While different types of fusions have been described extensively in prior literature, Liu et al provided additional biological context into not only general classes of fusions but specific fusions as well. This work does demonstrate some clinical and therapeutic implications, which I would be interested in seeing highlighted more as childhood cancers tend to be underrepresented with regard to targeted therapies and clinical trials.

[Response] We appreciate Reviewer #3’ encouraging comments. We added below statement to stress the fact that childhood cancers are underrepresented: “To address these questions, and with the consideration of childhood cancers are underrepresented in targeted therapies⁴, we comprehensively studied oncogenic fusions by using tumor transcriptome sequencing datasets of 5,190 patients from publicly available childhood cancer cohorts.”

Major points:

- I do not see a caption for Supplementary Figure 1

[Response] Thanks for catching this oversight. We have added the caption as following: “Recurrent ($n \geq 3$) oncogenic fusions identified in 5,019 childhood cancers. For each gene pair, coding exons (thick boxes) are colored white (frame 0), gray (frame 1), and black (frame 2). Intronic length is indicated by numbers. Gray lines indicate theoretically in-frame fusions. Red lines indicate in-frame fusions observed in patients and its width indicate patient prevalence. RefSeq identifiers are indicated for each involving gene. Fusions involving neo-splicing or chimeric exons were not included (see paper).”

- Extended Data Figure 3 (Lines 37-39) “On the other hand, FOXO1 appears to have higher expression than PAX3/PAX7 in normal brain tissues, consistent with the observation in brain tumors (Fig. 3b).” Why is normal brain tissue from GTEx being used as the comparator for FOXO1 expression when this fusion is found predominately in muscle/soft tissue?

[Response] We apologize for this typo. We updated the sentence as “On the other hand, FOXO1 appears to have higher expression than PAX3/PAX7 in normal muscle/soft tissues, consistent with the observation in rhabdomyosarcoma, a type of sarcoma made up of cells that normally develop into skeletal muscles (Fig. 3b).”

- For those fusions with alternative splicing, did the authors look at other transcript isoforms to see if fusions derived from alternative splicing may be due to multiple transcript isoforms for the same gene being expressed in the tissue type under study?

[Response] We updated the main text (section “Alternative splicing in oncogenic fusions”) as “For exons involved in alternative splicing, we also investigated whether they could match any known isoforms of the host gene. We detected only one recurrent alternative exon (exon 12 in NM_016320) of NUP98 that matches a second isoform NM_001365129 (Supplementary Table 11).”

- The section on alternative splicing in fusion generation was intriguing since through clinical testing we have noted a number of fusions, including those involving RUNX1 and ABL1 (to name a couple), more frequently present with multiple fusion breakpoints in the same analyzed tumor sample. We have wondered whether alternative splicing has a role here and clinically which breakpoint is more informative to the providers or clinical outcomes. Did you find any samples in your data where different breakpoints representing the same fusion were seen in the same patient sample and if so, how did you determine which one was the predominant isoform?

[Response] Indeed, we observed RUNX1 alternative splicing in the original submission: “As it turned out, alternative splicing in *ETV6-RUNX1* (identified in B-cell leukemia) is recapitulated in *RUNX1* gene in normal GTEx blood samples, and alternative splicing in *C11orf95-RELA* (identified in Ependymoma, a brain tumor) is recapitulated in *C11orf95* in normal GTEx brain samples (**Extended Data Fig. 4c**).”

We further clarified this in section “Discussion”: “Clearly, alternatively spliced exons and corresponding protein domains should be avoided in drug targeting due to their potentially dispensable nature. Our data also highlights the need to study the clinical implication of alternative splicing in oncogenic fusions. For example, in sample SJBALLO30563_D1 (with fusion *ETV6-RUNX1*), we discovered 11 reads supporting junction E5-E3 and 7 reads for junction E5-E4 (Supplementary Table ST7). Although we determined the predominant isoform (E5-E3) by read supports, it would be interesting to study if one or both of chimeric proteins are translated and the tumorigenesis function of these chimeric proteins.”

- Given the title, I was hoping for additional clinical and therapeutic implications with evidence provided from this study. The authors hypothesize that certain fusion etiologies may more appropriate for targeted therapies or CRISPR methodology but only provide data for a single fusion event and EFS. While, the CRISPR approach to targeting some of the fusions, particularly those in which there isn’t a targeted therapy available is interesting and promising, it is likely still a long off from clinical use. Additionally, those fusions that were hypothesized to be good drug targets lack a targeted therapeutic. Given that there is RNA-seq for all of these data, it may be interesting to look at patterns of expression

to see if there is overexpression or evidence for a pathway deregulation which may be targeted with currently available therapeutics (i.e. MEK inhibitors for KIAA1549-BRAF), although this may be beyond the scope of the manuscript. Can the authors provide any additional outcomes data for the recurrent fusions described in this manuscript?

[Response] We appreciate Reviewer #3's curiosity into the targeting of oncogenic fusions. We are currently testing our CRISPR approach in additional 20 cell lines harboring neo-splicing or chimeric exons. However, the extent of that study would warrant a separate manuscript. We clarified this in Discussion: "However, we provide proof-of-concept data for TCF3-HLF fusion, a rare ALL subtype associated with a high rate of treatment failure⁴⁸ that clearly would benefit from such targeted therapy. It would be interesting to test if this strategy is broadly applicable to all oncogenic fusions with neo-splicing or chimeric exon feature. Furthermore, additional studies are needed to develop innovative targeting strategies for patients without such "easy-to-design" targeting strategies."

As agreed by Reviewer #3, this work is primarily focused on genetic vulnerabilities encoded in the DNA sequences and therefore overexpression and pathway deregulation are beyond the scope of this work.

Unfortunately, the outcome data are typically not well curated for most cancer genomics profiling data. We expect such analysis to be better streamlined in future cohorts with a dedicated study design.

Minor points:

- Figure 1b and line 284 - The "NM1-PATZ1" fusion should be listed as "MN1-PATZ1"

[Response] Now line 284 reads "where all four MN1-PATZ1 cases in". Fig. 1b also updated.

- Ensure that gene names or fusions that are being discussed as the DNA or RNA are italicized

[Response] All gene names are italicized in main text.

- The tool FusionCatcher is written in as both FusionCatcher and Fusion-catcher. Please update to maintain consistency

[Response] We now use "FusionCatcher" throughout the manuscript

- Methods, Neo-Splicer (Line 501) – "where" should be "were"

[Response] Now it reads "RNAseq reads were then compared"

- Extended Data Figure 2 caption (Line 17) – "show" should be "shown"

[Response] Now it reads "Shown in y-axis is the number of DNA breakpoints"

- Extended Data Figure 2 caption (Line 23) – "RUNX1-RUN1T1" should be "RUNX1-RUNX1T1"

[Response] Now it reads "for oncogenic fusion RUNX1-RUNX1T1"

- Supplementary Table 3 – How was the number of supporting reads calculated? How did you account for potentially discrepant coordinates from the four fusion callers?

[Response] We clarified this question as below: “A significant challenge in using public fusion detection tools is the harmonization of their output. For breakpoints, we used 10 bp as buffer distance to harmonize their output (Supplementary Table 3). However, there is no straightforward way to harmonize read counts from different tools. Therefore, Neo-Versioner was used throughout the manuscript for all applicable analysis.”

- Figure 3b – “CBFB-MY11” should read “CBFB-MYH11”

[Response] Now it reads “CBFB-MYH11”

Point-by-point response to reviewer comments

(Round 2)

Reviewer #1 (Remarks to the Author):

The manuscript has been substantially improved and in the current form is acceptable for publication.

Reviewer #2 (Remarks to the Author):

2.1a While I appreciate several of the edits to the revised manuscript, the revisions unfortunately do little to address many of my earlier concerns. Most notably, a major failing of the work is the arbitrary decision to restrict all analyses to in-frame coding fusions. This ignores fusion variants to protein-coding genes that would lead to altered or truncated coding regions and important promoter-hijacking fusions. Such fusions are highly relevant to pediatric cancers.

[Response] We regret that Reviewer #2 still wants us to include truncated coding (tumor suppressor) genes after our expanded description on why tumor suppressor genes are ignored, in our first rebuttal. To stress that our rationale is carefully thought, we added **Supplementary Notes** in this revision, which will be collectively addressed in rebuttal section 2.1c.

2.1b For example, there is no mention of oncogenic fusion CRLF2::P2RY8, which is highly relevant to pediatric ALL, involves two protein-coding genes, but does not yield an in-frame fusion protein. Also, important promoter-hijacking oncogenic fusions relevant to pediatric cancer that are actually mentioned in the paper, including IGH::CRLF2/EPOR/DUX4, are not actually studied as part of this work, yet would be especially relevant in analyses involving fusion impact on expression as in Figure 3, related to the expression dominance score.

[Response] This “protein truncating” and “promoter-hijacking” question is collectively addressed in section 2.1c. We also added a **Supplementary Note 8b** to detail the reason why RNAseq data alone is not sufficient to study promoter hijacking as below:

8.b) Oncogenic fusions in the category of promoter/enhancer hijacking. In this category, a rearrangement can bring a strong promoter/enhancer to an otherwise silenced proto-oncogene and lead to its aberrant expression. When the novel promoter is far, the proto-oncogene may start its transcription from its own transcription start site, thereby leaving no split reads or discordant read pairs in RNAseq data for bioinformatic detection (**Fig. SN2a**). On the other hand, the transcripts may contain part of the novel promoter sequences when the novel promoter is closer (**Fig. SN2b**). Moreover, it is also possible that a point mutation in the native promoter can convert it to a strong active promoter (such as *TAL1* in pediatric T-ALL⁴⁵) and lead to aberrant expression (**Fig. SN2c**)— biologically this scenario is not promoter/enhancing *per se*. Clearly, without DNA (preferentially whole genome) sequencing data, scenarios a) and c) cannot be resolved by transcriptome sequencing and, a forced analysis will result in biased conclusions that does not meet our scientific rigor. Instead, such patterns are best studied in our ongoing project on the signatures of rearrangements (SVs) using whole genome sequencing in >1,500 pediatric cancer patients. Nevertheless, we provided results on known oncogenic fusions (*CRLF2*, *DUX4*, *EPOR*, *BCL11B*, **Table SN2a-e**) in promoter/enhancer-hijacking category to address Reviewer #2’s request though we did not perform systematic discovery.

Fig. SN2. Promoter/enhancer-hijacking. In hypothetical scenario (a), the blue chromosome (and a strong enhancer/promoter highlighted by blue oval) was brought proximity to the orange proto-oncogene and lead to its aberrant expression. Transcription only involves the proto-oncogene due to the space between the blue promoter and orange gene. In (b), the blue promoter contacts the transcription start site of orange gene, so that the transcription involves both the proto-oncogene and a small part of the blue chromosome. In (c), a point mutation in the promoter region may convert it to a strong active promoter to initiate the proto-oncogene without a fusion event (such as *TAL1* enhancer mutation in Mansour et al (2014)⁴⁵). Split reads or discordant read pairs are expected for scenario (b) but not scenario (a) or (c). DNA (preferentially whole genome) sequencing are needed to ascertain the fusion status. Nevertheless, the aberrant high expression of such proto-oncogene typically can help ascertain the tumor subtype.

2.1c Other analyses including the relative selection bias score continue to fail to consider transcript variant breakpoints that do not lead to in-frame coding regions. Because of this, there could very well be actual promoter hijacking fusion variants that yield the dominant fusion variant that are entirely disregarded when a lower expressed in-frame variant exists.

[Response] Although it is tempting to detect all cancer related alterations from RNAseq data, we regret that Reviewer #2 significantly over-estimated the power of RNAseq alone. In fact, in clinical testing (e.g., <https://pubmed.ncbi.nlm.nih.gov/34301788/>), DNA sequencing, in particular whole genome sequencing (WGS), is always preferred to include. This is because gene loss (such as *CDKN2A* deletion in leukemias) can be only definitively called in WGS while in RNAseq data this can remain as an inference by low/no expression when split read is absent. Although tumor suppressor gene can be lost due to gene truncation (as requested repeatedly by Reviewer #2), whole gene loss due to focal deletion is also frequently observed (e.g., *CDKN2A*). In this regard, detection of tumor suppressor gene loss in RNAseq data is NOT guaranteed, which is in clear contrast with activating mutations such as *KRAS* G12D or oncogenic fusions (that must be expressed to exert their biological functions) as investigated in this work. Therefore, in this work we intentionally skipped tumor suppressor genes using RNAseq data because we do not want to provide a potentially biased picture of such events that compromise our scientific rigor. Similarly, for promoter hijacking the detection of DNA breakpoints are not guaranteed in RNAseq (see rebuttal in 2.1b). In this revision we drafted a comprehensive background on the rationale of our study design as below. However, most of the information in this background is common knowledge in cancer genomics and we therefore put them into **Supplementary Notes** rather than in the main text/online methods.

1. Clinically recognized oncogenic fusions that generate chimeric proteins

Since the discovery of *BCR-ABL1* fusion oncoprotein in Philadelphia-chromosome leukemia⁴, classical cytogenetics method has revealed many additional cancer subtypes such as $t(1;19)(q23;q13)$ ⁴⁶ that was later determined to generate *TCF3-PBX1*⁴⁷, $t(17;19)$ ⁴⁸ that was later determined to generate *TCF3-HLF*³⁸ and $t(11;22)(q24;q12)$ that was determined to generate *EWSR1-FLI1*⁴⁹. With the advent of next generation sequencing technologies, oncogenic fusions are increasingly discovered in the past decade via simultaneously interrogating the genome (DNA sequencing) and the transcriptome (RNA sequencing) of tumors from patient cohorts of similar diagnosis, where DNA and RNA data cross validate each other. This include *C11orf95-RELA* (*C11orf95* recently renamed as *ZFTA*) in pediatric ependymomas (EPD)¹⁷, *KIAA1549-BRAF* in pediatric low-grade glioma (LGG)²¹. Supporting evidence from both DNA (on the rearrangement) and RNA (on the generation of chimeric protein) is a critical feature of these findings.

2. Clinically recognized oncogenic fusions that leads to aberrant expression of proto-oncogenes

In addition to the above “conventional” oncogenic fusions where a chimeric fusion oncoprotein is generated, there is another category of “promoter-hijacking” fusions. In this category, a constitutively active promoter or enhancer region is brought to a proto-oncogene (via chromosomal rearrangement) that is typically silenced in corresponding lineage of the cancer cells. Such rearrangement leads to aberrant expression of the proto-oncogene. Prominent examples of this category include *CRLF2/DUX4/EPOR* aberrant expression via rearrangement to immunoglobulin heavy chain (*IGH*) region B-ALL^{16,50}, *TAL1/TAL2* aberrant expression via rearrangement to T-cell receptor region (*TCR*) in T-ALL²³, *GFI1* aberrant expression via intra-chromosomal rearrangements to active enhancers in medulloblastoma⁵¹, *CRLF2* aberrant expression via intra-chromosomal rearrangement to *P2RY8* promoter⁵, as well as our newly discovered *BCL11B* aberrant expression in lineage-ambiguous leukemia³². Because no chimeric proteins are generated, this fusion category is typically termed “promoter/enhancer-hijacking”. Interestingly, other mutational mechanisms can also lead to such aberrant expression of proto-oncogenes. For example, the seminal work by Thomas Look and colleagues⁴⁵ has demonstrated that small insertions/deletions in enhancer regions of proto-oncogene *TAL1* can be sufficient to lead to its aberrant expression in pediatric T-ALL. Although corresponding tumors do not have chromosomal rearrangements (or fusion events) involving *TAL1*, we still consider these tumors as *TAL1* category.

3. Functional evidence of clinically recognized oncogenic fusions

Although experimentally challenging, putative oncogenic fusions such as *ZFTA-RELA* (also known as *C11orf95-RELA*) have recently been shown to be sufficient to drive pediatric ependymoma⁵². On the other hand, the success of imatinib on *BCR-ABL1*⁷, and the genetic knockout of oncogenic fusions such as *TCF3-HLF* in this work, have demonstrated that these oncogenic fusions, or more precisely the fusion oncoproteins they encode, plays an essential role to the survival of host cancer cells, which forms the basis of the hypothesis “oncogene addiction” that posits on the therapeutic value of targeting these oncogenic fusions.

4. Clinically recognized oncogenic fusions being invariable to clonal evolution and initiating driver

Comparison of tumors collected at initial diagnosis and at relapse for pediatric leukemia^{5,6} has reinforced the notion that subtype-defining oncogenic fusions are cancer initiating events¹¹. In these studies, the oncogenic fusions are always conserved between diagnosis and relapse

tumors, although other subclonal mutations (e.g., *CDKN2A* loss and *NT5C2* gain-of-function mutations) can be either eradicated or *de novo* acquired from diagnosis to relapse⁵. The clonal nature (i.e., being present in all cancer cells) of oncogenic fusions thus renders them ideal therapeutic targets.

5. Oncogene versus tumor suppressor gene (TSG)

In addition to the many oncogenic fusions mentioned above, extensive genome sequencing efforts in the past decade have led to the discovery of many additional significantly mutated genes also known as cancer drivers, in both adult^{53,54} and childhood cancers^{15,16}. In observation of these many cancer driver genes, Bert Vogelstein and colleagues⁵⁵ have pioneered the concept of classifying cancer driver genes into “tumor suppressor genes (TSG)” and “oncogenes”, where a “TSG” is a gene that, when *inactivated* by mutation, increases the selective growth advantage of the cell in which it resides, while an “oncogene” is a gene that, when *activated* by mutation, increases the selective growth advantage of the cell in which it resides. Under this broad concept, the oncogenic fusions mentioned above belong to the category of “oncogene” because corresponding fusion oncoproteins are hyperactive. On the other hand, the well-known TSGs including *CDKN2A* and *RB1*^{15,16} typically demonstrate inactivating (also known as loss-of-function) mutations, including gain of stop codon, protein-frame shifting, splice site altering, whole gene loss due to large deletion, or partial gene truncation due to focal deletion. A model of functional consequences on TSGs and oncogenes from diverse mutation types are illustrated in Fig. SN1 a-f.

Fig. SN1a. Diverse mutation types to disrupt a tumor suppressor gene (TSG; a) and less diverse mutation types for hyperactivation (b). In a TSG, a mutation (#1) that disrupts promoter or enhancer can lead to expression loss, a mutation (#2) that disrupts translation start codon ATG, a mutation (#3) that disrupts the gene structure via an intronic breakpoint, a mutation (#4) that disrupts the splice sites, and a mutation (#5) that disrupts the protein codon can all lead to loss of function (total gene deletion not illustrated). On the other hand, there are limited ways to make an oncogene hyperactive (panel b), which include a stronger promoter/enhancer via mutation #1, a stronger amino acid via mutation #3, or forming a chimeric protein via rearrangement mutation #2.

Fig. SN1b. Example oncogene *KRAS*. Chromosome coordinates (chr12) are shown on top and gene structure shown on bottom. *KRAS* mutations detected from pediatric cancers were shown as numbers that also indicate protein amino acid change. For example, there are 63 tumor specimens having mutations resulted in G12D, 22 tumor specimens having mutations resulted in A146T. Clearly, these protein coding mutations are in-frame and therefore can generate a mutant protein. Data in Fig. SN1b-f adapted (Oct 15, 2022) from <https://pecan.stjude.cloud/>⁴³.

Fig. SN1c. Example oncogene *ABL1*. In addition to the few point mutations detected from thousands of pediatric cancers, a prominent observation in *ABL1* is fusions, including 41 *BCR-ABL1*, 4 *ETV6-ABL1*, and a few other rare fusions such as *RCSD1-ABL1*, all shown as half-white-half-black circles. Breakpoints aligned with exon boundaries represent RNA breakpoints (i.e., splice junctions) while breakpoints not aligned with exon boundaries represent DNA breakpoints (black arrows). While DNA breakpoints can sometime be detected from RNAseq data (in total RNAseq protocol where pre-mRNA are included, see section SN 9 “Predicting DNA breakpoints from RNAseq data” in this Supplementary Notes), whole genome DNA sequencing typically ensure ascertainment of DNA breakpoints. On the other hand, DNA sequencing data typically cannot give definitive clue on RNA breakpoints (i.e., splice junctions) due to possibility of alternative splicing, such as *KMT2A* rearrangements in Fig. 4d of this work.

Fig. SN1d. Example oncogene *MYCN*. Beside P44L mutation in 29 tumor samples, *MYCN* does not have other highly recurrent alterations except copy number gain (red horizontal color bars in bottom) that are enriched in solid tumor (ST) neuroblastoma (NBL) that is detected in 19% of tumors (n=148). Further, *MYCN* amplification is also detected in Wilms tumor (ST, WLM) with frequency 11% (n=14). Note the common amplified region of *MYCN* and sometime the amplification can extend to far flanking regions of *MYCN*.

Fig. SN1e. Example tumor suppressor gene *RB1*. *RB1* gene has diverse mutation types in pediatric cancers, including stop gain mutations such as W78* in 8 specimens, R320* in 5 specimens. We also observed frameshifting mutations such as A74fs, L64fs, L317fs, D856fs. Moreover, the half-white-half-black circles indicate enrichment of structural rearrangements such as to gene *RCBTB2* in 5 patients and another 9 patients to another region in chr13. Further, focal deletions were detected in pediatric T-ALL (6%) and B-ALL (2%) that removed last several exons of *RB1*.

Fig. SN1f. Example tumor suppressor gene *CDKN2A*. Unlike TSG *RB1*, *CDKN2A* is enriched with copy number loss in pediatric T-ALL (54%) and B-ALL (11%). Although in some tumors the detection can be so focal that only few exons are affected (in this case it is possible to detect a truncating “fusion” from RNAseq data), in many tumors the size of deletion can be as big as arm level so that no truncating “fusion” transcripts are expected in RNAseq data.

6. Clinically recognized fusion-negative samples

Although oncogenic fusions have been routinely used for clinical subtyping, not all human cancers are fusion positive. For example, in pediatric B-ALL it has long been known that fusion-negative subtypes exist, including hyperdiploid (that with >50 chromosomes) and hypodiploid (those with <45 chromosomes) B-ALL⁵⁶. In pediatric neuroblastoma, extensive efforts in the study of whole genome, exome, and transcriptome sequencing data have not identified clinically

meaningful oncogenic fusions for most samples, except the well-known high *MYCN* amplification in ~20% of patients^{16,57,58}. In malignant rhabdoid tumours, *SMARCB1* homozygous loss is the only hallmark of nearly all patient tumors⁵⁹. Similarly, *RB1* homozygous loss is the only hallmark of nearly all retinoblastoma tumors⁶⁰. Clearly, candidate fusions detected in tumors of fusion-negative subtypes such as hyperdiploid B-ALL, neuroblastoma, rhabdoid or retinoblastoma tumors are more likely passenger events, if not artefacts, and scrutiny is warranted before accepting them as a true oncogenic fusion, as will be discussed in section **SN 11g** on mutual exclusivity pattern among oncogenic fusions. This data highlights the critical need of knowledge on well-defined tumor subtypes to ensure scientific rigor in reporting novel oncogenic fusions. In fact, clinically-relevant novel fusion-negative subtypes continue to be discovered, such as the novel subtype of *UBTF*-ITD in pediatric AML among the known clinical fusion-negative subtypes of *NPM1* and *CEBPA*³⁴.

7. Remarks on clinically recognized oncogenic fusions

The above data highlights a few characteristics of clinically recognized oncogenic fusions such as *BCR-ABL1*: 1) to date all of these fusions are in-frame and activating (i.e., TSG does not belong to the category of oncogenic fusions); 2) promoter/enhancer-hijacking can be regarded as a different category of oncogenic fusion because they do not generate chimeric proteins; 3) these fusions are subtype-defining so that typically we see no more than one fusion per tumor, also known as mutual exclusivity rule³⁴ that will be discussed in section **SN 11g**; 4) despite extensive clonal evolution during the course of the life span of a tumor, subtype-defining oncogenic fusions typically remain intact; 5) like *ZFTA-RELA* (also known as *C11orf95-RELA*) and *TCF3-HLF*, these fusions are expected to be functionally sufficient and necessary to the host cancer cells; 6) not all human cancers are expected to have oncogenic fusions. Interestingly, to date all clinically recognized oncogenic fusions in pediatric cancers have supporting evidence from both DNA and RNA sequencing data whenever both data types are available, highlighting a critical bioinformatic pattern during technical evaluation of candidate oncogenic fusions.

8. Study design of this work

The above molecular mechanistic insights lead us to following strategy in this study design.

8.a) Tumor suppressor genes. Due to the diverse mutation types (including substitutions (SNVs), small insertion/deletions (Indels), copy number loss (CNVs), or structural alterations (SVs)) that can all lead to loss-of-function, we always rely on DNA sequencing (especially whole genome sequencing) to definitively ascertain the mutation status for TSGs. Although occasionally truncating mutations can be detected in RNAseq, we deem a whole-genome sequencing cohort would better serve the goal of comprehensively and unbiasedly studying etiology. In fact, we are currently drafting a manuscript on the signatures of rearrangements (SVs) using whole genome sequencing (WGS) in >1,500 pediatric cancer patients. With this consideration, we decided to NOT include tumor suppressor gene in this study, which is designed to focused on oncogenic fusions like *BCR-ABL1*. However, in response to Reviewer #2's request, we analyzed highly frequent *CDKN2A* and *NBAS* truncating fusions in section **SN 12**.

8.b) Oncogenic fusions in the category of promoter/enhancer hijacking. In this category, a rearrangement can bring a strong promoter/enhancer to an otherwise silenced proto-oncogene and lead to its aberrant expression. When the novel promoter is far, the proto-oncogene may start its transcription from its own transcription start site, thereby leaving no split reads or discordant read pairs in RNAseq data for bioinformatic detection (**Fig. SN2a**). On the other hand,

the transcripts may contain part of the novel promoter sequences when the novel promoter is closer (Fig. SN2b). Moreover, it is also possible that a point mutation in the native promoter can convert it to a strong active promoter (such as *TAL1* in pediatric T-ALL⁴⁵) and lead to aberrant expression (Fig. SN2c)— biologically this scenario is not promoter/enhancer *per se*. Clearly, without DNA (preferentially whole genome) sequencing data, scenarios a) and c) cannot be resolved by transcriptome sequencing and, a forced analysis will result in biased conclusions that does not meet the scientific rigor requirement of this journal. Instead, such patterns are best studied in our ongoing project on the signatures of rearrangements (SVs) using whole genome sequencing in >1,500 pediatric cancer patients. Nevertheless, we provided results on known oncogenic fusions (*CRLF2*, *DUX4*, *EPOR*, *BCL11B*, Table SN2a-e) in promoter/enhancer-hijacking category to address Reviewer #2's request though we did not perform systematic discovery.

Fig. SN2. Promoter/enhancer-hijacking. In hypothetical scenario (a), the blue chromosome (and a strong enhancer/promoter highlighted by blue oval) was brought proximity to the orange proto-oncogene and lead to its aberrant expression. Transcription only involves the proto-oncogene due to the space between the blue promoter and orange gene. In (b), the blue promoter contacts the transcription start site of orange gene, so that the transcription involves both the proto-oncogene and a small part of the blue chromosome. In (c), a point mutation in the promoter region may convert it to a strong active promoter to initiate the proto-oncogene without a fusion event (such as *TAL1* enhancer mutation in Mansour et al (2014)⁴⁵). Split reads or discordant read pairs are expected for scenario (b) but not scenario (a) or (c). DNA (preferentially whole genome) sequencing are needed to ascertain the fusion status. Nevertheless, the aberrant high expression of such proto-oncogene typically can help ascertain the tumor subtype.

8.c) Conventional oncogenic fusions that generate chimeric proteins. As shown in our Fig. 1a, oncogenic fusions that generate chimeric proteins are obligated to have split read or discordant read pair signals in RNAseq data, either polyT protocol or total RNA protocol. It is this exact category that our large cohort of 5,190 RNAseq datasets can be used to generate scientifically rigor discoveries.

2.2a Another major shortcoming of the manuscript is that it continues to be heavily biased towards fusions selected by the authors as relevant to pediatric cancer based on having a gene partner or partners that have previously been defined as oncogenes – now made explicitly clear by the recent revision.

[Response] We appreciate this critique from Reviewer #2's. It significantly improved our work. Indeed, manual review of raw predictions from software tools (with good sensitivity) has been our routine

practice, even though there are thousands of samples. With many years of experience in such tasks, we can analyze many samples quickly with a high accuracy. However, we do realize that this practice is difficult to be transferred to the research community in general, thus compromising its reproducibility. Also, human error can happen during the manual review process. Therefore, in this revision we converted our many years of manual review experience into a highly efficient and effective computational pipeline, as detailed in **Method** section of main text (and Supplementary Notes). This method takes advantage of existing knowledge on some known oncogenic fusions to develop effective filters to greatly reduce the number of predictions from >5.7 million to 7,769 so that unbiased manual inspection is feasible for most researchers. Of note, in our first submission we detected fusions in 1,988 patients, while in this new method we detected fusions in 2,005 patients. The gain of 17 additional detections (0.8%) indicates not only the robustness of our previous manual review but also the power of the novel method.

Because of the large number (5,781,630) of predicted fusions from these four methods, manual inspection is impractical, if not impossible. We therefore developed a novel workflow (see detailed design principles and algorithmic descriptions in **Supplementary Notes**) using majority voting (a prediction is considered to have k votes if it is detected by k methods) to enable the effective and efficient detection of oncogenic fusions from 5,190 patients. This workflow has eight critical considerations. First, mutual exclusivity among oncogenic fusions. Using 63 well-known oncogenic fusions (Supplementary Note 10, **Supplementary Table 19**), we determined 1,743 patients to harbor ≥ 1 of these fusions detected by ≥ 2 detection methods. Interestingly, only 4 (0.23%) patient samples harbor ≥ 2 fusions, which indicates that each patient tumor typically harbors no more than 1 oncogenic fusions. Second, harmonizing coordinate differences among methods. By comparing predictions between methods, we determined that different methods can have ~ 10 nt differences in their predicted fusion coordinates (**Supplementary Notes 11a**). Third, multiple calls of the same oncogenic fusion pairs. As demonstrated in this work, intronic versioning is observed in many fusion pairs. Clearly, each intronic version corresponds to a unique prediction. Depending on the signal strength (number of supporting reads), some methods may “miss” a low abundance version (thus a low vote count) although other high abundance versions are commonly detected. By focusing on the high abundance versions, we determined that >93% of oncogenic fusion versions have 3+ votes. Fourth, although there are >5.7 million predictions, only 0.3% (16,348) predictions have 3+ votes. This data renders manual inspection possible by focusing on high-vote predictions, with a false negative rate <7%. Fifth, with the mutual exclusivity rule, we can establish blacklist by collecting all predictions from 1,743 fusion-positive patients that do not match any known fusions. Such blacklist allows us to further filter common artefacts such as readthrough¹², with a negligible false negative rate of 7%. Sixth, manual review of the remaining 7,769 predictions with 3+ votes. As detailed in **Supplementary Notes**, priority was given to in-frame predictions with $n \geq 2$ recurrence in our cohort, or with literature support, such as previously published childhood cancer studies (pan-cancer analyses^{15,16} and a number of disease-focused analyses including ependymoma¹⁷, Ewing sarcoma^{18,19}, rhabdomyosarcoma²⁰, low grade glioma²¹, high grade glioma²², T-cell acute lymphoblastic lymphoma²³, acute myeloid leukemia²⁴, as well as a recent clinical genomics report²⁵ and other literature^{5,9,15-22,24,26-34}). Quality indication from these methods were also considered. This review takes about 5 hours for an experience scientist. Seventh, upon manual review, we used the comprehensive list of oncogenic fusions to run a second round of systematic detection (rescue) as in the first consideration (those detected by 2 methods are included). This step allows us to minimize the impact of hard threshold of “ ≥ 3 ”

votes". Upon this comprehensive identification, we re-analyzed and further confirmed the mutual exclusivity rule: only 7 (0.35%) of 2,005 chimeric fusion-positive patients have 2+ oncogenic fusions. Eighth, determining the functional orientation of oncogenic fusions. When a patient tumor has balanced translocations, there might be reciprocal fusions, such as *ETV6-RUNX1* and *RUNX1-ETV6*. By collecting the number of patient samples with each orientation detected, we determined that all 52 fusions detected in 4+ patients have the clinically recognized orientation supported with higher frequency than the other orientation. Collectively, we detected 272 unique oncogenic fusion gene pairs that can generate chimeric proteins (**Table SN3c**). We also reported promoter-hijacking fusions for 12 genes (**Table SN2e**) in this cohort.

2.2b The authors continue to not make available as supplementary material the complete list of fusion candidates identified by running the various fusion-finding algorithms, but instead provide their preselected list of fusions they deem as relevant for this analysis. Hence, we continue to find no actual reporting of prevalence of pediatric cancer relevant fusions mentioned above as we would expect of a comprehensive survey.

[Response] We regret that Reviewer #2 failed to find the complete list of fusions that is provided in our Revision (screenshot from revised manuscript and rebuttal letter shown below). In this revision, the zenodo link has been updated to <https://doi.org/10.5281/zenodo.7510612>.

531 bams (typically ~1 day) and we were able to finish the re-run of all four fusion detectors on all full
532 bams/fastqs of our full cohort in less than 3 months for revision. All raw output from the four fusion
533 detectors were deposited in [zenodo \(https://doi.org/10.5281/zenodo.7033077\)](https://doi.org/10.5281/zenodo.7033077).

2.3a Often, the number of fusions analyzed in any specific analysis total few in number, and this is due to the need for sufficient patient prevalence as part of the analysis. This is understandable, but also unfortunate, because it reduces the impact of the work. The abstract is written in a grandiose style with little to no specific details reported. It is only when reading the main text that one discovers that only in-frame coding fusions were examined, up to 20 fusions were explored for impacts on splicing, and a single fusion was found to have clinically relevant prognostics. It would be best to include specifics in the abstract. Instead of 'a subset of oncogenic fusions', indicate which ones and/or how many.

[Response] We respectfully disagree with the comment that "the number ... total few in number." It is well known that prevalence of cancer driver genes has a "long tail" nature in almost all human cancers such as prostate cancer (<https://pubmed.ncbi.nlm.nih.gov/29610475/>), where only a few genes drive the development of tumors in many patients, and many more cancer genes only affect a few patients. Similarly, in this work, among the 272 gene pairs identified, only a few fusion pairs have sufficient number of samples for statistical analysis. The rare gene pairs can only be studied by using specifically designed patient cohorts. We have revised the abstract as following.

Oncogenic fusions formed through chromosomal rearrangements are hallmarks of childhood cancer that define cancer subtype, predict outcome, persist through treatment, and can be ideal therapeutic targets. However, mechanistic understanding of the etiology of oncogenic fusions remains elusive. Here we report a comprehensive detection of 272 oncogenic fusion gene pairs by using tumor transcriptome sequencing data from 5,190 childhood cancer patients. We identified diverse factors, including translation frame, protein domain, splicing, and gene length, that shape the formation of oncogenic fusions. Our mathematical modeling revealed a strong

link between differential selection pressure and clinical outcome in *CBFB-MYH11*. We discovered 4 oncogenic fusions, including *RUNX1-RUNX1T1*, *TCF3-PBX1*, *CBFA2T3-GLIS2*, and *KMT2A-AFDN*, with promoter-hijacking-like features that may offer novel strategies for therapeutic targeting. We uncovered extensive alternative splicing in oncogenic fusions including *KMT2A-MLLT3*, *KMT2A-MLLT10*, *C11orf95-RELA*, *NUP98-NSD1*, *KMT2A-AFDN* and *ETV6-RUNX1*. Strikingly, we discovered novel splice sites in 18 oncogenic fusion gene pairs and demonstrated that such splice sites confer novel therapeutic vulnerability for etiology-based genome editing. Our study reveals general principles on the etiology of oncogenic fusions in childhood cancer and suggests profound clinical implications including novel etiology-based risk stratification and genome-editing-based therapeutics.

2.3b Also, related to this last point in the main text, it isn't indicated how many other fusions were tested using hazard ratios and not found significant.

[Response] In Discussion, we added below statement as a limitation of our study:

Fourth, we detected clear selection bias in intronic versioning and established a strong link of such selection bias with clinical outcomes for *CBFB-MYH11*. This data indicated a differential oncogenicity of corresponding fusions due to the inclusion/exclusion of protein domains. However, no other fusion pairs can be studied in this work due to lack of publicly available clinical outcome data for corresponding cases for additional candidates with selection bias (*ETV6-RUNX1*, *KIAA1549-BRAF*, *NUT214-ABL1*, and *BCR-ABL1*). It would be interesting to test these candidate pairs in future dedicated studies.

2.4a The finding that patient prevalence of *KMT2A* fusion partners is well correlated with gene length is quite interesting and clearly significant. However, no specific evidence is given that other oncogenic fusions behave similarly.

[Response] We have added this analysis to two additional genes *ETV6* and *PAX5* (that have ≥ 5 fusion partner genes) that allowed statistical analysis. In section "Landscape of childhood oncogenic fusions" of main text, we added below:

By limiting the analysis to leukemia with oncogenic fusions with ≥ 5 fusion partners (*KMT2A*⁶¹, *ETV6*, and *PAX5*), we obtained a statistically significant linear association for *KMT2A* either when fusions with recurrence >3 were considered (R-squared=0.82; $P=0.002$; Fig. 2i) or when fusions with recurrence >1 were considered (R-squared=0.86; $P=1.5 \times 10^{-5}$; Extended Data Fig. 2i) but not for *ETV6* and *PAX5* ($P>0.1$; Extended Data Fig. 2j-k). The above observations are also observed if only involved introns are considered (Extended Data Fig. 2i-r). Excluding *KMT2A* fusions resulted an insignificant association in leukemia ($P=0.22$; data not shown). The overall insignificant association between gene length and patient prevalence in oncogenic fusions (except *KMT2A*) indicated that additional molecular factors (such as protein domain, frame, and splicing) may play a major role in the formation or selection of oncogenic fusions, as will be demonstrated next.

We also added these data in Extended Data Fig. 2:

Extended Data Fig. 2 Impact of gene length on prevalence of oncogenic fusions. (a). DNA breakpoints can be detected from RNAseq data (see raw data in Supplementary Table 8). Shown in y-axis is the number of DNA breakpoints as a function of distance between DNA breakpoints detected from RNA and DNA sequencing data (n=43 samples with both data

available) shown in x-axis (in base pairs). Note that each sample (oncogenic fusion) has two DNA breakpoints (totaling 86 breakpoints), one from N' gene and another from C' gene. Majority of detections from RNA data perfectly match that detected from DNA data. DNA breakpoints are uniformly distributed in *ETV6-RUNX1* (b), *NUP98-NSD1* (c), and demonstrated clustered distribution in promoter region of *RUNX1T1* gene for oncogenic fusion *RUNX1-RUNX1T1* (d), in addition to those shown in Fig. 2j-i. Also shown are linear regression between gene length and patient prevalence for fusions observed in leukemia (e), brain tumor (f), solid tumor (g) and all childhood cancer (h). In panels b-d, Bonferroni correction (Q values) is applied to the raw P values to account for multiple testing. For *KMT2A* (i), *ETV6* (j), and *PAX5* (k), there are sufficient number (≥ 5) of different fusion partner genes to study the association between patient prevalence and length of the partner genes. Only *KMT2A* partner genes resulted a significant association with R-squared of 0.86. Similar analysis was performed by only using the length of involved introns for leukemia (l), brain (m), solid (n), or all cancer types (o), as well as for partner genes for *KMT2A* (p), *ETV6* (q), and *PAX5* (r). Only *KMT2A* partner genes demonstrate significant association between intron length and patient prevalence (R-squared of 0.84). Removing *KMT2A* fusions from panels e and l resulted in similar insignificant P values (data not shown).

2.4b There may also be *KMT2A* fusion partners identified as candidate fusions in the initial survey but not selected by the authors for this particular analysis, but this is not clear given that the full set of initial fusion candidates was not reported.

[Response] We have added in section “**Landscape of childhood oncogenic fusions**” to highlight systematic analysis of *ETV6*, *PAX5*, and *KMT2A*:

We analyzed oncogenic fusions for leukemia, solid tumor, and brain tumor, separately. A marginally significant linear association (**Fig. 2h; Extended Data Fig. 2e**; R-squared=0.12; P=0.058) was observed between patient prevalence and total length of the involved gene pairs, and no significance were observed in brain and solid tumors (**Extended Data Fig. 2f-g**). By limiting the analysis to leukemia with oncogenic fusions with ≥ 5 fusion partners (*KMT2A*⁶¹, *ETV6*, and *PAX5*), we obtained a statistically significant linear association for *KMT2A* either when fusions with recurrence >3 were considered (R-squared=0.82; P=0.002; **Fig. 2i**) or when fusions with recurrence >1 were considered (R-squared=0.86; P=1.5 $\times 10^{-5}$; **Extended Data Fig. 2i**) but not for *ETV6* and *PAX5* (P >0.1 ; **Extended Data Fig. 2j-k**). The above observations are also observed if only involved introns are considered (**Extended Data Fig. 2l-r**). Excluding *KMT2A* fusions resulted an insignificant association in leukemia (P=0.22; data not shown). The overall insignificant association between gene length and patient prevalence in oncogenic fusions (except *KMT2A*) indicated that additional molecular factors (such as protein domain, frame, and splicing) may play a major role in the formation or selection of oncogenic fusions, as will be demonstrated next.

2.4c As mentioned in my earlier review, the correlation between gene length in leukemia fusions (p-value = 0.05) is tenuous, and there was no significant association found for the other tumor types analyzed. I suspect the *KMT2A* fusions contributed greatly towards the marginally significant finding with a p ≤ 0.05 threshold. Removing *KMT2A* fusions from that analysis would possibly make the finding non-significant.

[Response] We have added following analysis in section “**Landscape of childhood oncogenic fusions**” to highlight the contribution of *KMT2A* in leukemia:

We analyzed oncogenic fusions for leukemia, solid tumor, and brain tumor, separately. A marginally significant linear association (**Fig. 2h; Extended Data Fig. 2e**; R-squared=0.12; P=0.058) was observed between patient prevalence and total length of the involved gene pairs, and no significance were observed in brain and solid tumors (**Extended Data Fig. 2f-g**). By limiting the analysis to leukemia with oncogenic fusions with ≥ 5 fusion partners (*KMT2A*⁶¹, *ETV6*, and *PAX5*), we obtained a statistically significant linear association for *KMT2A* either when fusions with recurrence >3 were considered (R-squared=0.82; P=0.002; **Fig. 2i**) or when fusions with recurrence >1 were considered (R-squared=0.86; P=1.5 $\times 10^{-5}$; **Extended Data Fig. 2i**) but not for *ETV6* and *PAX5* (P >0.1 ; **Extended Data Fig. 2j-k**). The above observations are also observed if only involved introns are considered (**Extended Data Fig. 2l-r**). Excluding *KMT2A* fusions resulted an insignificant association in leukemia (P=0.22; data not shown). The overall insignificant association between gene length and patient prevalence in oncogenic fusions (except *KMT2A*) indicated that additional molecular factors (such as protein domain, frame, and splicing) may play a major role in the formation or selection of oncogenic fusions, as will be demonstrated next.

2.4c If the authors would make the data for these analyses available as supplementary materials, it would make it possible for others to more easily directly explore them.

[Response] We added Supplementary Tables (**Table ST21** for raw data of **Fig. 2** and **Extended Data Fig. 2** and **Table ST22** for **Fig. 5b**) to detail such intermediate results for readers to directly explore. This is no

need of such data for other figures/panels.

2.5a Reporting that DNA fusion breakpoints are found mostly to occur uniformly in relevant introns makes sense. Reporting that RNA-based fusion breakpoints are generally found within 5 bases of a DNA breakpoint does not. While I haven't examined the authors references for this finding, based on my own experience and based on basic splicing biology coupled with uniform DNA breakpoint data, it just simply does not make sense whatsoever. Uniform DNA breakpoints across long intronic regions cannot be within 5 bases on average from a splice junction. This is clear from the authors own figure 5a. I could be misinterpreting something here, because this is too bizarre to have a disagreement over.

[Response] We regret that Reviewer #2 misinterpreted the data, where the result on "5 bases" is a sanity check to ensure our DNA breakpoint detection from RNAseq data is reliable. We have added a section "Detecting DNA breakpoints from RNAseq data" in **Supplementary Note 9** as below to detail the rationale of our analysis.

In this work, we attempted to detect DNA breakpoints (here termed d-event) from RNAseq data to interrogate the uniformity of DNA breakpoints in relative intronic regions. As shown in the model of **Fig. SN 5**, although RNA splicing breakpoints (here termed m-event; commonly referred to as "fusion") are guaranteed to be observed in mRNA species that have underwent splicing, d-events are only occasionally observed in total RNA sequencing but NOT in poly(T)-based mRNA sequencing. In this work, we compared our d-event detections in RNA sequencing datasets against the ground truth d-events defined in DNA (whole genome) sequencing datasets and demonstrated that 91% of our detections are within 5-bp of ground truth. This accuracy enables us to reach reliable conclusion that DNA breakpoints are roughly uniformly distributed in relative introns of oncogenic fusions.

Fig. SN 5. Detecting DNA and RNA breakpoints from next generation sequencing data. (a) During gene expression, genetic information encoded in wildtype human chromosomes (gray, thin line indicates intron/intergenic region, thick boxes indicate exons) are first transcribed into pre-spliced transcripts (pre-mRNA; black), which is in turn spliced (to remove introns and retain exons) to generate mature RNA species (mRNA, green). (b) This Central Dogma also applies cancer genome (here blue and orange indicate two different chromosomal regions joined together by rearrangement). In this model, an intronic rearrangement (here termed as "d-event" to stress it is observed from DNA) happened between the two involved genes shown in blue and orange in the cancer genome. This d-event is observable in pre-mRNA; however, this d-event typically is not observed in mRNA because the intronic regions are spliced out. On

the other hand, in mRNA, the rearrangement is manifested as splicing junctions (here termed “m-event” to indicate it is observed in mRNA; commonly referred to as “fusion” events in bioinformatics) which typically are not directly observed in the DNA of cancer genome although biological inference is possible (with alternative splicing being the confounding factor in consideration). (c) In next generation sequencing, we can perform DNA sequencing (such as whole genome sequencing (WGS), targeted capture or Sanger sequencing, etc.) and RNA sequencing. Earlier RNA sequencing practices typically utilize poly(T) protocol, which can only interrogate mRNA species and therefore can NOT be used to detect d-events. Recent RNA sequencing practices typically utilize total-RNA protocol, which can simultaneously interrogate mRNA species and pre-mRNA species and enables simultaneous detection of d-event (if the total RNA contains sufficient pre-mRNA species and therefore is not guaranteed) and m-event.

2.5b The data for Figure 2a should be made available, and it should be examined to see if there are specific fusions that are biasing results, such as RUNX1/T1 where breakpoints are not found as uniform and the fusion is one of the most prevalent.

[Response] We believe this reviewer was referring our **Extended Data Fig. 2a** (but not Figure 2a) on the concordance between predicted DNA breakpoints from RNAseq data and those from DNAseq data. We have provided the raw data in **Supplementary Table 8** and we have updated the **Extended Data Fig. 2a** length as following:

Extended Data Fig. 2 Impact of gene length on prevalence of oncogenic fusions. (a). DNA breakpoints can be detected from RNAseq data (see raw data in **Supplementary Table 8**). Shown in y-axis is the number of DNA breakpoints as a function of distance between DNA breakpoints detected from RNA and DNA sequencing data (n=43 samples with both data available) shown in x-axis (in base pairs). Note that each sample (oncogenic fusion) has two DNA breakpoints (totaling 86 breakpoints), one from N' gene and another from C' gene. Majority of detections from RNA data perfectly match that detected from DNA data. DNA breakpoints are uniformly distributed in **ETV6-RUNX1** (b), **NUP98-NSD1** (c), and demonstrated clustered distribution in promoter region of **RUNX1T1** gene for oncogenic fusion **RUNX1-RUNX1T1** (d), in addition to those shown in **Fig. 2j-i**. Also shown are linear regression between gene length and patient prevalence for fusions observed in leukemia (e), brain tumor (f), solid tumor (g) and all childhood cancer (h). In panels **b-d**, Bonferroni correction (Q values) is applied to the raw P values to account for multiple testing. For **KMT2A** (i), **ETV6** (j), and **PAX5** (k), there are sufficient number (≥ 5) of different fusion partner genes to study the association between patient prevalence and length of the partner genes. Only **KMT2A** partner genes resulted a significant association with R-squared of 0.86. Similar analysis were performed by only using the length of involved introns for leukemia (l), brain (m), solid (n), or all cancer types (o), as well as for partner genes for **KMT2A** (p), **ETV6** (q), and **PAX5** (r). Only **KMT2A** partner genes demonstrate significant association between intron length and patient prevalence (R-squared of 0.84). Removing **KMT2A** fusions from panels e and l resulted in similar insignificant P values (data not shown).

2.6a The selection bias measurement has a null hypothesis stated as having patient prevalence proportional to intronic length. This seems intuitive, but I see no evidence presented for this null hypothesis to be valid. The authors provide evidence that DNA breakpoints are often found uniformly distributed at sites of fusions, and there is unconvincing evidence that (except for KMT2A fusions) that gene length is associated with patient prevalence, and so I suspect the authors are attempting to make a jump here in using these other limited data to support this null hypothesis.

[Response] We apologize for the confusion. We modified the justification in section “**Selection bias in intronic versioning**” to stress that our model only relies on the uniform distribution of DNA breakpoints (but not gene length).

Because intronic versioning can cause amino acid differences in the fusion protein which may in turn lead to potential functional difference, we hypothesized that intronic versioning could confer differential fitness to the host cells in some oncogenic fusions (**Supplementary Note 13**). To measure the effect size of potential selection bias between two intronic versions, we proposed a relative selection bias score (RSB; **Methods**) based on the observation that DNA breakpoints are distributed in relevant introns in a near-uniform fashion in 95% fusions (**Fig. 2j-i**; **Extended Data Fig. 2b-d**; **Supplementary Table 8**). In this model, the patient prevalence of DNA breakpoints falling in an intron should be proportional to its length if the resultant protein versions are functionally equivalent (i.e., confers the same positive selection pressure). The statistical significance of selection bias is measured by comparing (using a Chi-squared test) the observed patient prevalence in all intronic versions and corresponding expected patient prevalence under a null hypothesis that patient prevalence is proportional to intronic length. When the involved exon (**Fig. 5a**, star) encodes functionally important protein domain and thus lead to higher positive selection pressure, its corresponding intron will have disproportionately high patient prevalence.

Further, we added a section in **Supplementary Note 13** to illustrate the rationale of our analysis of selection bias:

13. Model of selection bias

Our study indicated that many fusion gene pairs have intronic versioning, where slightly different oncoproteins can be generated in different patient tumors. A natural question is whether such difference may carry biological significance such as oncogenicity. For this, we performed a theoretical analysis (Fig. SN5). Here, we introduced the concept of DNA rearrangement events before and after selection. First, DNA rearrangement events can happen either in a random fashion (Fig. SN5a) or in a non-random fashion (Fig. SN5b). Second, random DNA rearrangement events may be subjected to unbiased (Fig. SN5c) or biased (Fig. SN5d) selection, which will generate unbiased or biased patient prevalence among different intronic versions, respectively. Clearly, biased DNA rearrangement events (Fig. SN5b,e) can confound the statistical analysis and we therefore dropped *TCF3-PBX1* (Fig. 2I) and *RUNX1-RUNX1T1* (Extended Data Fig. 2d) from this analysis. Fusions with alternative splicing can also confound this analysis and are dropped (Fig. 5b).

Fig. SN5. Selection bias. DNA rearrangement events can happen in a random fashion (a; such as Fig. 2j-k) or in a non-random fashion due to an (unknown, such as the *TCF3-PBX1* example in Fig. 2I) molecular mechanism (b). In either scenario, two oncoproteins (A and B) are generated, regardless of the exact DNA breakpoints, due to splicing. Next, selection come to play. First, the oncogenicity of proteins A and B might be same. In this case, selection would be neutral between A and B versions, and the original un-biased rearrangements are mirrored in un-biased patient prevalence (c). Second, the oncogenicity of

proteins A and B might be different. In this case, selection bias will be reflected in biased patient prevalence (where version A has less frequency than expected; d). Clearly, selection bias cannot be analyzed when DNA rearrangement is biased (panels e and b) and when alternative splicing is observed (Fig. 5b).

2.6b Even so, based on my earlier recommendation, the authors included results based on not normalizing for intron length, and I found these results to be quite interesting, and rather than contributing false positives or false negatives, instead highlighting those fusions where the statistics are more likely to be robust indicators of relevant findings. Ultimately, while length normalization is intuitive, the authors have no compelling evidence that it is required, even though attempts are made to do so.

[Response] Please see our rebuttal in section 2.6c. Normalization is required under this rationale. We appreciate this comment that significantly improved our presentation.

Reviewer #3 (Remarks to the Author):

Thank you for the thoughtful edits in response to all reviewers. The authors have sufficiently responded to my comments.

Point-by-point response to reviewer comments

(Round 3)

Reviewer #2 (Remarks to the Author):

Thank you for the detailed response to my critiques. I appreciate the very thorough responses and the revisions to the manuscript, which address my earlier concerns. Very well done! and I look forward to seeing the revised manuscript in press.

- 1 Esgueva, R. *et al.* Prevalence of TMPRSS2-ERG and SLC45A3-ERG gene fusions in a large prostatectomy cohort. *Mod Pathol* **23**, 539-546, doi:10.1038/modpathol.2009.193 (2010).
- 2 Latysheva, N. S. & Babu, M. M. Discovering and understanding oncogenic gene fusions through data intensive computational approaches. *Nucleic Acids Res* **44**, 4487-4503, doi:10.1093/nar/gkw282 (2016).
- 3 Nowell, P. & Hungerford, D. A minute chromosome in human chronic granulocytic leukemia [abstract]. *Science* **132** (1960).
- 4 Rowley, J. D. Letter: A new consistent chromosomal abnormality in chronic myelogenous leukaemia identified by quinacrine fluorescence and Giemsa staining. *Nature* **243**, 290-293, doi:10.1038/243290a0 (1973).
- 5 Li, B. *et al.* Therapy-induced mutations drive the genomic landscape of relapsed acute lymphoblastic leukemia. *Blood* **135**, 41-55, doi:10.1182/blood.2019002220 (2020).
- 6 Ma, X. *et al.* Rise and fall of subclones from diagnosis to relapse in pediatric B-acute lymphoblastic leukaemia. *Nature communications* **6**, 6604, doi:10.1038/ncomms7604 (2015).

- 7 Druker, B. J. *et al.* Efficacy and safety of a specific inhibitor of the BCR-ABL tyrosine kinase in chronic myeloid leukemia. *N Engl J Med* **344**, 1031-1037, doi:10.1056/NEJM200104053441401 (2001).
- 8 Weinstein, I. B. Cancer. Addiction to oncogenes--the Achilles heel of cancer. *Science* **297**, 63-64, doi:10.1126/science.1073096 (2002).
- 9 Fischer, U. *et al.* Genomics and drug profiling of fatal TCF3-HLF-positive acute lymphoblastic leukemia identifies recurrent mutation patterns and therapeutic options. *Nat Genet* **47**, 1020-1029, doi:10.1038/ng.3362 (2015).
- 10 Haas, B. J. *et al.* Accuracy assessment of fusion transcript detection via read-mapping and de novo fusion transcript assembly-based methods. *Genome Biol* **20**, 213, doi:10.1186/s13059-019-1842-9 (2019).
- 11 Tian, L. *et al.* CICERO: a versatile method for detecting complex and diverse driver fusions using cancer RNA sequencing data. *Genome Biol* **21**, 126, doi:10.1186/s13059-020-02043-x (2020).
- 12 Uhrig, S. *et al.* Accurate and efficient detection of gene fusions from RNA sequencing data. *Genome Res* **31**, 448-460, doi:10.1101/gr.257246.119 (2021).
- 13 Haas, B. J. STAR-Fusion: Fast and Accurate Fusion Transcript Detection from RNA-Seq. *Preprint at <https://www.biorxiv.org/content/10.1101/120295v1>*. (2017).
- 14 Nicorici, D. e. a. FusionCatcher - a tool for finding somatic fusion genes in paired-end RNA-sequencing data. *Preprint at <https://www.biorxiv.org/content/early/2014/11/19/011650>* (2014).
- 15 Grobner, S. N. *et al.* The landscape of genomic alterations across childhood cancers. *Nature* **555**, 321-327, doi:10.1038/nature25480 (2018).
- 16 Ma, X. *et al.* Pan-cancer genome and transcriptome analyses of 1,699 paediatric leukaemias and solid tumours. *Nature* **555**, 371-376, doi:10.1038/nature25795 (2018).
- 17 Parker, M. *et al.* C11orf95-RELA fusions drive oncogenic NF-kappaB signalling in ependymoma. *Nature* **506**, 451-455, doi:10.1038/nature13109 (2014).
- 18 Tirode, F. *et al.* Genomic landscape of Ewing sarcoma defines an aggressive subtype with co-association of STAG2 and TP53 mutations. *Cancer Discov* **4**, 1342-1353, doi:10.1158/2159-8290.CD-14-0622 (2014).
- 19 Crompton, B. D. *et al.* The genomic landscape of pediatric Ewing sarcoma. *Cancer Discov* **4**, 1326-1341, doi:10.1158/2159-8290.CD-13-1037 (2014).
- 20 Shern, J. F. *et al.* Comprehensive genomic analysis of rhabdomyosarcoma reveals a landscape of alterations affecting a common genetic axis in fusion-positive and fusion-negative tumors. *Cancer Discov* **4**, 216-231, doi:10.1158/2159-8290.CD-13-0639 (2014).
- 21 Zhang, J. *et al.* Whole-genome sequencing identifies genetic alterations in pediatric low-grade gliomas. *Nat Genet* **45**, 602-612, doi:10.1038/ng.2611 (2013).
- 22 Wu, G. *et al.* The genomic landscape of diffuse intrinsic pontine glioma and pediatric non-brainstem high-grade glioma. *Nat Genet* **46**, 444-450, doi:10.1038/ng.2938 (2014).
- 23 Liu, Y. *et al.* The genomic landscape of pediatric and young adult T-lineage acute lymphoblastic leukemia. *Nat Genet* **49**, 1211-1218, doi:10.1038/ng.3909 (2017).
- 24 Bolouri, H. *et al.* The molecular landscape of pediatric acute myeloid leukemia reveals recurrent structural alterations and age-specific mutational interactions. *Nat Med* **24**, 103-112, doi:10.1038/nm.4439 (2018).
- 25 Newman, S. *et al.* Genomes for Kids: The Scope of Pathogenic Mutations in Pediatric Cancer Revealed by Comprehensive DNA and RNA Sequencing. *Cancer Discov*, doi:10.1158/2159-8290.CD-20-1631 (2021).
- 26 Andersson, A. K. *et al.* The landscape of somatic mutations in infant MLL-rearranged acute lymphoblastic leukemias. *Nat Genet* **47**, 330-337, doi:10.1038/ng.3230 (2015).

- 27 Faber, Z. J. *et al.* The genomic landscape of core-binding factor acute myeloid leukemias. *Nat Genet* **48**, 1551-1556, doi:10.1038/ng.3709 (2016).
- 28 Schwartz, J. R. *et al.* The genomic landscape of pediatric myelodysplastic syndromes. *Nature communications* **8**, 1557, doi:10.1038/s41467-017-01590-5 (2017).
- 29 Rusch, M. *et al.* Clinical cancer genomic profiling by three-platform sequencing of whole genome, whole exome and transcriptome. *Nature communications* **9**, 3962, doi:10.1038/s41467-018-06485-7 (2018).
- 30 Mouttet, B. *et al.* Durable remissions in TCF3-HLF positive acute lymphoblastic leukemia with blinatumomab and stem cell transplantation. *Haematologica* **104**, e244-e247, doi:10.3324/haematol.2018.210104 (2019).
- 31 Huang, B. J. *et al.* CFBF-MYH11 fusion transcripts distinguish acute myeloid leukemias with distinct molecular landscapes and outcomes. *Blood Adv* **5**, 4963-4968, doi:10.1182/bloodadvances.2021004965 (2021).
- 32 Montefiori, L. E. *et al.* Enhancer Hijacking Drives Oncogenic BCL11B Expression in Lineage-Ambiguous Stem Cell Leukemia. *Cancer Discov* **11**, 2846-2867, doi:10.1158/2159-8290.CD-21-0145 (2021).
- 33 Schwartz, J. R. *et al.* The acquisition of molecular drivers in pediatric therapy-related myeloid neoplasms. *Nature communications* **12**, 985, doi:10.1038/s41467-021-21255-8 (2021).
- 34 Umeda, M. *et al.* Integrated genomic analysis identifies UBTF tandem duplications as a recurrent lesion in pediatric acute myeloid leukemia. *Blood Cancer Discov*, doi:10.1158/2643-3230.BCD-21-0160 (2022).
- 35 Hunger, S. P., Devaraj, P. E., Foroni, L., Secker-Walker, L. M. & Cleary, M. L. Two types of genomic rearrangements create alternative E2A-HLF fusion proteins in t(17;19)-ALL. *Blood* **83**, 2970-2977 (1994).
- 36 Huang, Y. *et al.* The Leukemogenic TCF3-HLF Complex Rewires Enhancers Driving Cellular Identity and Self-Renewal Conferring EP300 Vulnerability. *Cancer Cell* **36**, 630-644 e639, doi:10.1016/j.ccell.2019.10.004 (2019).
- 37 Hunger, S. P. Chromosomal translocations involving the E2A gene in acute lymphoblastic leukemia: clinical features and molecular pathogenesis. *Blood* **87**, 1211-1224 (1996).
- 38 Hunger, S. P., Ohyashiki, K., Toyama, K. & Cleary, M. L. Hlf, a novel hepatic bZIP protein, shows altered DNA-binding properties following fusion to E2A in t(17;19) acute lymphoblastic leukemia. *Genes Dev* **6**, 1608-1620, doi:10.1101/gad.6.9.1608 (1992).
- 39 Chen, X. *et al.* Targeting oxidative stress in embryonal rhabdomyosarcoma. *Cancer Cell* **24**, 710-724, doi:10.1016/j.ccr.2013.11.002 (2013).
- 40 Roberts, K. G. *et al.* Targetable kinase-activating lesions in Ph-like acute lymphoblastic leukemia. *N Engl J Med* **371**, 1005-1015, doi:10.1056/NEJMoa1403088 (2014).
- 41 Lu, C. *et al.* The genomic landscape of childhood and adolescent melanoma. *J Invest Dermatol* **135**, 816-823, doi:10.1038/jid.2014.425 (2015).
- 42 Hyrenius-Wittsten, A. *et al.* De novo activating mutations drive clonal evolution and enhance clonal fitness in KMT2A-rearranged leukemia. *Nature communications* **9**, 1770, doi:10.1038/s41467-018-04180-1 (2018).
- 43 McLeod, C. *et al.* St. Jude Cloud: A Pediatric Cancer Genomic Data-Sharing Ecosystem. *Cancer Discov* **11**, 1082-1099, doi:10.1158/2159-8290.CD-20-1230 (2021).
- 44 Wang, J. *et al.* CREST maps somatic structural variation in cancer genomes with base-pair resolution. *Nat Methods* **8**, 652-654, doi:10.1038/nmeth.1628 (2011).
- 45 Mansour, M. R. *et al.* Oncogene regulation. An oncogenic super-enhancer formed through somatic mutation of a noncoding intergenic element. *Science* **346**, 1373-1377, doi:10.1126/science.1259037 (2014).

- 46 Carroll, A. J. *et al.* Pre-B cell leukemia associated with chromosome translocation 1;19. *Blood* **63**, 721-724 (1984).
- 47 Hunger, S. P. *et al.* The t(1;19)(q23;p13) results in consistent fusion of E2A and PBX1 coding sequences in acute lymphoblastic leukemias. *Blood* **77**, 687-693 (1991).
- 48 Raimondi, S. C. *et al.* New recurring chromosomal translocations in childhood acute lymphoblastic leukemia. *Blood* **77**, 2016-2022 (1991).
- 49 Delattre, O. *et al.* Gene fusion with an ETS DNA-binding domain caused by chromosome translocation in human tumours. *Nature* **359**, 162-165, doi:10.1038/359162a0 (1992).
- 50 Tian, L. *et al.* Long-read sequencing unveils IGH-DUX4 translocation into the silenced IGH allele in B-cell acute lymphoblastic leukemia. *Nature communications* **10**, 2789, doi:10.1038/s41467-019-10637-8 (2019).
- 51 Northcott, P. A. *et al.* Enhancer hijacking activates GFI1 family oncogenes in medulloblastoma. *Nature* **511**, 428-434, doi:10.1038/nature13379 (2014).
- 52 Arabzade, A. *et al.* ZFTA-RELA Dictates Oncogenic Transcriptional Programs to Drive Aggressive Supratentorial Ependymoma. *Cancer Discov* **11**, 2200-2215, doi:10.1158/2159-8290.CD-20-1066 (2021).
- 53 Lawrence, M. S. *et al.* Mutational heterogeneity in cancer and the search for new cancer-associated genes. *Nature* **499**, 214-218, doi:10.1038/nature12213 (2013).
- 54 Zack, T. I. *et al.* Pan-cancer patterns of somatic copy number alteration. *Nat Genet* **45**, 1134-1140, doi:10.1038/ng.2760 (2013).
- 55 Vogelstein, B. *et al.* Cancer genome landscapes. *Science* **339**, 1546-1558, doi:10.1126/science.1235122 (2013).
- 56 Pui, C. H. & Evans, W. E. Acute lymphoblastic leukemia. *N Engl J Med* **339**, 605-615, doi:10.1056/NEJM199808273390907 (1998).
- 57 Pugh, T. J. *et al.* The genetic landscape of high-risk neuroblastoma. *Nat Genet* **45**, 279-284, doi:10.1038/ng.2529 (2013).
- 58 Brady, S. W. *et al.* Pan-neuroblastoma analysis reveals age- and signature-associated driver alterations. *Nature communications* **11**, 5183, doi:10.1038/s41467-020-18987-4 (2020).
- 59 Versteeg, I. *et al.* Truncating mutations of hSNF5/INI1 in aggressive paediatric cancer. *Nature* **394**, 203-206, doi:10.1038/28212 (1998).
- 60 Zhang, J. *et al.* A novel retinoblastoma therapy from genomic and epigenetic analyses. *Nature* **481**, 329-334, doi:10.1038/nature10733 (2012).
- 61 Marschalek, R. Systematic Classification of Mixed-Lineage Leukemia Fusion Partners Predicts Additional Cancer Pathways. *Ann Lab Med* **36**, 85-100, doi:10.3343/alm.2016.36.2.85 (2016).